# No-Regret Bandit Exploration
# based on Soft Tree Ensemble Model

**Shogo Iwazaki**
LY Corporation
Tokyo, Japan
siwazaki@lycorp.co.jp

**Shinya Suzumura**
LY Corporation
Tokyo, Japan
ssuzumur@lycorp.co.jp

## Abstract

We propose a novel stochastic bandit algorithm that employs reward estimates using a tree ensemble model. Specifically, our focus is on a soft tree model, a variant of the conventional decision tree that has undergone both practical and theoretical scrutiny in recent years. By deriving several non-trivial properties of soft trees, we extend the existing analytical techniques used for neural bandit algorithms to our soft tree-based algorithm. We demonstrate that our algorithm achieves a smaller cumulative regret compared to the existing ReLU-based neural bandit algorithms. We also show that this advantage comes with a trade-off: the hypothesis space of the soft tree ensemble model is more constrained than that of a ReLU-based neural network.

## 1 Introduction

The stochastic bandit framework is a powerful tool for addressing sequential decision-making tasks in uncertain environments. A significant challenge in applying stochastic bandits is managing large action spaces. For example, in recommendation systems, there is often a vast action space generated by various combinations of users and items [38]. Standard algorithms designed for finite-armed bandits are inadequate in these scenarios. Consequently, numerous studies have focused on structurally modeling the reward process and using limited observed data to estimate rewards for unobserved actions. These approaches include algorithms that employ estimation methods such as linear models [3, 5, 12], kernel regression [11, 32], and neural networks [30, 41], which are referred to as linear bandit (LB), kernel bandit (KB), and neural bandit (NB) respectively. The effectiveness of these algorithms largely depends on the accuracy of the underlying reward models. Therefore, developing the bandit algorithms that leverage suitable reward estimation models is crucial.

Motivated by these considerations, this paper explores the stochastic bandit algorithm using tree ensembles, a model type that has gained popularity following neural networks but remains relatively underexplored in the bandit context. Specifically, we focus on the soft tree ensemble model, which has recently been the subject of both practical and theoretical investigations and has demonstrated strong empirical performance on tabular data [18, 21, 22, 25, 28]. Unlike hard trees, which update decision rules greedily and sequentially, soft trees employ gradient descent to update decision rules for the entire tree. This characteristic of soft trees facilitates the extension of existing analyses of NB and ensures a no-regret performance under suitable assumptions.

**Related works.** In the field of stochastic bandits, prior research has established various structural assumptions about underlying rewards. For instance, the assumption of Lipschitz continuity of rewards is explored in Lipschitz bandits [8], linearity of rewards is examined in LB [3, 5, 12], and more generally, the assumption that rewards lie in a known reproducing kernel Hilbert space (RKHS) is studied in KB [11, 32].

38th Conference on Neural Information Processing Systems (NeurIPS 2024).

Our paper studies a type of bandit algorithm that employs a tree structure model, a topic with limited prior exploration. Féraud et al. [15] proposed a bandit algorithm using random forests, but the theory of their algorithm exhibits linear dependence on the number of actions, making it unsuitable for large action spaces. Elmachtoub et al. [14] introduced a Thompson sampling-style algorithm utilizing decision trees; however, their algorithm's construction relies on heuristics and does not provide a regret guarantee.

Additionally, our theory is closely related to NB. Zhou et al. [41] proposed an upper confidence bound (UCB) algorithm using a deep neural net (DNN) regressor, and Zhang et al. [40] extended this analysis to Thompson sampling. Their analysis yields a regret upper bound of $\tilde{\mathcal{O}}(\tilde{d}\sqrt{T})$, where $\tilde{d}$ denotes the effective dimension of the problem, and $\tilde{\mathcal{O}}(\cdot)$ represents an order notation that ignores logarithmic dependence. However, generally, DNNs employing ReLU activation functions lead to $\tilde{d} = \tilde{\mathcal{O}}(T^{(d-1)/d})$, resulting in super-linear growth of $\mathcal{O}(\tilde{d}\sqrt{T})$ regret, which becomes meaningless [23]. Several studies address this issue by employing algorithms in the form of a sup-variant of UCB [37] or phased elimination-style algorithms [7, 26], proving a regret upper bound of $\tilde{\mathcal{O}}(T^{(2d-1)/(2d)})$ [23, 24, 30]. These studies combine theoretical analysis via the neural tangent kernel (NTK) [4, 19] for DNN regression with regret analysis techniques from KB, constructing algorithms and performing regret analysis. Our proposed algorithm can be seen as a generalization of NB theory using a soft-tree regressor from DNN.

**Contributions.** Our contributions are as follows:

- In Sec. 3.1, we introduce a new UCB-based algorithm: soft tree-based upper confidence bound (ST-UCB), which leverages the soft tree ensemble model. This algorithm can be considered an extension of the existing NN-UCB algorithm [41], incorporating the theory of the tree neural tangent kernel (TNTK) in soft trees [21, 22]. To our knowledge, this paper represents the first effort to extend the theory of NB to a tree-based structural model.

- In Sec. 3.2, we derive several non-trivial properties of the soft tree ensemble model. These include the decay rates of eigenvalues of the TNTK (Lemma 3.1), concentration properties of TNTK (Lemma 3.2), and upper bounds on the spectral norm of the Hessian matrix (Lemma 3.3). Leveraging these results, we demonstrate that the ST-UCB algorithm achieves a regret of $\tilde{\mathcal{O}}(\sqrt{T})$ under appropriate regularity conditions.

- In Sec. 4, we elucidate the distinctions in properties and assumptions between the existing NN-UCB and ST-UCB algorithms. Specifically, while NN-UCB generally lacks a no-regret guarantee in general action (or context) spaces, ST-UCB consistently offers a no-regret guarantee across general action spaces. Additionally, we examine the relation between the hypothesis spaces induced by the TNTK and those induced by the NTK using ReLU activation. This comparison reveals that the hypothesis space derived from soft trees, although more constrained, may lead to lower regret.

## 2 Preliminaries

**Problem setting.** We consider a sequential decision-making problem whose goal is to maximize the total reward under bandit feedback. Let $f : \mathcal{X} \to \mathbb{R}$ be an unknown reward function, where $\mathcal{X} \subset \mathbb{R}^d$ is a finite set of action candidates. At each time step $t$, the environment reveals an action set $\mathcal{X}_t \subset \mathcal{X}$; thereafter, the learner chooses an action $\boldsymbol{x}_t$ and receives the corresponding reward $y_t = f(\boldsymbol{x}_t) + \epsilon_t$, where $\epsilon_t$ is a noise random variable whose mean is zero. As a performance metric, we adopt the pseudo cumulative regret $R_T := \sum_{t=1}^{T} [f(\boldsymbol{x}_t^*) - f(\boldsymbol{x}_t)]$, where $\boldsymbol{x}_t^* \in \arg\max_{\boldsymbol{x} \in \mathcal{X}_t} f(\boldsymbol{x})$. In our problem setup, the action set $\mathcal{X}_t$ is allowed to change at each step $t$. In addition to the standard bandit setup that assumes $\mathcal{X}_t = \mathcal{X}$, this formulation includes a contextual bandit setup by setting $\mathcal{X}_t = \{(\boldsymbol{c}_t, \boldsymbol{a}) \mid \boldsymbol{a} \in \mathcal{A}(\boldsymbol{c}_t)\}$, where $\boldsymbol{c}_t$ is a context vector at step $t$, and $\mathcal{A}(\boldsymbol{c}_t)$ is the corresponding action set.

**Soft tree ensemble.** At each time step $t$, our algorithm constructs a soft tree-based estimator of the reward function $f$. We describe the definition of soft trees based on Kanoh and Sugiyama [21]. Now, let us consider $M \in \mathbb{N}_+$ perfect binary trees whose depths are $\mathcal{D} \in \mathbb{N}_+$. Note that each tree has $\mathcal{N} := 2^{\mathcal{D}} - 1$ internal nodes and $\mathcal{L} := 2^{\mathcal{D}}$ leaf nodes. Furthermore, for technical reasons, we assume

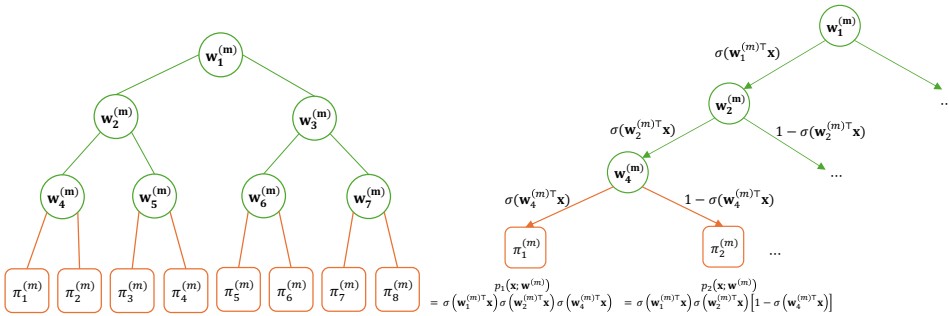

Figure 1: An illustrative image of a soft tree structure with $\mathcal{D} = 3$. As shown in the left plot, we have $\mathcal{N} := 2^{\mathcal{D}} - 1$ internal nodes (green) and $\mathcal{L} := 2^{\mathcal{D}}$ leaf nodes (orange), indexed using breadth-first ordering. The right plot shows an illustrative example where a soft tree calculates the weight probabilities $p_l(\cdot)$ for the leaf nodes.

that $M$ is an even number. Let $\boldsymbol{w}_n^{(m)} \in \mathbb{R}^d$ and $\pi_l^{(m)} \in \mathbb{R}$ be the parameters of the $n$-th internal and $l$-th leaf node of the $m$-th tree, respectively. We index these parameters according to breadth-first ordering, as described in the left plot of Fig. 1. Moreover, we also denote all internal and leaf node parameters as $\boldsymbol{w}^{(m)} := (\boldsymbol{w}_1^{(m)\top}, \ldots, \boldsymbol{w}_{\mathcal{N}}^{(m)\top})^\top \in \mathbb{R}^{\mathcal{N}d}$ and $\boldsymbol{\pi}^{(m)} := (\pi_1^{(m)}, \ldots, \pi_{\mathcal{L}}^{(m)})^\top \in \mathbb{R}^{\mathcal{L}}$. The output of a standard decision tree is obtained as the parameter of some leaf node, which is chosen deterministically based on the hard-splitting rules of internal nodes. On the other hand, the output of the soft tree is given by replacing the hard-splitting operation of the standard decision tree with a probabilistic one. Specifically, given parameters $\boldsymbol{\theta}^{(m)} := (\boldsymbol{w}^{(m)\top}, \boldsymbol{\pi}^{(m)\top})^\top$ and any input $\boldsymbol{x} \in \mathcal{X}$, the corresponding output $\tilde{h}(\boldsymbol{x}; \boldsymbol{\theta}^{(m)})$ of the $m$-th soft tree is defined as

$$\tilde{h}(\boldsymbol{x}; \boldsymbol{\theta}^{(m)}) = \sum_{l=1}^{\mathcal{L}} \pi_l^{(m)} p_l(\boldsymbol{x}; \boldsymbol{w}^{(m)}), \quad \text{where} \quad p_l(\boldsymbol{x}; \boldsymbol{w}) = \prod_{n=1}^{\mathcal{N}} \sigma(\boldsymbol{w}_n^\top \boldsymbol{x})^{\mathbb{1}_{l \swarrow n}} \left[1 - \sigma(\boldsymbol{w}_n^\top \boldsymbol{x})\right]^{\mathbb{1}_{n \searrow l}}.$$

Here, $\mathbb{1}_{l \swarrow n}$ and $\mathbb{1}_{n \searrow l}$ are indicator functions. If the $l$-th leaf node belongs to the left (resp. right) sub-tree whose root is the $n$-th internal node, $\mathbb{1}_{l \swarrow n}$ (resp. $\mathbb{1}_{n \searrow l}$) is one; otherwise, zero. Furthermore, $\sigma(\cdot) : \mathbb{R} \to [0, 1]$ is a *soft* decision function. The right plot of Fig. 1 shows an illustrative image of the calculation of $p_l(\cdot)$. As with [21], we use the scaled error function $\sigma(\boldsymbol{w}_n^\top \boldsymbol{x}) := \frac{1}{2}\text{erf}(\alpha \boldsymbol{w}_n^\top \boldsymbol{x}) + \frac{1}{2}$ with some pre-specified scaling parameter $\alpha \geq 0$, where $\text{erf}(b) = \frac{2}{\sqrt{\pi}} \int_0^b \exp(-z^2)\mathrm{d}z$ for any $b \in \mathbb{R}$. By aggregating $M$ soft trees, the whole output $h(\boldsymbol{x}; \boldsymbol{\theta})$ of the soft tree ensemble model is defined as $h(\boldsymbol{x}; \boldsymbol{\theta}) = \sum_{m=1}^{M} \tilde{h}(\boldsymbol{x}; \boldsymbol{\theta}^{(m)})/\sqrt{M}$, where $\boldsymbol{\theta} := (\boldsymbol{\theta}^{(1)\top}, \ldots, \boldsymbol{\theta}^{(M)\top})^\top \in \mathbb{R}^{M(d\mathcal{N}+\mathcal{L})}$. Under the model structures as described above, the training of the model parameters $\boldsymbol{\theta}$ is conducted based on the gradient descent optimizer, which aims to minimize some pre-specified loss functions. In our algorithm, we adopt a regularized square loss, whose detailed definition is given in Sec. 3.1.

**Neural tangent kernel theory for overparameterized model.** The neural tangent kernel (NTK) [19] is an effective theoretical tool for understanding the learning properties of overparameterized neural networks. Let $h_{\text{NN}}(\cdot; \boldsymbol{\theta}) : \mathbb{R}^d \to \mathbb{R}$ be a feed-forward neural network with a ReLU activation function, $L$ hidden layers whose width is $M$, and network parameters $\boldsymbol{\theta}$. Given any fixed inputs $\boldsymbol{x}, \tilde{\boldsymbol{x}} \in \mathbb{R}^d$, and $\tilde{\boldsymbol{\theta}}_0 \sim \mathcal{N}(\mathbf{0}, \boldsymbol{I})$, it has been shown that the inner product $\langle \nabla_{\boldsymbol{\theta}} h_{\text{NN}}(\boldsymbol{x}; \tilde{\boldsymbol{\theta}}_0), \nabla_{\boldsymbol{\theta}} h_{\text{NN}}(\tilde{\boldsymbol{x}}; \tilde{\boldsymbol{\theta}}_0) \rangle$ of gradients converges to a fixed kernel function $k_{\text{NTK}}(\boldsymbol{x}, \tilde{\boldsymbol{x}})$ (i.e., $\langle \nabla_{\boldsymbol{\theta}} h_{\text{NN}}(\boldsymbol{x}; \tilde{\boldsymbol{\theta}}_0), \nabla_{\boldsymbol{\theta}} h_{\text{NN}}(\tilde{\boldsymbol{x}}; \tilde{\boldsymbol{\theta}}_0) \rangle \xrightarrow{p} k_{\text{NTK}}(\boldsymbol{x}, \tilde{\boldsymbol{x}})$ as $M \to \infty$). The kernel function $k_{\text{NTK}}$ is called the NTK. Moreover, in the overparameterized regime, $h_{\text{NN}}(\boldsymbol{x}; \boldsymbol{\theta})$ trained with gradient descent with an infinitesimally small learning rate coincides with the kernel ridge-less regressor $h_{\text{NTK}}(\boldsymbol{x})$, whose kernel function is $k_{\text{NTK}}$ [4]. This property motivates us to analyze NB problems by bridging original NB to KB problems whose underlying kernel function is the NTK. Indeed, some existing works [23, 24, 30, 41] show the regret upper bound of NB problems by carefully combining NTK theory with existing theoretical tools of KB. In our paper, we consider soft tree variants of these existing works.

Recently, Kanoh and Sugiyama [21] generalized the NTK theory to the soft tree ensemble model. Let $\boldsymbol{g}(\boldsymbol{x}, \boldsymbol{\theta}) := \nabla_{\boldsymbol{\theta}} h(\boldsymbol{x}; \boldsymbol{\theta}) \in \mathbb{R}^p$ be the gradient vector of the soft tree ensemble model at parameter

---
**Algorithm 1** The soft tree-based upper confidence bound (ST-UCB) algorithm
---
**Input:** $\mathcal{X} \subset \mathbb{S}^{d-1}, \mathcal{D} \in \mathbb{N}_+, J \in \mathbb{N}_+, \eta > 0, \rho > 0, \alpha > 0, M \in \mathbb{N}_+, T \in \mathbb{N}_+, \beta > 0.$
 1: Initialize $\boldsymbol{\theta}_0$ randomly as described in Sec. 3.1.
 2: Define $\boldsymbol{G}_0 = \boldsymbol{0} \in \mathbb{R}^p$.
 3: **for** $t = 1, \ldots, T$ **do**
 4:     Obtain $\mathcal{X}_t$.
 5:     Calculate $\tilde{\sigma}_{t-1}^2(\boldsymbol{x}) \coloneqq \boldsymbol{g}(\boldsymbol{x}; \boldsymbol{\theta}_0)^\top \left(\boldsymbol{I}_p + \rho^{-1} \boldsymbol{G}_{t-1} \boldsymbol{G}_{t-1}^\top\right)^{-1} \boldsymbol{g}(\boldsymbol{x}; \boldsymbol{\theta}_0)$ on $\mathcal{X}_t$.
 6:     $\boldsymbol{x}_t \leftarrow \underset{\boldsymbol{x} \in \mathcal{X}_t}{\arg\max}\ [h(\boldsymbol{x}; \boldsymbol{\theta}_{t-1}) + \beta \tilde{\sigma}_{t-1}(\boldsymbol{x})].$
 7:     Obtain $y_t = f(\boldsymbol{x}_t) + \epsilon_t$.
 8:     $\boldsymbol{\theta}_t \leftarrow \text{TrainST}(t, \boldsymbol{\theta}_0, (\boldsymbol{x}_i, y_i)_{i \in [t]}, J, \eta, \rho, \mathcal{D}, \alpha, m).$
 9:     Define $\boldsymbol{G}_t = [\boldsymbol{g}(\boldsymbol{x}_1; \boldsymbol{\theta}_0), \ldots, \boldsymbol{g}(\boldsymbol{x}_t; \boldsymbol{\theta}_0)] \in \mathbb{R}^{p \times t}$
10: **end for**
---

---
**Algorithm 2** TrainST $(t, \boldsymbol{\theta}_0, (\boldsymbol{x}_i, y_i)_{i \in [t]}, J, \eta, \mathcal{D}, \alpha, M)$
---
 1: $\boldsymbol{\theta}_{t;0} \leftarrow \boldsymbol{\theta}_0$
 2: **for** $j = 1, \ldots, J$ **do**
 3:     Calculate gradient of $L_t(\boldsymbol{\theta}_{t;j-1}) \coloneqq \sum_{i=1}^t [h(\boldsymbol{x}_i; \boldsymbol{\theta}_{t;j-1}) - y_i]^2 + \rho \|\boldsymbol{\theta}_{t;j-1} - \boldsymbol{\theta}_0\|_2^2.$
 4:     Update parameter: $\boldsymbol{\theta}_{t;j} \leftarrow \boldsymbol{\theta}_{t;j-1} - \eta \nabla_{\boldsymbol{\theta}} L_t(\boldsymbol{\theta}_{t;j-1}).$
 5: **end for**
 6: **return** $\boldsymbol{\theta}_{t;J}.$
---

$\boldsymbol{\theta} \in \mathbb{R}^p$, where $p \coloneqq M(d\mathcal{N} + \mathcal{L})$ denotes the total number of parameters. Then, given fixed inputs $\boldsymbol{x}, \tilde{\boldsymbol{x}} \in \mathcal{X}$ and $\tilde{\boldsymbol{\theta}}_0 \sim \mathcal{N}(0, \boldsymbol{I}_p)$, the inner product $\langle \boldsymbol{g}(\boldsymbol{x}, \tilde{\boldsymbol{\theta}}_0), \boldsymbol{g}(\tilde{\boldsymbol{x}}, \tilde{\boldsymbol{\theta}}_0)\rangle$ has also been shown to converge in probability to some kernel function $k_{\text{TNTK}}(\boldsymbol{x}, \tilde{\boldsymbol{x}})$ as the number of ensemble models $M$ grows infinitely (see Theorem 1 in [21]). This limiting kernel $k_{\text{TNTK}}$ is called the *tree neural tangent kernel* (TNTK) as an analogy to the NTK and is defined as follows:

$$k_{\text{TNTK}}(\boldsymbol{x}, \tilde{\boldsymbol{x}}) = 2^{\mathcal{D}} \boldsymbol{x}^\top \tilde{\boldsymbol{x}} (\mathcal{T}(\boldsymbol{x}, \tilde{\boldsymbol{x}}))^{\mathcal{D}-1} \dot{\mathcal{T}}(\boldsymbol{x}, \tilde{\boldsymbol{x}}) + (2\mathcal{T}(\boldsymbol{x}, \tilde{\boldsymbol{x}}))^{\mathcal{D}}, \tag{1}$$

where:

$$\mathcal{T}(\boldsymbol{x}, \tilde{\boldsymbol{x}}) = \frac{1}{2\pi} \arcsin\left(\frac{\alpha^2 \boldsymbol{x}^\top \tilde{\boldsymbol{x}}}{\sqrt{(\alpha^2 \boldsymbol{x}^\top \boldsymbol{x} + 0.5)(\alpha^2 \tilde{\boldsymbol{x}}^\top \tilde{\boldsymbol{x}} + 0.5)}}\right) + \frac{1}{4}, \tag{2}$$

$$\dot{\mathcal{T}}(\boldsymbol{x}, \tilde{\boldsymbol{x}}) = \frac{\alpha^2}{\pi} \frac{1}{\sqrt{(1 + 2\alpha^2 \boldsymbol{x}^\top \boldsymbol{x})(1 + 2\alpha^2 \tilde{\boldsymbol{x}}^\top \tilde{\boldsymbol{x}}) - 4\alpha^4 (\boldsymbol{x}^\top \tilde{\boldsymbol{x}})^2}}. \tag{3}$$

It should be noted that even if we follow the existing NTK-based techniques of NB, generalizing the result of Kanoh and Sugiyama [21] to the analysis of sequential decision-making tasks is non-trivial. Specifically, the existing analysis of NB heavily relies on the following results of ReLU-based NTK: i) non-asymptotic bounds of NTK [4], ii) the spectral properties of the Hessian matrix around the initial model parameters [27], and iii) the upper bounds of maximum information gain (MIG) of NTK [35], which measure the complexity of the KB problem depending on the underlying kernel. These results are unique to DNN architectures with a ReLU-based activation function and are not applicable to the soft tree ensemble model.

## 3 UCB strategy based on soft tree ensemble model

### 3.1 Proposed algorithm: ST-UCB

The pseudo-code of our proposed algorithm, soft tree-based UCB (ST-UCB), is shown in Algorithm 1. ST-UCB is interpreted as the soft tree-based variant of NN-UCB [41]. We summarize each part of ST-UCB below.

**Initialization.** ST-UCB first chooses the initial parameter $\boldsymbol{\theta}_0 \in \mathbb{R}^p$ for the gradient descent method as follows. Let $\boldsymbol{\theta}_{\text{base}} \sim \mathcal{N}(0, \boldsymbol{I}_{p/2})$ be a base initial parameter, with $p = M(d\mathcal{N} + \mathcal{L})$. Using $\boldsymbol{\theta}_{\text{base}}$, we set the initial parameters $\boldsymbol{\theta}_0$ as $\boldsymbol{\theta}_0 = (\boldsymbol{\theta}_{0+}^\top, \boldsymbol{\theta}_{0-}^\top)^\top$, where $\boldsymbol{\theta}_{0+} \in \mathbb{R}^{p/2}$ and $\boldsymbol{\theta}_{0-} \in \mathbb{R}^{p/2}$ are defined as $\boldsymbol{\theta}_{0+} = (\boldsymbol{w}_{\text{base}}^{(1)\top}, \boldsymbol{\pi}_{\text{base}}^{(1)\top}, \ldots, \boldsymbol{w}_{\text{base}}^{(M/2)\top}, \boldsymbol{\pi}_{\text{base}}^{(M/2)\top})^\top$ and $\boldsymbol{\theta}_{0-} = (\boldsymbol{w}_{\text{base}}^{(M/2+1)\top}, -\boldsymbol{\pi}_{\text{base}}^{(M/2+1)\top}, \ldots, \boldsymbol{w}_{\text{base}}^{(M)\top}, -\boldsymbol{\pi}_{\text{base}}^{(M)\top})^\top$, respectively. This initialization procedure ensures that the initial model output is 0 (i.e., $h(\boldsymbol{x}; \boldsymbol{\theta}_0) = 0$ for all $\boldsymbol{x} \in \mathcal{X}$), which is essential for our theoretical analysis.

**Learning.** At each step $t$, ST-UCB learns the model parameter $\boldsymbol{\theta}_t$ based on a regularized squared loss $L_t(\boldsymbol{\theta}) := \sum_{i=1}^{t} (h(\boldsymbol{x}_i; \boldsymbol{\theta}) - y_i)^2 + \rho \|\boldsymbol{\theta} - \boldsymbol{\theta}_0\|_2^2$, where $\rho > 0$ is a regularization parameter.

**UCB-based selection of $\boldsymbol{x}_t$.** At each step $t$, ST-UCB selects $\boldsymbol{x}_t$ as follows:
$$\boldsymbol{x}_t \in \arg\max_{\boldsymbol{x} \in \mathcal{X}_t} [h(\boldsymbol{x}; \boldsymbol{\theta}_{t-1}) + \beta \tilde{\sigma}_{t-1}(\boldsymbol{x})], \tag{4}$$
where $\tilde{\sigma}_{t-1}^2(\boldsymbol{x}) = \boldsymbol{g}(\boldsymbol{x}; \boldsymbol{\theta}_0)^\top \left( \boldsymbol{I}_p + \rho^{-1} \boldsymbol{G}_{t-1} \boldsymbol{G}_{t-1}^\top \right)^{-1} \boldsymbol{g}(\boldsymbol{x}; \boldsymbol{\theta}_0)$ with $\boldsymbol{g}(\boldsymbol{x}; \boldsymbol{\theta}) := \nabla_{\boldsymbol{\theta}} h(\boldsymbol{x}; \boldsymbol{\theta}) \in \mathbb{R}^p$ and $\boldsymbol{G}_{t-1} := (\boldsymbol{g}(\boldsymbol{x}_1; \boldsymbol{\theta}_0), \ldots, \boldsymbol{g}(\boldsymbol{x}_{t-1}; \boldsymbol{\theta}_0)) \in \mathbb{R}^{p \times t}$. In ST-UCB, the quantity $\tilde{\sigma}_{t-1}^2(\boldsymbol{x})$ quantifies the uncertainty of the model output $h(\boldsymbol{x}; \boldsymbol{\theta}_t)$ and is essential for the construction of confidence bounds. Furthermore, the quantity $\tilde{\sigma}_{t-1}^2(\boldsymbol{x})$ is interpreted as the predictive variance of a Bayesian linear regression whose feature map is the gradient of the initial model output $h(\boldsymbol{x}; \boldsymbol{\theta}_0)$. We note that a similar quantity is leveraged in existing NB algorithms [23, 30, 41].

### 3.2 Theory of ST-UCB

**Assumptions for theoretical analysis.** We make the following assumptions for our theory:

**Assumption 3.1.** *(i) The output noise $\epsilon_t$ is conditionally $\sigma$-sub-Gaussian for some $\sigma > 0$. Specifically, $\mathbb{E}[\exp(\lambda \epsilon_t) \mid \mathcal{H}_{t-1}] \leq \exp(\lambda^2 \sigma^2/2)$ holds for any $t \in [T] := \{1, \ldots, T\}$ and any history $\mathcal{H}_{t-1} := (\boldsymbol{x}_1, y_1, \ldots, \boldsymbol{x}_{t-1}, y_{t-1})$. (ii) The input space $\mathcal{X} \subset \mathbb{R}^d$ is a subset of the hyper-sphere $\mathbb{S}^{d-1} := \{\boldsymbol{x} \in \mathbb{R}^d \mid \|\boldsymbol{x}\|_2 = 1\}$. (iii) The underlying reward function $f$ is an element of the RKHS corresponding to $k_{TNTK}$, where $k_{TNTK}$ is the TNTK induced by the same soft tree structure used in ST-UCB. (iv) The RKHS norm of $f$ is bounded by a known constant $B < \infty$. That is, $\|f\|_{\text{TNTK}} \leq B$ holds, where $\|\cdot\|_{\text{TNTK}}$ denotes the RKHS norm corresponding to $k_{TNTK}$.*

**Remark 3.1.** *In Assumption 3.1, (i) is the standard assumption for the stochastic bandit problem and is quite mild. For example, Bernoulli, Gaussian, and any bounded reward models are included in this assumption. Assumption (ii) is often assumed in existing NB literature [23, 24, 30, 40, 41] and holds without loss of generality by transforming the original input space through a bijection map. For example, given any original input space $\tilde{\mathcal{X}} \subset \mathbb{R}^d$, we can construct a new input space $\mathcal{X}$ on the hyper sphere $\mathbb{S}^d$ as $\mathcal{X} = \left\{ \left( \bar{l}^{-1} \tilde{\boldsymbol{x}}^\top, (1 - \|\tilde{\boldsymbol{x}}\|_2^2 \bar{l}^{-2})^{1/2} \right)^\top \mid \tilde{\boldsymbol{x}} \in \tilde{\mathcal{X}} \right\} \subset \mathbb{S}^d$, where $\bar{l} = \max_{\tilde{\boldsymbol{x}} \in \tilde{\mathcal{X}}} \|\tilde{\boldsymbol{x}}\|_2$. Assumptions (iii) and (iv) are similar to those in existing NB works [23, 24, 30]. The only difference is that we use TNTK instead of NTK to define the hypothesis space (RKHS) to which $f$ belongs. We omit the basic definition and properties of RKHS; see, e.g., [20] for details. In Sec. 4, we further discuss the relationship between the RKHSs corresponding to NTK and TNTK.*

Similar to NB with ReLU, our theoretical guarantees rely on two crucial tools in the context of KB. The first is the maximum information gain (MIG) [32], which quantifies the complexity of the problem in the context of kernel-based sequential decision-making tasks. MIGs depend on the underlying kernels, and their upper bounds have been provided when using well-known kernels, including the NTK corresponding to NNs with ReLU [23, 35, 36]. We show the upper bound of MIG when the underlying kernel is TNTK. The second tool is the confidence bound. Constructing valid confidence bounds is crucial for obtaining meaningful regret bounds in stochastic bandit algorithms. These two elements are not only essential for the theoretical analysis of ST-UCB but also of independent interest in general sequential decision-making problems. Hereafter, we present our MIG and confidence bounds results for our ST-UCB algorithm, concluding with the regret upper bound for ST-UCB.

**Maximum information gain (MIG) of TNTK.** Let us define the quantity $\gamma_T$ as
$$\gamma_T = \frac{1}{2} \max_{\boldsymbol{x}_1, \ldots, \boldsymbol{x}_T \in \mathcal{X}} \ln \det \left( \boldsymbol{I}_T + \rho^{-1} \boldsymbol{K}_T \right), \tag{5}$$

where $\boldsymbol{K}_T$ is the $T \times T$ kernel matrix whose $(i, j)$-th entry is $k_{\text{TNTK}}(\boldsymbol{x}_i, \boldsymbol{x}_j)$. This $\gamma_T$ is called the maximum information gain (MIG) since the quantity $0.5 \ln \det(\boldsymbol{I}_T + \rho^{-1} \boldsymbol{K}_T)$ is equal to the information gain from $T$ observations in a Gaussian process regression model, characterized by the covariance function $k_{\text{TNTK}}$ and the noise variance parameter $\rho$ [32]. The following Theorem 3.1 is our main result about MIG, which shows that $\gamma_T$ grows logarithmically.

**Theorem 3.1** (Upper bound of MIG of TNTK). *Fix any $\alpha \in (0, \infty)$, $d \geq 2$, $\mathcal{D} \in \mathbb{N}_+$, and $\mathcal{X} \subset \mathbb{S}^{d-1}$. Then, $\gamma_T = \mathcal{O}(\ln^d T)$. Here, the implied constant depends on $d$, $\alpha$, and $\mathcal{D}$.*

The proof of Theorem 3.1 is given in Appendix A.2. The analysis of MIG is well-studied in existing KB literature [32, 36]. The key component to quantify the upper bound of MIG is the decaying rate of the eigenvalues of the underlying kernel. The following lemma gives the decay rate of TNTK eigenvalues, which plays a central role in the proof of Theorem 3.1.

**Lemma 3.1** (Eigendecomposition of TNTK). *Fix any $d \geq 2$, $\alpha \in (0, \infty)$, and $\mathcal{D} \in \mathbb{N}_+$. Furthermore, let us define $N_{d,n}$ as $N_{d,n} = \frac{2n+d-2}{n} \binom{n+d-3}{d-2}$, for any $n \in \mathbb{N}$, where $\binom{a}{b} := \frac{a!}{b!(a-b)!}$ is a binomial coefficient. Then, for any $\boldsymbol{x}, \tilde{\boldsymbol{x}} \in \mathbb{S}^{d-1}$, the TNTK corresponding to $\alpha$ and $\mathcal{D}$ satisfies*

$$k_{\text{TNTK}}(\boldsymbol{x}, \tilde{\boldsymbol{x}}) = \sum_{n=0}^{\infty} \sum_{j=1}^{N_{d,n}} \lambda_n Y_{n,j}(\boldsymbol{x}) Y_{n,j}(\tilde{\boldsymbol{x}}), \tag{6}$$

*where $(\lambda_n)_{n \in \mathbb{N}}$ and $(Y_{n,j})_{n \in \mathbb{N}, j \in [N_{d,n}]}$ are eigenvalues and eigenfunctions of (the integral operator of) TNTK that satisfy $\lambda_0 \geq \lambda_1 \geq \cdots \geq 0$. In addition, for any $n \in \mathbb{N}$, the eigenvalue $\lambda_n$ satisfies*

$$\lambda_n \leq C_{\alpha, \mathcal{D}}^{(1)} \exp\left(-n\mathcal{D} \ln\left(1 + \frac{1}{4\alpha^2}\right)\right), \tag{7}$$

*where $C_{\alpha, \mathcal{D}}^{(1)} > 0$ is a constant, which depends on $\alpha$ and $\mathcal{D}$.*

**Remark 3.2.** *The eigenfunctions $(Y_{n,j})_{j \in [N_{d,n}]}$ are known as* spherical harmonics *of degree $n$ with multiplicity $N_{d,n}$ (see, e.g., [13]). Furthermore, on the hyper-sphere $\mathbb{S}^{d-1}$, the kernels that have rotationally invariant form can be represented in the form of Eq. (6). TNTK and NTK with ReLU activation function are included in the rotationally invariant class of kernels; therefore, NTK can also be decomposed as Eq. (6) [35], while corresponding eigenvalues differ from those of TNTK.*

The proof of Lemma 3.1 is given in Appendix A.1. Lemma 3.1 demonstrates the exponential eigenvalue decay of TNTK, in contrast to the polynomial eigenvalue decay of NTK with ReLU activation [6, 35]. This difference leads to faster convergence of ST-UCB compared to NN-UCB, albeit with a smaller corresponding RKHS of TNTK. We discuss more details in Sec. 4.

**Confidence bound.** The following shows the confidence bounds for the soft tree-based model.

**Theorem 3.2** (Confidence bounds based on the soft tree ensemble model). *Suppose Assumption 3.1 holds. Fix any $\delta \in (0, 1)$, $\rho > 0$, $\alpha \geq 1$, and $\mathcal{D} \geq 2$. Let $\boldsymbol{K}_{\text{TNTK}}(\mathcal{X}) := [k_{\text{TNTK}}(\boldsymbol{x}, \tilde{\boldsymbol{x}})]_{\boldsymbol{x}, \tilde{\boldsymbol{x}} \in \mathcal{X}} \in \mathbb{R}^{|\mathcal{X}| \times |\mathcal{X}|}$ and $\lambda_0 = \lambda_{\min}(\boldsymbol{K}_{\text{TNTK}}(\mathcal{X})) > 0$ be the kernel matrix over $\mathcal{X} \times \mathcal{X}$ and the minimum eigenvalue of $\boldsymbol{K}_{\text{TNTK}}(\mathcal{X})$, respectively. If the number of soft tree ensemble models $M$ is sufficiently large to satisfy $M \geq \text{Poly}(T, \rho^{-1}, B, \alpha, 2^{\mathcal{D}}, \lambda_0^{-1}, |\mathcal{X}|, \ln(1/\delta))$ and the learning rate $\eta$ satisfies $\eta \leq \mathcal{O}((T^2 2^{4\mathcal{D}} \alpha^2 \ln(M/\delta) + \rho)^{-1})$, then, the following event holds with probability at least $1 - \delta$:*

$$\forall t \in [T], \forall \boldsymbol{x} \in \mathcal{X}, |f(\boldsymbol{x}) - h(\boldsymbol{x}; \boldsymbol{\theta}_{t-1})| \leq \mathcal{O}\left(\frac{T^2 (\ln T)^2 (\ln M)}{\sqrt{M}}\right) + \beta \tilde{\sigma}_{t-1}(\boldsymbol{x}), \tag{8}$$

*where:*

$$\beta = \mathcal{O}\left(\sqrt{\gamma_T + \frac{T^{3/2}}{M^{1/2}}} + \frac{T^3 (\ln T)(\ln M^{3/2})}{\sqrt{M}} + T^{3/2}(\ln T)(\ln M)(1 - 2\eta\rho)^{J/2}\right). \tag{9}$$

**Remark 3.3.** *The minimum eigenvalue $\lambda_0$ of the kernel matrix of TNTK is guaranteed to be strictly positive if $\mathcal{X} \subset \mathbb{S}^{d-1}$. See Proposition 1 in [21].*

We provide the proof of Theorem 3.2 in Appendix B.3 with the precise conditions about $M$ and the dependence of constant factors. Our proof strategy for Theorem 3.2 follows the existing analysis of confidence bounds in NB works; however, the application of their proof techniques to the soft tree regressor is not straightforward. Specifically, the existing proof of the confidence bounds in NB depends on the concentration results of NTK (Theorem 3.1 in [4]), and the spectral norm bounds of the Hessian matrix of NN (Theorem 3.2 in [27]). To prove Theorem 3.2, we provide the following soft tree versions of their results.

**Lemma 3.2** (Concentration to TNTK). *Fix any $\boldsymbol{x}, \tilde{\boldsymbol{x}} \in \mathbb{S}^{d-1}$, $\delta \in (0,1)$, and $\epsilon \in (0, C_{\alpha,\mathcal{D}}^{(2)})$ with $C_{\alpha,\mathcal{D}}^{(2)} = 2^{2\mathcal{D}+2}\alpha^2 C$. If $M \geq \tilde{C}\max\{C_{\alpha,\mathcal{D}}^{(2)2}, 2^{2\mathcal{D}}\}\epsilon^{-2}\ln(16/\delta)$, then,*

$$\mathbb{P}(|k_{\mathrm{TNTK}}(\boldsymbol{x}, \tilde{\boldsymbol{x}}) - \langle \boldsymbol{g}(\boldsymbol{x}, \boldsymbol{\theta}_0), \boldsymbol{g}(\tilde{\boldsymbol{x}}, \boldsymbol{\theta}_0)\rangle| \leq 4\epsilon) \geq 1 - \delta, \tag{10}$$

*where $\boldsymbol{\theta}_0$ is the initial parameter of ST-UCB, and $C, \tilde{C} > 0$ are absolute constants.*

**Lemma 3.3** (Spectral norm upper bound). *For any $\delta \in (0,1)$ and $\alpha \geq 1$, with probability at least $1 - \delta$, the following holds for any $R > 0$, $\boldsymbol{\theta} \in \mathbb{R}^p$, and $\boldsymbol{x} \in \mathbb{S}^{d-1}$:*

$$\|\boldsymbol{\theta} - \boldsymbol{\theta}_0\|_2 \leq R \Rightarrow \|\boldsymbol{H}(\boldsymbol{x}, \boldsymbol{\theta})\| \leq \frac{C_{\alpha,\mathcal{D}}^{(3)}(R + \sqrt{2})^2}{\sqrt{M}}\ln\frac{2^{\mathcal{D}+2}M}{\delta}, \tag{11}$$

*where $\boldsymbol{H}(\boldsymbol{x}, \boldsymbol{\theta}) := \nabla_{\boldsymbol{\theta}}^2 h(\boldsymbol{x}; \boldsymbol{\theta}) \in \mathbb{R}^{p \times p}$ is the Hessian matrix of the model output, and $C_{\alpha,\mathcal{D}}^{(3)} = \sqrt{6}\alpha^2 2^{2\mathcal{D}}$. Furthermore, for any $\boldsymbol{A} \in \mathbb{R}^{p \times p}$, $\|\boldsymbol{A}\| := \max_{\boldsymbol{z} \in \mathbb{S}^{p-1}}\|\boldsymbol{A}\boldsymbol{z}\|_2$ denotes the spectral norm.*

The proofs of Lemma 3.2 and Lemma 3.3 are given in Appendix B. By carefully combining Lemma 3.2 and Lemma 3.3 with the existing proof strategy of NB, we derive Theorem 3.2. The overview of the proof is summarized in Appendix B.3.1.

**Regret upper bound of ST-UCB.** By combining Theorem 3.1 and Theorem 3.2 with the standard proof technique of the kernelized UCB algorithm, we obtain the $\tilde{\mathcal{O}}(\sqrt{T})$ regret upper bound for ST-UCB as stated in the following theorem. The proof is provided in Appendix C.

**Theorem 3.3.** *Suppose that Assumption 3.1 holds. Fix any $\delta \in (0,1)$, $\alpha \geq 1$, $\rho > 0$, and $\mathcal{D} \geq 2$. Furthermore, assume that the confidence width parameter $\beta$ satisfies Eq. (9). If the number of soft tree ensemble models $M$ and the total step size $J$ of the gradient descent are sufficiently large to satisfy $M \geq \mathrm{Poly}(T, \rho^{-1}, B, \alpha, 2^{\mathcal{D}}, \lambda_0^{-1}, |\mathcal{X}|, \ln(1/\delta))$, and the learning rate $\eta$ satisfies $\eta \leq \mathcal{O}((T^2 2^{4\mathcal{D}}\alpha^2 \ln(M/\delta) + \rho)^{-1})$, then, the following holds with probability at least $1 - \delta$:*

$$R_T \leq 1 + \left(\sqrt{2}B + 1 + \frac{\sigma}{\sqrt{\rho}}\sqrt{2\left(\gamma_T + 1 + \ln\frac{6}{\delta}\right)}\right)\sqrt{\frac{8T(\gamma_T + 1)}{\ln(1 + \rho^{-2})}} = \mathcal{O}\left(\sqrt{T}\ln^d T\right). \tag{12}$$

# 4   Comparison of NN-UCB and ST-UCB

**Comparison of regret.** In the existing NN-UCB algorithm [41], a regret upper bound of $\mathcal{O}(\tilde{d}\sqrt{T})$ is provided, where $\tilde{d}$ represents the effective dimension of ReLU-based NTK. It is generally known that the worst-case bound of the effective dimension and MIG are equivalent up to logarithmic dependencies [37]. Considering the upper bound on MIG of NTK, $\gamma_T^{(\mathrm{NTK})} = \tilde{\mathcal{O}}(T^{(d-1)/d})$ [23, 35], the regret of NN-UCB becomes $\tilde{\mathcal{O}}(T^{(d-1)/d+1/2})(= \tilde{\mathcal{O}}(\gamma_T^{(\mathrm{NTK})}\sqrt{T}))$. This results in a super-linear regret, and meaningful guarantees for NN-UCB are not achievable without further restricted assumptions on the input set $\mathcal{X}_t$ (e.g., see the discussion in Appendix D in [40]). To address these issues in a general setting, it is necessary to construct more complex algorithms that incorporate concepts such as a sup-variant of UCB [23, 30] or phased elimination [24], yielding a regret upper bound of $\mathcal{O}(\sqrt{\gamma_T^{(\mathrm{NTK})}T})$. In contrast, due to Theorem 3.1, the MIG of TNTK $\gamma_T^{(\mathrm{TNTK})}$ diverges on a logarithmic scale. Therefore, ST-UCB achieves a regret bound of $\tilde{\mathcal{O}}(\sqrt{T})$ without requiring additional assumptions on the input set $\mathcal{X}_t$, maintaining a simple UCB-style algorithmic structure.

**Comparison of hypothesis space.** In our analysis, we assume in Assumption 3.1 that the reward function $f$ belongs to the RKHS $\mathcal{H}_{\text{TNTK}}$ associated with TNTK. Conversely, in existing NB research, it is assumed that $f$ belongs to the RKHS $\mathcal{H}_{\text{NTK}}$ associated with NTK. By combining Lemma 3.1 with the well-known Mercer's representation theorem (e.g., Theorem 4.51 in [33]), we derive the following lemma, which describes the relationship between $\mathcal{H}_{\text{TNTK}}$ and $\mathcal{H}_{\text{NTK}}$.

**Lemma 4.1.** *Fix any $\alpha \geq 0$ and $\mathcal{D} \in \mathbb{N}_+$, and define the corresponding TNTK as $k_{\text{TNTK}}$ : $\mathbb{S}^{d-1} \times \mathbb{S}^{d-1} \to \mathbb{R}$. Let $k_{\text{NTK}} : \mathbb{S}^{d-1} \times \mathbb{S}^{d-1} \to \mathbb{R}$ be an NTK corresponding to a ReLU-based $L$-layer neural network structure, where $L$ is any natural number. Then, $\mathcal{H}_{\text{TNTK}} \subset \mathcal{H}_{\text{NTK}}$ holds, where $\mathcal{H}_{\text{NTK}}$ and $\mathcal{H}_{\text{TNTK}}$ are RKHSs corresponding to $k_{\text{NTK}}$ and $k_{\text{TNTK}}$, respectively.*

The proof of Lemma 4.1 is provided in Appendix D. Lemma 4.1 indicates that the regret upper bound of ST-UCB is guaranteed in a more constrained hypothesis space compared to NN-UCB. While NN-UCB generally does not guarantee a no-regret property, the $\tilde{\mathcal{O}}(\sqrt{T})$ guarantee in ST-UCB can be interpreted as being due to focusing on a more constrained hypothesis space.

It should be noted that whether this property is specific to the tree structure of the model or depends on the choice of the soft-decision function is unknown. We constructed and analyzed our algorithm based on the definition of soft trees from [21]; however, we conjecture that by using a more non-smooth soft decision function, although the regret may degrade to a level similar to NN-UCB, we can align the hypothesis spaces used in NN-UCB and ST-UCB to be almost the same. We leave the detailed analysis to future work.

# 5 Numerical experiments

In this section, we compare ST-UCB and NN-UCB to empirically demonstrate the usefulness of the tree-based model. Additionally, to evaluate the characteristics of UCB-based algorithms, we include $\epsilon$-greedy based ST-greedy and NN-greedy as comparative methods.

**Real-world dataset.** We use *Energy Efficiency* dataset [34] registered in UCI Machine Learning Repository [1]. This dataset provides the load required to maintain comfortable indoor air conditions for each of the 768 residential buildings – two types of data are provided as non-negative real values: heating load (HL) and cooling load (CL). For each building, eight types of context are included as explanatory variables. We randomly sample residential buildings without replacement to create a dataset of $\tilde{K} \leq 768$ arms, where $\tilde{K}$ is a hyperparameter. The inputs are denoted as $\boldsymbol{x} = (\tilde{\boldsymbol{x}}_{\text{building}}, \tilde{\boldsymbol{x}}) \in \mathcal{X}$, where $\tilde{\boldsymbol{x}}_{\text{building}}$ is a $\tilde{K}$-dimensional one-hot vector used to identify the arms, and $\tilde{\boldsymbol{x}}$ is a vector that aggregates the eight types of context. In most real-world data, the rewards depend not only on the observable context $\tilde{\boldsymbol{x}}$ but also on other information. To account for arm-specific characteristics that cannot be represented by $\tilde{\boldsymbol{x}}$ alone, we use $\tilde{\boldsymbol{x}}_{\text{building}}$ as part of the input.

We consider each arm of the multi-armed bandit problem as an individual residential building, and we define the reward of the arm selected in each round as $f_t = -(\text{HL}_t + \text{CL}_t)$. Additionally, we standardize the rewards across $\tilde{K}$ arms to have a mean of 0 and a standard deviation of 1.

**Synthetic dataset.** We evaluate the algorithms using synthetic data similar to that used in [41]. Here, the number of arms is set to 20, and the dimension of the input vector $\boldsymbol{x}$ for each arm is set to 50. Additionally, the input vectors are chosen uniformly at random from the unit ball. We consider the three reward functions: (i) $f^{(1)}(\boldsymbol{x}) = 10(\boldsymbol{x}^\top \boldsymbol{a})^2$, (ii) $f^{(2)}(\boldsymbol{x}) = \boldsymbol{x}^\top \boldsymbol{A}^\top \boldsymbol{A} \boldsymbol{x}$, and (iii) $f^{(3)}(\boldsymbol{x}) = \cos(3\boldsymbol{x}^\top \boldsymbol{a})$ where $\boldsymbol{a} \in \mathbb{R}^{50}$ is randomly generated from uniform distribution over unit ball, and each entry of $\boldsymbol{A} \in \mathbb{R}^{50 \times 50}$ is randomly generated from standard normal distribution. Similar to the real-world dataset, we standardize the rewards across all arms.

**Setup.** We define the cumulative regret up to round $T$ as $R_T = \sum_{t=1}^{T} f^* - f_t$ where $f^*$ represents the maximum reward among all arms. We assume that the response used for training the machine learning model is generated from $y_t = f_t + \epsilon_t$ where $\epsilon_t$ is randomly drawn from a normal distribution with mean 0 and standard deviation $\sigma_{\text{noise}} = 0.2$. Since the rewards are standardized, this setting of $\sigma_{\text{noise}}$ effectively acts as noise.

In this experiment, we will use an $\epsilon$-greedy based algorithm as an additional comparative method; In each round, an arm is selected randomly with a probability of $\epsilon$, while the arm with the highest

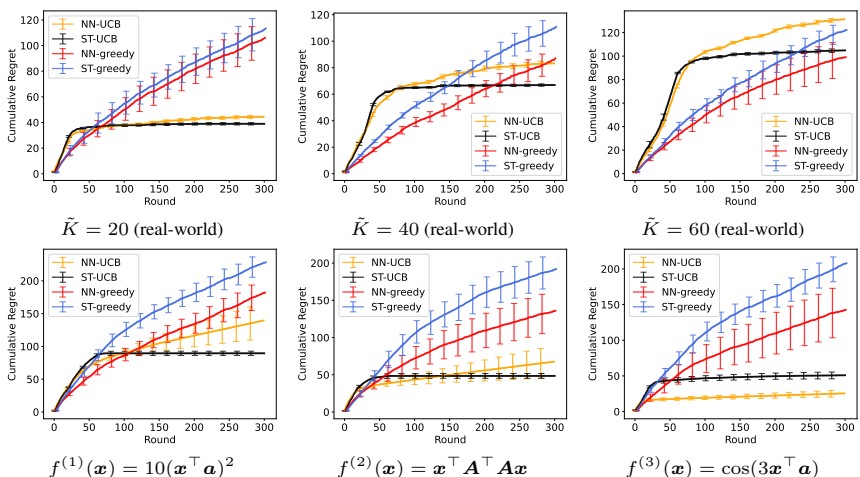

$\tilde{K} = 20$ (real-world)   $\tilde{K} = 40$ (real-world)   $\tilde{K} = 60$ (real-world)

$$f^{(1)}(\boldsymbol{x}) = 10(\boldsymbol{x}^\top \boldsymbol{a})^2 \qquad f^{(2)}(\boldsymbol{x}) = \boldsymbol{x}^\top \boldsymbol{A}^\top \boldsymbol{A} \boldsymbol{x} \qquad f^{(3)}(\boldsymbol{x}) = \cos(3\boldsymbol{x}^\top \boldsymbol{a})$$

Figure 2: The average cumulative regret with one standard error. The experiment was conducted over 10 episodes with different initial parameters for the model.

predicted value from the machine learning model is selected with a probability of $1 - \epsilon$. Here, we will perform a grid search to choose the value of $\epsilon$ from the three candidates $\epsilon \in \{0.05, 0.1, 0.2\}$. Meanwhile, in UCB-based algorithms, $\beta$ is provided as a parameter to control the degree of exploration. We use a grid search to select the value of $\beta$ from the three candidates $\beta \in \{0.01, 0.1, 1\}$.

We employ a fully connected neural network model with two intermediate layers. Including the input and output layers, the total number of layers is four. Each of the two intermediate layers contains 33 units, one of which is a bias term. As for the tree-based model, we consider an ensemble of four soft-trees, the depth of each soft-tree is three. The regularization coefficient $\lambda$ for the parameters is fixed at $10^{-4}$, regardless of the machine learning model. Supplementary details related to the implementation of the algorithms are summarized in Appendix F.1.

**Results.** The results for each algorithm are shown in Fig. 2. In real-world dataset, three different numbers of arms were considered, with $\tilde{K}$ being one of $\{20, 40, 60\}$. These experiments were conducted over 10 episodes with different initial parameters $\boldsymbol{\theta}_0$ for the model. Additional results without the grid search for $\epsilon, \beta$ are summarized in Appendix F.2.

In all settings of real-world dataset, the regret of ST-UCB was not smaller in the early rounds, but the increase in the cumulative regret became more gradual as the rounds progressed. For example, in the setting of $\tilde{K} = 60$, after round 150, there was no change in the cumulative regret of ST-UCB. However, from round 1 to 70, the regret of ST-UCB was relatively high compared to other methods. In our experiment, UCB-based policies (NN-UCB, ST-UCB) tended to actively select arms that had not been chosen before in the early rounds. As the rounds increased, exploratory behavior was suppressed, and there was a stronger tendency to select only arms with high rewards. On the other hand, in policies based on $\epsilon$-greedy (NN-greedy, ST-greedy), the exploration rate is kept at $\epsilon$ across all rounds. Therefore, the regret continues to accumulate gradually as the rounds increase, raising concerns about worsening cumulative regret over extended long rounds. In the $f^{(1)}$ and $f^{(2)}$ settings of synthetic dataset, ST-UCB outperformed the other policies, and the convergence stability of cumulative regret in $f^{(3)}$ was comparable between ST-UCB and NN-UCB.

## 6 Conclusion and future direction

In this paper, we propose a new regret-minimization algorithm based on a soft tree ensemble model. Our analysis extends the theoretical framework of existing neural bandit (NB) approaches to the soft tree ensemble model, demonstrating, under appropriate assumptions, the achievement of $\tilde{\mathcal{O}}(\sqrt{T})$ regret. To our knowledge, this is the first application of NB theory to models other than neural networks; we believe that our work marks an important first step toward developing exploration and exploitation theory using various complex models beyond neural nets.

Our future research directions are outlined below. Firstly, it is important to study the extension when employing hard decision trees. In this paper, as the scale parameter $\alpha$ approaches infinity, the soft tree regressor approaches that of a hard tree. We conjecture that our algorithm also works in this regime; however, since our regret analysis assumes a fixed $\alpha$, our proposed method is not guaranteed to maintain the no-regret property with a varying scale parameter $\alpha$. Hence, a more careful theoretical treatment is needed for this extension. Secondly, we plan to generalize the theory to encompass more common learning methods of the ensemble tree model. Specifically, learning algorithms using hard trees often utilize optimization methods in a greedy format rather than gradient descent. Therefore, developing theoretical foundations for ensemble tree learning methods that are more practically applicable is crucial.

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

# A   Information gain of TNTK

## A.1   Proof of Lemma 3.1

Firstly, we formally define the dot product kernel on the sphere.

**Definition A.1** (Dot product kernel on the sphere [31]). *Let $d \geq 2$ and $\mathbb{S}^{d-1}$ be the unit sphere of $\mathbb{R}^d$. Then, a kernel $k : \mathbb{S}^{d-1} \times \mathbb{S}^{d-1} \to \mathbb{R}$ of the following form is called a* dot product kernel on the sphere $\mathbb{S}^{d-1}$:

$$k(\boldsymbol{x}, \tilde{\boldsymbol{x}}) = \sum_{n=0}^{\infty} b_n (\boldsymbol{x}^\top \tilde{\boldsymbol{x}})^n \text{ for all } \boldsymbol{x}, \tilde{\boldsymbol{x}} \in \mathbb{S}^{d-1}, \tag{13}$$

*where $(b_n)_{n \in \mathbb{N}}$ is an absolutely summable sequence. Furthermore, if $b_n \geq 0$ for any $n \in \mathbb{N}$, $k$ is a* continuous positive semi-definite kernel *on the sphere $\mathbb{S}^{d-1}$.*

As described in Sec. 2 in [31], continuous positive semi-definite dot-product kernels are decomposed as Eq. (6) by using spherical harmonics $(Y_{n,j})$.

The following lemma shows the eigendecay of dot product kernels depending on coefficients $(b_n)_{n \in \mathbb{N}}$.

**Lemma A.1** (Proposition 2.3 in [31]). *Let $d \geq 2$ and $(Y_{n,j})_{j \in [N_{d,n}]}$ be the spherical harmonics of degree $n$. Furthermore, let $k(\boldsymbol{x}, \tilde{\boldsymbol{x}}) := \sum_{n=1}^{\infty} b_n (\boldsymbol{x}^\top \tilde{\boldsymbol{x}})^n$ be a continuous positive semi-definite dot-product kernel on $\mathbb{S}^{d-1}$. Here, if there exist $r \in (0, 1)$ and $c > 0$ such that $b_n \leq cr^n$ holds for any $n \in \mathbb{N}$, then, there exists constant $C > 0$ and $(\lambda_n)_{n \in \mathbb{N}}$ such that $\lambda_n \leq Cr^n$ and $k(\boldsymbol{x}, \tilde{\boldsymbol{x}}) = \sum_{n=0}^{\infty} \sum_{j=1}^{N_{d,n}} \lambda_n Y_{n,j}(\boldsymbol{x}) Y_{n,j}(\tilde{\boldsymbol{x}})$ hold for all $\boldsymbol{x}, \tilde{\boldsymbol{x}} \in \mathbb{S}^{d-1}$, and $n \in \mathbb{N}$.*

To prove Lemma 3.1, we consider the Maclaurin series expansion of TNTK; then, Lemma 3.1 is given from Lemma A.1.

*Proof of Lemma 3.1.* First, we respectively define functions $f_1 : [-1, 1] \to \mathbb{R}$ and $f_2 : [-1, 1] \to \mathbb{R}$ as

$$f_1(a) = \frac{1}{2\pi} \arcsin \left( \frac{\alpha^2 a}{\alpha^2 + 0.5} \right) + \frac{1}{4}, \tag{14}$$

$$f_2(a) = \frac{\alpha^2}{\pi} \frac{1}{\sqrt{(1 + 2\alpha^2)^2 - 4\alpha^4 a^2}}. \tag{15}$$

Then, since $\boldsymbol{x}, \tilde{\boldsymbol{x}} \in \mathbb{S}^{d-1}$, the following holds directly from the analytical expression of TNTK [21]:

$$k_{\mathrm{TNTK}}(\boldsymbol{x}, \tilde{\boldsymbol{x}}) = 2^{\mathcal{D}} \mathcal{D}(\boldsymbol{x}^\top \tilde{\boldsymbol{x}}) f_1(\boldsymbol{x}^\top \tilde{\boldsymbol{x}})^{\mathcal{D}-1} f_2(\boldsymbol{x}^\top \tilde{\boldsymbol{x}}) + 2^{\mathcal{D}} f_1(\boldsymbol{x}^\top \tilde{\boldsymbol{x}})^{\mathcal{D}}. \tag{16}$$

Here, since $-1 < \frac{\alpha^2 a}{\alpha^2 + 0.5} < 1$ holds for any $a \in [-1, 1]$,

$$f_1(a) = \frac{1}{2\pi} \arcsin \left( \frac{\alpha^2 a}{\alpha^2 + 0.5} \right) + \frac{1}{4} \tag{17}$$

$$= \frac{1}{2\pi} \sum_{n=0}^{\infty} \frac{(2n)!}{4^n (n!)^2 (2n+1)} \left( \frac{\alpha^2}{\alpha^2 + 0.5} \right)^{2n+1} a^{2n+1} + \frac{1}{4}, \tag{18}$$

from the Maclaurin series expansion of the inverse sine function. Furthermore, since $-1 < \left( \frac{2\alpha^2 a}{1 + 2\alpha^2} \right)^2 < 1$ holds for any $a \in [-1, 1]$,

$$f_2(a) = \frac{\alpha^2}{\pi} \frac{1}{\sqrt{(1 + 2\alpha^2)^2 - 4\alpha^4 a^2}} \tag{19}$$

$$= \frac{\alpha^2}{\pi(1 + 2\alpha^2)} \frac{1}{\sqrt{1 - \left( \frac{2\alpha^2}{1 + 2\alpha^2} \right)^2 a^2}} \tag{20}$$

$$= \frac{\alpha^2}{\pi(1 + 2\alpha^2)} \sum_{n=0}^{\infty} (-1)^n \binom{-0.5}{n} \left( \frac{\alpha^2}{\alpha^2 + 0.5} \right)^{2n} a^{2n}, \tag{21}$$

where the last line follows from the fact that $(1 + x)^c = \sum_{n=0}^{\infty} \binom{c}{n} x^n$ holds for any $c \in \mathbb{R}$ and $x \in (-1, 1)$. Here, $\binom{c}{n}$ denotes a generalized binomial coefficient, which is defined as $\binom{c}{n} = 1$ if $n = 0$; otherwise, $\binom{c}{n} = \frac{c(c-1)\cdots(c-n+1)}{n!}$. By rearranging Eq. (18) and Eq. (21), $f_1$ and $f_2$ can respectively be rewritten as $f_1(a) = \sum_{i=1}^{\infty} b_i^{(1)} a^i$ and $f_2(a) = \sum_{i=1}^{\infty} b_i^{(2)} a^i$, where the coefficients $b_i^{(1)}$ and $b_i^{(2)}$ are defined as

$$b_i^{(1)} = \begin{cases} \frac{1}{4} & \text{if } i = 0, \\ \frac{(i-1)!}{(2\pi)2^{i-1}i(((i-1)/2)!)^2} \left(\frac{\alpha^2}{\alpha^2+0.5}\right)^i & \text{if } \exists n \in \mathbb{N}, \ i = 2n+1, , \\ 0 & \text{otherwise}, \end{cases} \tag{22}$$

$$b_i^{(2)} = \begin{cases} \frac{\alpha^2}{\pi(1+2\alpha^2)} & \text{if } i = 0, \\ \frac{\alpha^2}{\pi(1+2\alpha^2)} \left(\frac{\alpha^2}{\alpha^2+0.5}\right)^i \frac{1}{(i/2)!} [0.5 \cdot 1.5 \cdots (0.5 + 0.5i - 1)] & \text{if } \exists n \in \mathbb{N}, \ i = 2n, \\ 0 & \text{otherwise}. \end{cases} \tag{23}$$

From the Stirling's inequality: $e(n/e)^n \le n! \le en(n/e)^n$, for any $i$ such that $i = 2n + 1$ holds,

$$\frac{(i-1)!}{(2\pi)2^{i-1}i(((i-1)/2)!)^2} = \frac{(2n)!}{(2\pi)2^{2n}(2n+1)(n!)^2} \tag{24}$$

$$\le \frac{2en(2n/e)^{2n}}{(2\pi)2^{2n}(2n+1)e^2(n/e)^{2n}} \tag{25}$$

$$\le \frac{2n}{(2\pi)(2n+1)e} \tag{26}$$

$$\le \frac{1}{(2\pi)e} \tag{27}$$

$$\le \frac{1}{e}. \tag{28}$$

Therefore, $0 \le b_i^{(1)} \le e^{-1} \left(\frac{\alpha^2}{\alpha^2+0.5}\right)^i$ holds for any $i \in \mathbb{N}$. Furthermore, for any $i$ such that $i = 2n$ holds,

$$\frac{\alpha^2}{\pi(1 + 2\alpha^2)} \frac{1}{(i/2)!} [0.5 \cdot 1.5 \cdots (0.5 + 0.5i - 1)] \tag{29}$$

$$= \frac{\alpha^2}{\pi(1 + 2\alpha^2)} \frac{1}{n!} [0.5 \cdot 1.5 \cdots (0.5 + n - 1)] \tag{30}$$

$$\le \frac{\alpha^2}{\pi(1 + 2\alpha^2)} \frac{1}{n!} (1 \cdot 2 \cdots n) \tag{31}$$

$$= \frac{\alpha^2}{\pi(1 + 2\alpha^2)}. \tag{32}$$

Therefore, $0 \le b_i^{(2)} \le \frac{\alpha^2}{\pi(1+2\alpha^2)} \left(\frac{\alpha^2}{\alpha^2+0.5}\right)^i$ holds for any $i \in \mathbb{N}$. Now, we rewrite Eq. (16) by using the multiple Cauchy product formula as follows:

$$k_{\text{TNTK}}(\boldsymbol{x}, \tilde{\boldsymbol{x}}) = \sum_{i=0}^{\infty} b_i (\boldsymbol{x}^\top \tilde{\boldsymbol{x}})^i, \tag{33}$$

where,

$$b_i = 2^{\mathcal{D}} \mathcal{D} \sum_{i_2=0}^{i-1} \sum_{i_3=0}^{i_2} \cdots \sum_{i_{\mathcal{D}-1}=0}^{i_{\mathcal{D}-2}} \sum_{i_{\mathcal{D}}=0}^{i_{\mathcal{D}-1}} \left( b_{i-i_2}^{(1)} b_{i_2-i_3}^{(1)} \cdots b_{i_{\mathcal{D}-1}-i_{\mathcal{D}}}^{(1)} b_{i_{\mathcal{D}}}^{(2)} \right)$$

$$+ 2^{\mathcal{D}} \sum_{i_2=0}^{i} \sum_{i_3=0}^{i_2} \cdots \sum_{i_{\mathcal{D}-1}=0}^{i_{\mathcal{D}-2}} \sum_{i_{\mathcal{D}}=0}^{i_{\mathcal{D}-1}} \left( b_{i-i_2}^{(1)} b_{i_2-i_3}^{(1)} \cdots b_{i_{\mathcal{D}-1}-i_{\mathcal{D}}}^{(1)} b_{i_{\mathcal{D}}}^{(1)} \right). \tag{34}$$

By combining Eq. (34) with the upper bounds of $b_i^{(1)}$ and $b_i^{(2)}$,

$$b_i \leq 2^{\mathcal{D}}\mathcal{D}\left(\frac{1}{e}\right)^{\mathcal{D}-1}\frac{\alpha^2}{\pi(1+2\alpha^2)}\left(\frac{\alpha^2}{\alpha^2+0.5}\right)^{i\mathcal{D}}\sum_{i_2=0}^{i-1}\sum_{i_3=0}^{i_2}\cdots\sum_{i_{\mathcal{D}-1}=0}^{i_{\mathcal{D}-2}}\sum_{i_{\mathcal{D}}=0}^{i_{\mathcal{D}-1}}1$$

$$+2^{\mathcal{D}}\left(\frac{1}{e}\right)^{\mathcal{D}}\left(\frac{\alpha^2}{\alpha^2+0.5}\right)^{i\mathcal{D}}\sum_{i_2=0}^{i}\sum_{i_3=0}^{i_2}\cdots\sum_{i_{\mathcal{D}-1}=0}^{i_{\mathcal{D}-2}}\sum_{i_{\mathcal{D}}=0}^{i_{\mathcal{D}-1}}1 \tag{35}$$

$$\leq 2^{\mathcal{D}}\mathcal{D}\left(\frac{1}{e}\right)^{\mathcal{D}-1}\frac{\alpha^2}{\pi(1+2\alpha^2)}\left(\frac{\alpha^2}{\alpha^2+0.5}\right)^{i\mathcal{D}}(i-1)^{\mathcal{D}}+2^{\mathcal{D}}\left(\frac{1}{e}\right)^{\mathcal{D}}\left(\frac{\alpha^2}{\alpha^2+0.5}\right)^{i\mathcal{D}}i^{\mathcal{D}} \tag{36}$$

$$\leq \left[2^{\mathcal{D}}\mathcal{D}\left(\frac{1}{e}\right)^{\mathcal{D}-1}\frac{\alpha^2}{\pi(1+2\alpha^2)}+2^{\mathcal{D}}\left(\frac{1}{e}\right)^{\mathcal{D}}\right]\left(\frac{\alpha^2}{\alpha^2+0.5}\right)^{i\mathcal{D}}i^{\mathcal{D}}. \tag{37}$$

Therefore, there exist constant $\tilde{C}_{\alpha,\mathcal{D}} > 0$ such that

$$b_i \leq \tilde{C}_{\alpha,\mathcal{D}}\left(\frac{\alpha^2}{\alpha^2+0.25}\right)^{i\mathcal{D}} \tag{38}$$

holds for any $i \in \mathbb{N}$. By applying Lemma A.1 with Eq. (38), we have

$$\lambda_i \leq C_{\alpha,\mathcal{D}}^{(1)}\left(\frac{\alpha^2}{\alpha^2+0.25}\right)^{i\mathcal{D}} \tag{39}$$

$$= C_{\alpha,\mathcal{D}}^{(1)}\exp\left(i\mathcal{D}\ln\left(\frac{\alpha^2}{\alpha^2+0.25}\right)\right) \tag{40}$$

$$= C_{\alpha,\mathcal{D}}^{(1)}\exp\left(-i\mathcal{D}\ln\left(1+\frac{1}{4\alpha^2}\right)\right) \tag{41}$$

for some constant $C_{\alpha,\mathcal{D}}^{(1)} > 0$. $\square$

## A.2 Proof of Theorem 3.1

Our proof strategy of Theorem 3.1 is adapted from [23, 35].

*Proof of Theorem 3.1.* Fix any deterministic sequence $x_1, \ldots, x_t \in \mathcal{X} \subset \mathbb{S}^{d-1}$. For any $M \in \mathbb{N}+$, let us define kernel functions $k_{\text{TNTK}}^{(M)}$ and $\tilde{k}_{\text{TNTK}}^{(M)}$ as

$$k_{\text{TNTK}}^{(M)}(x, \tilde{x}) = \sum_{n=0}^{M}\sum_{j=1}^{N_{d,n}}\lambda_n Y_{n,j}(x)Y_{n,j}(\tilde{x}), \tag{42}$$

$$\tilde{k}_{\text{TNTK}}^{(M)}(x, \tilde{x}) = \sum_{n=M+1}^{\infty}\sum_{j=1}^{N_{d,n}}\lambda_n Y_{n,j}(x)Y_{n,j}(\tilde{x}). \tag{43}$$

Furthermore, let $K_{\text{TNTK}}^{(M)}$ and $\tilde{K}_{\text{TNTK}}^{(M)}$ be $t \times t$-kernel matrices whose $(i, j)$-th entry are $k_{\text{TNTK}}^{(M)}(x_i, x_j)$ and $\tilde{k}_{\text{TNTK}}^{(M)}(x_i, x_j)$, respectively. As with the proof of Theorem 3 in [36], we have the following decomposition:

$$\frac{1}{2}\ln\det\left(I_t + \rho^{-1}K_{\text{TNTK}}\right)$$

$$= \frac{1}{2}\ln\det\left(I_t + \rho^{-1}K_{\text{TNTK}}^{(M)}\right) + \frac{1}{2}\ln\det\left(I_t + \rho^{-1}\left(I_t + \rho^{-1}K_{\text{TNTK}}^{(M)}\right)^{-1}\tilde{K}_{\text{TNTK}}^{(M)}\right). \tag{44}$$

By following the same argument as the proof of Theorem 2 in [35], the first term of Eq. (44) is bounded from above as follows:

$$\frac{1}{2}\ln\det\left(I_t + \rho^{-1}K_{\text{TNTK}}^{(M)}\right) \leq \frac{N_M}{2}\ln\left(1+\frac{\overline{k}t}{\rho N_M}\right). \tag{45}$$

where $N_M = \sum_{n=1}^{M} N_{d,n}$ and $\bar{k} = \max_{\boldsymbol{x} \in \mathcal{X}} k_{\mathrm{TNTK}}(\boldsymbol{x}, \boldsymbol{x})$. Furthermore, by following the same argument as the proof of Theorem 3.2 in [23], the second term of Eq. (44) is bounded from above as follows:

$$\frac{1}{2} \ln \det \left( \boldsymbol{I}_t + \rho^{-1} \left( \boldsymbol{I}_t + \rho^{-1} \boldsymbol{K}_{\mathrm{TNTK}}^{(M)} \right)^{-1} \tilde{\boldsymbol{K}}_{\mathrm{TNTK}}^{(M)} \right) \tag{46}$$

$$\leq \frac{t}{2} \ln \left( 1 + \frac{\rho^{-1} \mathrm{tr} \left( \tilde{\boldsymbol{K}}_{\mathrm{TNTK}}^{(M)} \right)}{t} \right) \tag{47}$$

$$\leq \frac{t}{2} \ln \left( 1 + \rho^{-1} \sum_{n=M+1}^{\infty} \lambda_n N_{d,n} \right) \tag{48}$$

$$\leq \frac{t}{2\rho} \sum_{n=M+1}^{\infty} \lambda_n N_{d,n}. \tag{49}$$

Then, from Lemma 3.1, there exists some constants $C > 0$ and $C_{\alpha,d} > 0$ such that

$$\sum_{n=M+1}^{\infty} \lambda_n N_{d,n} \leq \sum_{n=M+1}^{\infty} C_{\alpha,\mathcal{D}}^{(1)} C \exp\left(-C_\alpha \mathcal{D} n\right) n^{d-2} \tag{50}$$

$$\leq \sum_{n=M+1}^{\infty} C_{\alpha,\mathcal{D}}^{(1)} C C_{\alpha,d} \exp\left(-0.5 C_\alpha \mathcal{D} n\right). \tag{51}$$

where we set $C_\alpha$ as $C_\alpha = \ln(1 + 1/(4\alpha^2))$. Furthermore, Eq. (50) follows from $N_{d,n} = \Theta(n^{d-2})$ (see, e.g., [23]). Therefore,

$$\sum_{n=M+1}^{\infty} \lambda_n N_{d,n} \leq C_{\alpha,\mathcal{D}}^{(1)} C C_{\alpha,d} \int_M^{\infty} \exp\left(-0.5 C_\alpha \mathcal{D} x\right) \mathrm{d}x \tag{52}$$

$$\leq \tilde{C}_{\alpha,\mathcal{D},d} \exp\left(-\frac{C_\alpha \mathcal{D} M}{2}\right), \tag{53}$$

where we set $\tilde{C}_{\alpha,\mathcal{D},d}$ as $\tilde{C}_{\alpha,\mathcal{D},d} = C_{\alpha,\mathcal{D}}^{(1)} C C_{\alpha,d}$. Now, by noting $N_M = \mathcal{O}(M^{d-1})$ [23], there exists the constant $\tilde{C} > 0$ such that

$$\frac{1}{2} \ln \det \left( \boldsymbol{I}_t + \rho^{-1} \boldsymbol{K}_{\mathrm{TNTK}} \right) \leq \frac{N_M}{2} \ln \left( 1 + \frac{\bar{k} t}{\rho N_M} \right) + \frac{t}{2\rho} \sum_{n=M+1}^{\infty} \lambda_n N_{d,n} \tag{54}$$

$$\leq \frac{\tilde{C} M^{d-1}}{2} \ln \left( 1 + \frac{\bar{k} t}{\rho} \right) + \frac{\tilde{C}_{\alpha,\mathcal{D},d} t}{2\rho} \exp\left(-\frac{C_\alpha \mathcal{D} M}{2}\right). \tag{55}$$

By choosing $M$ as

$$M = \left\lceil 2 C_\alpha^{-1} \mathcal{D}^{-1} \ln \left( \tilde{C}_{\alpha,\mathcal{D},d} t \rho^{-1} \tilde{C}^{-1} \left[ \ln \left( 1 + \frac{\bar{k} t}{\rho} \right) \right]^{-1} \right) \right\rceil, \tag{56}$$

$M \geq 1$ for sufficiently large $t$, and we have

$$M = \left\lceil 2C_\alpha^{-1}\mathcal{D}^{-1}\ln\left(\tilde{C}_{\alpha,\mathcal{D},d}t\rho^{-1}\tilde{C}^{-1}\left[\ln\left(1+\frac{\overline{k}t}{\rho}\right)\right]^{-1}\right)\right\rceil \tag{57}$$

$$\Rightarrow M \geq 2C_\alpha^{-1}\mathcal{D}^{-1}\ln\left(\tilde{C}_{\alpha,\mathcal{D},d}t\rho^{-1}\tilde{C}^{-1}\left[\ln\left(1+\frac{\overline{k}t}{\rho}\right)\right]^{-1}\right) \tag{58}$$

$$\Leftrightarrow \exp\left(\frac{C_\alpha\mathcal{D}M}{2}\right) \geq \tilde{C}_{\alpha,\mathcal{D},d}t\rho^{-1}\tilde{C}^{-1}\left[\ln\left(1+\frac{\overline{k}t}{\rho}\right)\right]^{-1} \tag{59}$$

$$\Leftrightarrow 1 \geq \tilde{C}_{\alpha,\mathcal{D},d}t\rho^{-1}\tilde{C}^{-1}\left[\ln\left(1+\frac{\overline{k}t}{\rho}\right)\right]^{-1}\exp\left(-\frac{C_\alpha\mathcal{D}M}{2}\right) \tag{60}$$

$$\Rightarrow M^{d-1} \geq \tilde{C}_{\alpha,\mathcal{D},d}t\rho^{-1}\tilde{C}^{-1}\left[\ln\left(1+\frac{\overline{k}t}{\rho}\right)\right]^{-1}\exp\left(-\frac{C_\alpha\mathcal{D}M}{2}\right) \tag{61}$$

$$\Rightarrow \frac{\tilde{C}M^{d-1}}{2}\ln\left(1+\frac{\overline{k}t}{\rho}\right) \geq \frac{\tilde{C}_{\alpha,\mathcal{D},d}t}{2\rho}\exp\left(-\frac{C_\alpha\mathcal{D}M}{2}\right). \tag{62}$$

Therefore,

$$\frac{1}{2}\ln\det\left(\boldsymbol{I}_t + \rho^{-1}\boldsymbol{K}_{\mathrm{TNTK}}\right) \tag{63}$$

$$\leq \tilde{C}M^{d-1}\ln\left(1+\frac{\overline{k}t}{\rho}\right) \tag{64}$$

$$\leq \left\lceil 2C_\alpha^{-1}\mathcal{D}^{-1}\ln\left(\tilde{C}_{\alpha,\mathcal{D},d}t\rho^{-1}\tilde{C}^{-1}\left[\ln\left(1+\frac{\overline{k}t}{\rho}\right)\right]^{-1}\right)\right\rceil^{d-1}\tilde{C}\ln\left(1+\frac{\overline{k}t}{\rho}\right) \tag{65}$$

$$= \mathcal{O}\left(\ln^d t\right). \tag{66}$$

The above inequality holds for any choice of $\boldsymbol{x}_1,\ldots,\boldsymbol{x}_t$; hence, the proof is completed. $\qquad\square$

## B  Confidence bounds of soft trees

### B.1  Proof of Lemma 3.2

To prevent the subscript from becoming redundant hereafter, unless specifically stated otherwise, we denote the initial parameter $\boldsymbol{\theta}_0$ by $\overline{\boldsymbol{\theta}} := \boldsymbol{\theta}_0$, and the initial parameters of the $m$-th tree are denoted by $\overline{\boldsymbol{\theta}}^{(m)}$. Moreover, the initial parameter vectors corresponding to the internal nodes and leaf nodes for $\overline{\boldsymbol{\theta}}^{(m)}$ are denoted by $\overline{\boldsymbol{w}}^{(m)}$ and $\overline{\boldsymbol{\pi}}^{(m)}$, respectively. First, following [21], we decompose the finite sample approximation of the TNTK as follows:

$$\langle\nabla_{\boldsymbol{\theta}_0}h(\boldsymbol{x};\boldsymbol{\theta}_0), \nabla_{\boldsymbol{\theta}_0}h\left(\tilde{\boldsymbol{x}};\boldsymbol{\theta}_0\right)\rangle \tag{67}$$

$$= \frac{1}{M}\sum_{m=1}^{M}\left\langle\nabla_{\overline{\boldsymbol{w}}^{(m),(T)}}h^{(m)}\left(\boldsymbol{x};\boldsymbol{\theta}_0^{(m)}\right), \nabla_{\overline{\boldsymbol{w}}^{(m),(T)}}h^{(m)}\left(\tilde{\boldsymbol{x}};\boldsymbol{\theta}_0^{(m)}\right)\right\rangle \tag{68}$$

$$+ \frac{1}{M}\sum_{m=1}^{M}\left\langle\nabla_{\overline{\boldsymbol{w}}^{(m),(L)}}h^{(m)}\left(\boldsymbol{x};\boldsymbol{\theta}_0^{(m)}\right), \nabla_{\overline{\boldsymbol{w}}^{(m),(L)}}h^{(m)}\left(\tilde{\boldsymbol{x}};\boldsymbol{\theta}_0^{(m)}\right)\right\rangle$$

$$+ \frac{1}{M}\sum_{m=1}^{M}\left\langle\nabla_{\overline{\boldsymbol{w}}^{(m),(R)}}h^{(m)}\left(\boldsymbol{x};\boldsymbol{\theta}_0^{(m)}\right), \nabla_{\overline{\boldsymbol{w}}^{(m),(R)}}h^{(m)}\left(\tilde{\boldsymbol{x}};\boldsymbol{\theta}_0^{(m)}\right)\right\rangle \tag{69}$$

$$+ \frac{1}{M}\sum_{m=1}^{M}\left\langle\nabla_{\overline{\boldsymbol{\pi}}^{(m)}}h^{(m)}\left(\boldsymbol{x};\boldsymbol{\theta}_0^{(m)}\right), \nabla_{\overline{\boldsymbol{\pi}}^{(m)}}h^{(m)}\left(\tilde{\boldsymbol{x}};\boldsymbol{\theta}_0^{(m)}\right)\right\rangle. \tag{70}$$

Here, $\overline{\boldsymbol{w}}^{(m),(T)}$, $\overline{\boldsymbol{w}}^{(m),(L)}$, and $\overline{\boldsymbol{w}}^{(m),(R)}$ represent the parameters of the root (top) node, all internal nodes of the left subtree, and all internal nodes of the right subtree of the $m$-th tree at the initial values, respectively. Now, we define $k_{\mathrm{TNTK}}^{(T)}(\boldsymbol{x}, \tilde{\boldsymbol{x}})$, $k_{\mathrm{TNTK}}^{(L)}(\boldsymbol{x}, \tilde{\boldsymbol{x}})$, $k_{\mathrm{TNTK}}^{(R)}(\boldsymbol{x}, \tilde{\boldsymbol{x}})$, and $k_{\mathrm{TNTK}}^{(B)}(\boldsymbol{x}, \tilde{\boldsymbol{x}})$ as follows:

$$k_{\mathrm{TNTK}}^{(T)}(\boldsymbol{x}, \tilde{\boldsymbol{x}}) = \mathbb{E}\left[\left\langle \nabla_{\overline{\boldsymbol{w}}^{(m),(T)}} h^{(m)}\left(\boldsymbol{x}; \boldsymbol{\theta}_0^{(m)}\right), \nabla_{\overline{\boldsymbol{w}}^{(m),(T)}} h^{(m)}\left(\tilde{\boldsymbol{x}}; \boldsymbol{\theta}_0^{(m)}\right)\right\rangle\right], \tag{71}$$

$$k_{\mathrm{TNTK}}^{(L)}(\boldsymbol{x}, \tilde{\boldsymbol{x}}) = \mathbb{E}\left[\left\langle \nabla_{\overline{\boldsymbol{w}}^{(m),(L)}} h^{(m)}\left(\boldsymbol{x}; \boldsymbol{\theta}_0^{(m)}\right), \nabla_{\overline{\boldsymbol{w}}^{(m),(L)}} h^{(m)}\left(\tilde{\boldsymbol{x}}; \boldsymbol{\theta}_0^{(m)}\right)\right\rangle\right], \tag{72}$$

$$k_{\mathrm{TNTK}}^{(R)}(\boldsymbol{x}, \tilde{\boldsymbol{x}}) = \mathbb{E}\left[\left\langle \nabla_{\overline{\boldsymbol{w}}^{(m),(R)}} h^{(m)}\left(\boldsymbol{x}; \boldsymbol{\theta}_0^{(m)}\right), \nabla_{\overline{\boldsymbol{w}}^{(m),(R)}} h^{(m)}\left(\tilde{\boldsymbol{x}}; \boldsymbol{\theta}_0^{(m)}\right)\right\rangle\right], \tag{73}$$

$$k_{\mathrm{TNTK}}^{(B)}(\boldsymbol{x}, \tilde{\boldsymbol{x}}) = \mathbb{E}\left[\left\langle \nabla_{\overline{\boldsymbol{\pi}}^{(m)}} h^{(m)}\left(\boldsymbol{x}; \boldsymbol{\theta}_0^{(m)}\right), \nabla_{\overline{\boldsymbol{\pi}}^{(m)}} h^{(m)}\left(\tilde{\boldsymbol{x}}; \boldsymbol{\theta}_0^{(m)}\right)\right\rangle\right]. \tag{74}$$

Note that, since the initial parameters of each tree follow the same distribution, the definitions mentioned above do not depend on the choice of $m$. Now, assuming that the initial parameters follow a multivariate normal distribution independent across dimensions, by using the law of large numbers, Eqs. (68), (69), and (70) converge in probability, respectively, to $k_{\mathrm{TNTK}}^{(T)}(\boldsymbol{x}, \tilde{\boldsymbol{x}})$, $k_{\mathrm{TNTK}}^{(L)}(\boldsymbol{x}, \tilde{\boldsymbol{x}}) + k_{\mathrm{TNTK}}^{(R)}(\boldsymbol{x}, \tilde{\boldsymbol{x}})$, and $k_{\mathrm{TNTK}}^{(B)}(\boldsymbol{x}, \tilde{\boldsymbol{x}})$. From the continuous mapping theorem, it follows that $k_{\mathrm{TNTK}}(\boldsymbol{x}, \tilde{\boldsymbol{x}}) = k_{\mathrm{TNTK}}^{(T)}(\boldsymbol{x}, \tilde{\boldsymbol{x}}) + k_{\mathrm{TNTK}}^{(L)}(\boldsymbol{x}, \tilde{\boldsymbol{x}}) + k_{\mathrm{TNTK}}^{(R)}(\boldsymbol{x}, \tilde{\boldsymbol{x}}) + k_{\mathrm{TNTK}}^{(B)}(\boldsymbol{x}, \tilde{\boldsymbol{x}})$ can be expressed [21]. Note that the convergence to the above TNTK also holds for the initialization strategy of ST-UCB. Actually, regarding Eq. (68), we have

$$\frac{1}{M} \sum_{m=1}^{M} \left\langle \nabla_{\overline{\boldsymbol{w}}^{(m),(T)}} h^{(m)}\left(\boldsymbol{x}; \boldsymbol{\theta}_0^{(m)}\right), \nabla_{\overline{\boldsymbol{w}}^{(m),(T)}} h^{(m)}\left(\tilde{\boldsymbol{x}}; \boldsymbol{\theta}_0^{(m)}\right)\right\rangle \tag{75}$$

$$= \frac{1}{2}\Bigg[\frac{2}{M} \sum_{m=1}^{M/2} \left\langle \nabla_{\overline{\boldsymbol{w}}^{(m),(T)}} h^{(m)}\left(\boldsymbol{x}; \boldsymbol{\theta}_0^{(m)}\right), \nabla_{\overline{\boldsymbol{w}}^{(m),(T)}} h^{(m)}\left(\tilde{\boldsymbol{x}}; \boldsymbol{\theta}_0^{(m)}\right)\right\rangle$$

$$+ \frac{2}{M} \sum_{m=1}^{M/2} \left\langle \nabla_{\overline{\boldsymbol{w}}^{(M/2+m),(T)}} h^{(M/2+m)}\left(\boldsymbol{x}; \boldsymbol{\theta}_0^{(M/2+m)}\right), \nabla_{\overline{\boldsymbol{w}}^{(M/2+m),(T)}} h^{(m)}\left(\tilde{\boldsymbol{x}}; \boldsymbol{\theta}_0^{(M/2+m)}\right)\right\rangle\Bigg]. \tag{76}$$

The first and second terms correspond to the inner products of gradients when initializing $M/2$ soft trees with the standard normal distribution and converge in probability to $k_{\mathrm{TNTK}}^{(T)}(\boldsymbol{x}, \tilde{\boldsymbol{x}})$. Therefore, by the continuous mapping theorem, Eq. (75) converges in probability to $k_{\mathrm{TNTK}}^{(T)}(\boldsymbol{x}, \tilde{\boldsymbol{x}})$. Similar arguments apply to $k_{\mathrm{TNTK}}^{(L)}(\boldsymbol{x}, \tilde{\boldsymbol{x}}) + k_{\mathrm{TNTK}}^{(R)}(\boldsymbol{x}, \tilde{\boldsymbol{x}})$ and $k_{\mathrm{TNTK}}^{(B)}(\boldsymbol{x}, \tilde{\boldsymbol{x}})$, indicating that in the initialization strategy of ST-UCB, $\langle \nabla_{\boldsymbol{\theta}_0} h(\boldsymbol{x}; \boldsymbol{\theta}_0), \nabla_{\boldsymbol{\theta}_0} h(\tilde{\boldsymbol{x}}; \boldsymbol{\theta}_0)\rangle$ also converges in probability to $k_{\mathrm{TNTK}}(\boldsymbol{x}, \tilde{\boldsymbol{x}})$. The following three lemmas each evaluate the concentration to $k_{\mathrm{TNTK}}^{(T)}(\boldsymbol{x}, \tilde{\boldsymbol{x}})$, $k_{\mathrm{TNTK}}^{(L)}(\boldsymbol{x}, \tilde{\boldsymbol{x}}) + k_{\mathrm{TNTK}}^{(R)}(\boldsymbol{x}, \tilde{\boldsymbol{x}})$, and $k_{\mathrm{TNTK}}^{(B)}(\boldsymbol{x}, \tilde{\boldsymbol{x}})$ for Eqs. (68), (69), and (70), respectively.

**Lemma B.1.** *For any $\boldsymbol{x}, \tilde{\boldsymbol{x}} \in \mathbb{S}^{d-1}$ and $\epsilon \geq 0$, we have*

$$\mathbb{P}\left(\left|k_{\mathrm{TNTK}}^{(T)}(\boldsymbol{x}, \tilde{\boldsymbol{x}}) - \frac{1}{M} \sum_{m=1}^{M} \left\langle \nabla_{\overline{\boldsymbol{w}}^{(m),(T)}} h^{(m)}\left(\boldsymbol{x}; \boldsymbol{\theta}_0^{(m)}\right), \nabla_{\overline{\boldsymbol{w}}^{(m),(T)}} h^{(m)}\left(\tilde{\boldsymbol{x}}; \boldsymbol{\theta}_0^{(m)}\right)\right\rangle\right| \leq \epsilon\right)$$

$$\geq 1 - 4\exp\left(-c\min\left\{\frac{\epsilon^2}{K^2}, \frac{\epsilon}{K}\right\} M\right), \tag{77}$$

*where $K = 4\alpha^2 C\mathcal{L}^2$. Furthermore, $C, c > 0$ are absolute constants.*

**Lemma B.2.** *For any $x, \tilde{x} \in \mathbb{S}^{d-1}$, $\epsilon \geq 0$, and $\mathcal{D} \geq 2$, we have*

$$
\mathbb{P}\left(\left|k_{\mathrm{TNTK}}^{(L)}(\boldsymbol{x}, \tilde{\boldsymbol{x}}) - \frac{1}{M}\sum_{m=1}^{M}\left\langle \nabla_{\overline{\boldsymbol{w}}^{(m),(L)}} h^{(m)}\left(\boldsymbol{x}; \boldsymbol{\theta}_0^{(m)}\right), \nabla_{\overline{\boldsymbol{w}}^{(m),(L)}} h^{(m)}\left(\tilde{\boldsymbol{x}}; \boldsymbol{\theta}_0^{(m)}\right)\right\rangle\right| \leq \epsilon\right)
$$
$$
\geq 1 - 4\exp\left(-c\min\left\{\frac{\epsilon^2}{K^2}, \frac{\epsilon}{K}\right\}M\right).
$$

(78)

*Furthermore, we have*

$$
\mathbb{P}\left(\left|k_{\mathrm{TNTK}}^{(R)}(\boldsymbol{x}, \tilde{\boldsymbol{x}}) - \frac{1}{M}\sum_{m=1}^{M}\left\langle \nabla_{\overline{\boldsymbol{w}}^{(m),(R)}} h^{(m)}\left(\boldsymbol{x}; \boldsymbol{\theta}_0^{(m)}\right), \nabla_{\overline{\boldsymbol{w}}^{(m),(R)}} h^{(m)}\left(\tilde{\boldsymbol{x}}; \boldsymbol{\theta}_0^{(m)}\right)\right\rangle\right| \leq \epsilon\right)
$$
$$
\geq 1 - 4\exp\left(-c\min\left\{\frac{\epsilon^2}{K^2}, \frac{\epsilon}{K}\right\}M\right).
$$

(79)

**Lemma B.3.** *For any $x, \tilde{x} \in \mathbb{S}^{d-1}$ and $\epsilon \geq 0$, we have*

$$
\mathbb{P}\left(\left|k_{\mathrm{TNTK}}^{(B)}(\boldsymbol{x}, \tilde{\boldsymbol{x}}) - \frac{1}{M}\sum_{m=1}^{M}\left\langle \nabla_{\overline{\boldsymbol{\pi}}^{(m)}} h^{(m)}\left(\boldsymbol{x}; \boldsymbol{\theta}_0^{(m)}\right), \nabla_{\overline{\boldsymbol{\pi}}^{(m)}} h^{(m)}\left(\tilde{\boldsymbol{x}}; \boldsymbol{\theta}_0^{(m)}\right)\right\rangle\right| \leq \epsilon\right)
$$
$$
\geq 1 - 4\exp\left(-\frac{\tilde{c}\epsilon^2 M}{\mathcal{L}^2}\right).
$$

(80)

*Here, $\tilde{c} > 0$ is an absolute constant.*

In proving the above lemmas, following [21], we denote a single soft tree of depth $\tilde{\mathcal{D}}$ determined by the internal node parameters $\boldsymbol{w} \in \mathbb{R}^{d(2^{\tilde{\mathcal{D}}}-1)}$ and leaf node parameters $\boldsymbol{\pi} \in \mathbb{R}^{2^{\tilde{\mathcal{D}}}}$ as $h_{\tilde{\mathcal{D}}}(\cdot, \boldsymbol{w}, \boldsymbol{\pi})$.

*Proof of Lemma B.1.* Fix any $\tilde{\mathcal{D}} \leq \mathcal{D}$, $\boldsymbol{w} \in \mathbb{R}^{d(2^{\tilde{\mathcal{D}}}-1)}$, and $\boldsymbol{\pi} \in \mathbb{R}^{2^{\tilde{\mathcal{D}}}}$. From the definition of the soft tree, the following recursive formula holds [21]:

$$
\begin{aligned}
&h_{\tilde{\mathcal{D}}}(\boldsymbol{x}, \boldsymbol{w}, \boldsymbol{\pi}) \\
&= \sigma\left(\boldsymbol{w}^{(T)\top}\boldsymbol{x}\right) h_{\tilde{\mathcal{D}}-1}\left(\boldsymbol{x}, \boldsymbol{w}^{(L)}, \boldsymbol{\pi}^{(L)}\right) + \left[1 - \sigma\left(\boldsymbol{w}^{(T)\top}\boldsymbol{x}\right)\right] h_{\tilde{\mathcal{D}}-1}\left(\boldsymbol{x}, \boldsymbol{w}^{(R)}, \boldsymbol{\pi}^{(R)}\right).
\end{aligned}
$$

(81)

Note that $h^{(m)}(\boldsymbol{x}; \boldsymbol{\theta}^{(m)}) = h_{\mathcal{D}}\left(\boldsymbol{x}, \boldsymbol{w}^{(m)}, \boldsymbol{\pi}^{(m)}\right)$. Here, $\boldsymbol{\pi}^{(L)}, \boldsymbol{\pi}^{(R)}$ represent the parameters of the leaves belonging to the left and right subtrees, respectively. From Eq. (81), we have

$$
\nabla_{\boldsymbol{w}^{(T)}} h_{\tilde{\mathcal{D}}}(\boldsymbol{x}, \boldsymbol{w}, \boldsymbol{\pi}) = \boldsymbol{x}\dot{\sigma}\left(\boldsymbol{w}^{(T)\top}\boldsymbol{x}\right)\left[h_{\tilde{\mathcal{D}}-1}\left(\boldsymbol{x}, \boldsymbol{w}^{(L)}, \boldsymbol{\pi}^{(L)}\right) - h_{\tilde{\mathcal{D}}-1}\left(\boldsymbol{x}, \boldsymbol{w}^{(R)}, \boldsymbol{\pi}^{(R)}\right)\right], \quad (82)
$$

where $\dot{\sigma}(b) := \alpha\exp(-\alpha^2 b^2)/\sqrt{\pi}$ is the derivative of $\sigma(\cdot)$. Therefore,

$$
\begin{aligned}
&\langle \nabla_{\boldsymbol{w}^{(T)}} h_{\tilde{\mathcal{D}}}(\boldsymbol{x}, \boldsymbol{w}, \boldsymbol{\pi}), \nabla_{\boldsymbol{w}^{(T)}} h_{\tilde{\mathcal{D}}}(\tilde{\boldsymbol{x}}, \boldsymbol{w}, \boldsymbol{\pi})\rangle \\
&= \boldsymbol{x}^\top\tilde{\boldsymbol{x}}\dot{\sigma}\left(\boldsymbol{w}^{(T)\top}\boldsymbol{x}\right)\dot{\sigma}\left(\boldsymbol{w}^{(T)\top}\tilde{\boldsymbol{x}}\right)\Big[h_{\tilde{\mathcal{D}}-1}\left(\boldsymbol{x}, \boldsymbol{w}^{(L)}, \boldsymbol{\pi}^{(L)}\right) h_{\tilde{\mathcal{D}}-1}\left(\tilde{\boldsymbol{x}}, \boldsymbol{w}^{(L)}, \boldsymbol{\pi}^{(L)}\right) \\
&\quad - h_{\tilde{\mathcal{D}}-1}\left(\boldsymbol{x}, \boldsymbol{w}^{(L)}, \boldsymbol{\pi}^{(L)}\right) h_{\tilde{\mathcal{D}}-1}\left(\tilde{\boldsymbol{x}}, \boldsymbol{w}^{(R)}, \boldsymbol{\pi}^{(R)}\right) \\
&\quad - h_{\tilde{\mathcal{D}}-1}\left(\boldsymbol{x}, \boldsymbol{w}^{(R)}, \boldsymbol{\pi}^{(R)}\right) h_{\tilde{\mathcal{D}}-1}\left(\tilde{\boldsymbol{x}}, \boldsymbol{w}^{(L)}, \boldsymbol{\pi}^{(L)}\right) \\
&\quad + h_{\tilde{\mathcal{D}}-1}\left(\boldsymbol{x}, \boldsymbol{w}^{(R)}, \boldsymbol{\pi}^{(R)}\right) h_{\tilde{\mathcal{D}}-1}\left(\tilde{\boldsymbol{x}}, \boldsymbol{w}^{(R)}, \boldsymbol{\pi}^{(R)}\right)\Big].
\end{aligned}
$$

(83)

Here, let us define $p_{\tilde{\mathcal{D}},l}(\boldsymbol{x}, \boldsymbol{w}) := \prod_{n=1}^{2^{\tilde{\mathcal{D}}}-1}\sigma\left(\boldsymbol{w}_n^\top\boldsymbol{x}\right)^{\mathbb{1}_{l\swarrow n}}\left[1 - \sigma\left(\boldsymbol{w}_n^\top\boldsymbol{x}\right)\right]^{\mathbb{1}_{l\searrow n}}$ as the weight probability function of leaf $l$ in a soft tree of depth $\tilde{\mathcal{D}}$; then, we have

$$
h_{\tilde{\mathcal{D}}-1}\left(\boldsymbol{x}, \boldsymbol{w}^{(L)}, \boldsymbol{\pi}^{(L)}\right) = \sum_{l=1}^{2^{\tilde{\mathcal{D}}-1}}\pi_l^{(L)} p_{\tilde{\mathcal{D}}-1,l}\left(\boldsymbol{x}, \boldsymbol{w}^{(L)}\right).
$$

(84)

Since the sub-Gaussian norm of the normal distribution is bounded from above by a constant multiple of its standard deviation (see, e.g., Example 2.5.6 in [39]), for any $m \in [M]$, we have

$$\left\| h_{\mathcal{D}-1}\left(\boldsymbol{x}, \overline{\boldsymbol{w}}^{(m),(L)}, \overline{\boldsymbol{\pi}}^{(m),(L)}\right) \right\|_{\psi_2} = \left\| \sum_{l=1}^{2^{\mathcal{D}-1}} \overline{\pi}_l^{(m),(L)} p_{\mathcal{D}-1,l}\left(\boldsymbol{x}, \overline{\boldsymbol{w}}^{(m),(L)}\right) \right\|_{\psi_2} \tag{85}$$

$$\leq \left\| \sum_{l=1}^{2^{\mathcal{D}-1}} \overline{\pi}_l^{(m),(L)} \right\|_{\psi_2} \tag{86}$$

$$\leq C\mathcal{L}, \tag{87}$$

where the first inequality follows from $\left| p_{\mathcal{D}-1,l}\left(\boldsymbol{x}, \overline{\boldsymbol{w}}^{(m),(L)}\right) \right| \leq 1$. Similarly,

$$\left\| h_{\mathcal{D}-1}\left(\boldsymbol{x}, \overline{\boldsymbol{w}}^{(m),(R)}, \overline{\boldsymbol{\pi}}^{(m),(R)}\right) \right\|_{\psi_2} \leq C\mathcal{L}. \tag{88}$$

Due to $\|\dot{\sigma}(\cdot)\|_\infty \leq \alpha/\sqrt{\pi}$, $|\boldsymbol{x}^\top \tilde{\boldsymbol{x}}| \leq 1$, and Lemma E.4, we obtain

$$\left\| \left\langle \nabla_{\overline{\boldsymbol{w}}^{(T)}} h^{(m)}\left(\boldsymbol{x}; \overline{\boldsymbol{\theta}}^{(m)}\right), \nabla_{\overline{\boldsymbol{w}}^{(T)}} h^{(m)}\left(\tilde{\boldsymbol{x}}; \overline{\boldsymbol{\theta}}^{(m)}\right) \right\rangle \right\|_{\psi_1} \leq \frac{4C^2\mathcal{L}^2\alpha^2}{\pi}. \tag{89}$$

From the centering lemma (Lemma E.3), there exists an absolute constant $\tilde{C} > 0$ such that

$$\left\| k_{\mathrm{TNTK}}^{(T)}(\boldsymbol{x}, \tilde{\boldsymbol{x}}) - \left\langle \nabla_{\overline{\boldsymbol{w}}^{(T)}} h^{(m)}\left(\boldsymbol{x}; \overline{\boldsymbol{\theta}}^{(m)}\right), \nabla_{\overline{\boldsymbol{w}}^{(T)}} h^{(m)}\left(\tilde{\boldsymbol{x}}; \overline{\boldsymbol{\theta}}^{(m)}\right) \right\rangle \right\|_{\psi_1} \leq \frac{4\tilde{C}C^2\mathcal{L}^2\alpha^2}{\pi} \tag{90}$$

$$\leq 4\tilde{C}C^2\mathcal{L}^2\alpha^2. \tag{91}$$

Therefore, taking $\tilde{C}C^2$ as a new absolute constant $C$ and using the independence of parameters for each $m \in [M/2]$, the application of Bernstein's inequality (Lemma E.2) yields

$$\mathbb{P}\left( \left| k_{\mathrm{TNTK}}^{(T)}(\boldsymbol{x}, \tilde{\boldsymbol{x}}) - \frac{2}{M} \sum_{m=1}^{M/2} \left\langle \nabla_{\overline{\boldsymbol{w}}^{(m),(T)}} h^{(m)}\left(\boldsymbol{x}; \boldsymbol{\theta}_0^{(m)}\right), \nabla_{\overline{\boldsymbol{w}}^{(m),(T)}} h^{(m)}\left(\tilde{\boldsymbol{x}}; \boldsymbol{\theta}_0^{(m)}\right) \right\rangle \right| \geq \epsilon \right)$$
$$\leq 2\exp\left( -c\min\left\{ \frac{\epsilon^2}{2K^2}, \frac{\epsilon}{2K} \right\} M \right). \tag{92}$$

Note that the similar inequality also holds for $m \in [M] \setminus [M/2]$:

$$\mathbb{P}\left( \left| k_{\mathrm{TNTK}}^{(T)}(\boldsymbol{x}, \tilde{\boldsymbol{x}}) - \frac{2}{M} \sum_{m=M/2+1}^{M} \left\langle \nabla_{\overline{\boldsymbol{w}}^{(m),(T)}} h^{(m)}\left(\boldsymbol{x}; \boldsymbol{\theta}_0^{(m)}\right), \nabla_{\overline{\boldsymbol{w}}^{(m),(T)}} h^{(m)}\left(\tilde{\boldsymbol{x}}; \boldsymbol{\theta}_0^{(m)}\right) \right\rangle \right| \geq \epsilon \right)$$
$$\leq 2\exp\left( -c\min\left\{ \frac{\epsilon^2}{2K^2}, \frac{\epsilon}{2K} \right\} M \right). \tag{93}$$

By taking union bound in Eqs. (92) and (93) and taking $c/2$ as an new absolute constant $c$, we obtain the desired result.

$$\square$$

*Proof of Lemma B.2.* We only show Eq. (78) for simplicity. Fix any $\boldsymbol{w} \in \mathbb{R}^{d\mathcal{N}}$ and $\boldsymbol{\pi} \in \mathbb{R}^{\mathcal{L}}$ corresponding to the parameters of a soft tree of depth $\mathcal{D}$. Furthermore, let $\boldsymbol{w}_{i:}$ and $\boldsymbol{\pi}_{i:}$ ($1 \leq i \leq \mathcal{N}$) represent the internal node parameter vectors and the leaf node parameter vectors, respectively, for the subtree rooted at the $i$-th internal node (note that the parameter indices are assigned in breadth-first order, hence by definition, $\boldsymbol{w}_{2:} = \boldsymbol{w}^{(L)}$, $\boldsymbol{w}_{3:} = \boldsymbol{w}^{(R)}$). From Eq. (81), we have:

$$\nabla_{\boldsymbol{w}^{(L)}} h_{\mathcal{D}}(\boldsymbol{x}, \boldsymbol{w}, \boldsymbol{\pi}) = \sigma\left(\boldsymbol{w}^{(T)\top}\boldsymbol{x}\right) \nabla_{\boldsymbol{w}^{(L)}} h_{\mathcal{D}-1}\left(\boldsymbol{x}, \boldsymbol{w}^{(L)}, \boldsymbol{\pi}^{(L)}\right). \tag{94}$$

Given that $\|\sigma(\cdot)\|_\infty \leq 1$, for any $m \in [M]$, we have:

$$\left\| \left\langle \nabla_{\overline{\boldsymbol{w}}^{(m),(L)}} h^{(m)}\left(\boldsymbol{x}; \boldsymbol{\theta}_0^{(m)}\right), \nabla_{\boldsymbol{w}^{(m),(L)}} h^{(m)}\left(\tilde{\boldsymbol{x}}; \boldsymbol{\theta}_0^{(m)}\right)\right\rangle \right\|_{\psi_1}$$
$$\leq \left\| \left\langle \nabla_{\overline{\boldsymbol{w}}^{(m),(L)}} h_{\mathcal{D}-1}\left(\boldsymbol{x}, \overline{\boldsymbol{w}}^{(m),(L)}, \overline{\boldsymbol{\pi}}^{(m),(L)}\right), \nabla_{\overline{\boldsymbol{w}}^{(m),(L)}} h_{\mathcal{D}-1}\left(\tilde{\boldsymbol{x}}, \overline{\boldsymbol{w}}^{(m),(L)}, \overline{\boldsymbol{\pi}}^{(m),(L)}\right)\right\rangle \right\|_{\psi_1}. \tag{95}$$

Now, decomposing the gradient of the subtree rooted at the left child of the root node, we have:

$$\left\langle \nabla_{\boldsymbol{w}^{(L)}} h_{\mathcal{D}-1}\left(\boldsymbol{x}, \boldsymbol{w}^{(L)}, \boldsymbol{\pi}^{(L)}\right), \nabla_{\boldsymbol{w}^{(L)}} h_{\mathcal{D}-1}\left(\tilde{\boldsymbol{x}}, \boldsymbol{w}^{(L)}, \boldsymbol{\pi}^{(L)}\right)\right\rangle \tag{96}$$
$$= \left\langle \nabla_{\boldsymbol{w}_{2:}} h_{\mathcal{D}-1}\left(\boldsymbol{x}, \boldsymbol{w}_{2:}, \boldsymbol{\pi}_{2:}\right), \nabla_{\boldsymbol{w}_{2:}} h_{\mathcal{D}-1}\left(\tilde{\boldsymbol{x}}, \boldsymbol{w}_{2:}, \boldsymbol{\pi}_{2:}\right)\right\rangle \tag{97}$$
$$= \left\langle \nabla_{\boldsymbol{w}_{2:}^{(T)}} h_{\mathcal{D}-1}\left(\boldsymbol{x}, \boldsymbol{w}_{2:}, \boldsymbol{\pi}_{2:}\right), \nabla_{\boldsymbol{w}_{2:}^{(T)}} h_{\mathcal{D}-1}\left(\tilde{\boldsymbol{x}}, \boldsymbol{w}_{2:}, \boldsymbol{\pi}_{2:}\right)\right\rangle$$
$$+ \left\langle \nabla_{\boldsymbol{w}_{2:}^{(L)}} h_{\mathcal{D}-1}\left(\boldsymbol{x}, \boldsymbol{w}_{2:}, \boldsymbol{\pi}_{2:}\right), \nabla_{\boldsymbol{w}_{2:}^{(L)}} h_{\mathcal{D}-1}\left(\tilde{\boldsymbol{x}}, \boldsymbol{w}_{2:}, \boldsymbol{\pi}_{2:}\right)\right\rangle \tag{98}$$
$$+ \left\langle \nabla_{\boldsymbol{w}_{2:}^{(R)}} h_{\mathcal{D}-1}\left(\boldsymbol{x}, \boldsymbol{w}_{2:}, \boldsymbol{\pi}_{2:}\right), \nabla_{\boldsymbol{w}_{2:}^{(R)}} h_{\mathcal{D}-1}\left(\tilde{\boldsymbol{x}}, \boldsymbol{w}_{2:}, \boldsymbol{\pi}_{2:}\right)\right\rangle.$$

Considering that $\boldsymbol{w}_{2:}$ are parameters for a soft tree with $\mathcal{L}/2$ leaves, similar to the proof of Lemma B.1, there exists an absolute constant $C$ such that for any $m \in [M]$:

$$\left\| \left\langle \nabla_{\overline{\boldsymbol{w}}_{2:}^{(m),(T)}} h_{\mathcal{D}-1}\left(\boldsymbol{x}, \overline{\boldsymbol{w}}_{2:}^{(m)}, \overline{\boldsymbol{\pi}}_{2:}^{(m)}\right), \nabla_{\overline{\boldsymbol{w}}_{2:}^{(m),(T)}} h_{\mathcal{D}-1}\left(\tilde{\boldsymbol{x}}, \overline{\boldsymbol{w}}_{2:}^{(m)}, \overline{\boldsymbol{\pi}}_{2:}^{(m)}\right)\right\rangle \right\|_{\psi_1} \leq 4\alpha^2 C \pi^{-1}(\mathcal{L}/2)^2. \tag{99}$$

Similarly to Eq. (95), we have:

$$\left\| \left\langle \nabla_{\overline{\boldsymbol{w}}_{2:}^{(m),(L)}} h_{\mathcal{D}-1}\left(\boldsymbol{x}, \overline{\boldsymbol{w}}_{2:}^{(m)}, \overline{\boldsymbol{\pi}}_{2:}^{(m)}\right), \nabla_{\overline{\boldsymbol{w}}_{2:}^{(m),(L)}} h_{\mathcal{D}-1}\left(\tilde{\boldsymbol{x}}, \overline{\boldsymbol{w}}_{2:}^{(m)}, \overline{\boldsymbol{\pi}}_{2:}^{(m)}\right)\right\rangle \right\|_{\psi_1}$$
$$\leq \left\| \left\langle \nabla_{\overline{\boldsymbol{w}}_{2:}^{(m),(L)}} h_{\mathcal{D}-2}\left(\boldsymbol{x}, \overline{\boldsymbol{w}}_{2:}^{(m),(L)}, \overline{\boldsymbol{\pi}}_{2:}^{(m),(L)}\right), \nabla_{\overline{\boldsymbol{w}}_{2:}^{(m),(L)}} h_{\mathcal{D}-2}\left(\tilde{\boldsymbol{x}}, \overline{\boldsymbol{w}}_{2:}^{(m),(L)}, \overline{\boldsymbol{\pi}}_{2:}^{(m),(L)}\right)\right\rangle \right\|_{\psi_1}. \tag{100}$$

Similarly, for the right subtree:

$$\left\| \left\langle \nabla_{\overline{\boldsymbol{w}}_{2:}^{(m),(R)}} h_{\mathcal{D}-1}\left(\boldsymbol{x}, \overline{\boldsymbol{w}}_{2:}^{(m)}, \overline{\boldsymbol{\pi}}_{2:}^{(m)}\right), \nabla_{\overline{\boldsymbol{w}}_{2:}^{(m),(R)}} h_{\mathcal{D}-1}\left(\tilde{\boldsymbol{x}}, \overline{\boldsymbol{w}}_{2:}^{(m)}, \overline{\boldsymbol{\pi}}_{2:}^{(m)}\right)\right\rangle \right\|_{\psi_1}$$
$$\leq \left\| \left\langle \nabla_{\overline{\boldsymbol{w}}_{2:}^{(m),(R)}} h_{\mathcal{D}-2}\left(\boldsymbol{x}, \overline{\boldsymbol{w}}_{2:}^{(m),(R)}, \overline{\boldsymbol{\pi}}_{2:}^{(m),(R)}\right), \nabla_{\overline{\boldsymbol{w}}_{2:}^{(m),(R)}} h_{\mathcal{D}-2}\left(\tilde{\boldsymbol{x}}, \overline{\boldsymbol{w}}_{2:}^{(m),(R)}, \overline{\boldsymbol{\pi}}_{2:}^{(m),(R)}\right)\right\rangle \right\|_{\psi_1}. \tag{101}$$

Therefore,

$$\left\| \left\langle \nabla_{\overline{\boldsymbol{w}}^{(m),(L)}} h_{\mathcal{D}-1}\left(\boldsymbol{x}, \overline{\boldsymbol{w}}^{(m),(L)}, \overline{\boldsymbol{\pi}}^{(m),(L)}\right), \nabla_{\overline{\boldsymbol{w}}^{(m),(L)}} h_{\mathcal{D}-1}\left(\tilde{\boldsymbol{x}}, \overline{\boldsymbol{w}}^{(m),(L)}, \overline{\boldsymbol{\pi}}^{(m),(L)}\right)\right\rangle \right\|_{\psi_1}$$
$$\leq 4\alpha^2 C \pi^{-1}(\mathcal{L}/2)^2$$
$$+ \left\| \left\langle \nabla_{\overline{\boldsymbol{w}}_{2:}^{(m),(L)}} h_{\mathcal{D}-2}\left(\boldsymbol{x}, \overline{\boldsymbol{w}}_{2:}^{(m),(L)}, \overline{\boldsymbol{\pi}}_{2:}^{(m),(L)}\right), \nabla_{\overline{\boldsymbol{w}}_{2:}^{(m),(L)}} h_{\mathcal{D}-2}\left(\tilde{\boldsymbol{x}}, \overline{\boldsymbol{w}}_{2:}^{(m),(L)}, \overline{\boldsymbol{\pi}}_{2:}^{(m),(L)}\right)\right\rangle \right\|_{\psi_1}$$
$$+ \left\| \left\langle \nabla_{\overline{\boldsymbol{w}}_{2:}^{(m),(R)}} h_{\mathcal{D}-2}\left(\boldsymbol{x}, \overline{\boldsymbol{w}}_{2:}^{(m),(R)}, \overline{\boldsymbol{\pi}}_{2:}^{(m),(R)}\right), \nabla_{\overline{\boldsymbol{w}}_{2:}^{(m),(R)}} h_{\mathcal{D}-2}\left(\tilde{\boldsymbol{x}}, \overline{\boldsymbol{w}}_{2:}^{(m),(R)}, \overline{\boldsymbol{\pi}}_{2:}^{(m),(R)}\right)\right\rangle \right\|_{\psi_1}. \tag{102}$$

By repeating the above described argument, we can further decompose the second and third term of Eq. (102) as follows:

$$\left\| \left\langle \nabla_{\overline{\boldsymbol{w}}^{(m),(L)}} h_{\mathcal{D}-1}\left(\boldsymbol{x}, \overline{\boldsymbol{w}}^{(m),(L)}, \overline{\boldsymbol{\pi}}^{(m),(L)}\right), \nabla_{\overline{\boldsymbol{w}}^{(m),(L)}} h_{\mathcal{D}-1}\left(\tilde{\boldsymbol{x}}, \overline{\boldsymbol{w}}^{(m),(L)}, \overline{\boldsymbol{\pi}}^{(m),(L)}\right) \right\rangle \right\|_{\psi_1} \tag{103}$$

$$\leq 4\alpha^2 C\pi^{-1}(\mathcal{L}/2)^2 \tag{104}$$

$$+ 2 \times 4\alpha^2 C\pi^{-1}(\mathcal{L}/4)^2 \tag{105}$$

$$+ \left\| \left\langle \nabla_{\overline{\boldsymbol{w}}_{8:}^{(m)}} h_{\mathcal{D}-3}\left(\boldsymbol{x}, \overline{\boldsymbol{w}}_{8:}^{(m)}, \overline{\boldsymbol{\pi}}_{8:}^{(m)}\right), \nabla_{\overline{\boldsymbol{w}}_{8:}^{(m)}} h_{\mathcal{D}-3}\left(\tilde{\boldsymbol{x}}, \overline{\boldsymbol{w}}_{8:}^{(m)}, \overline{\boldsymbol{\pi}}_{8:}^{(m)}\right) \right\rangle \right\|_{\psi_1} \tag{106}$$

$$+ \left\| \left\langle \nabla_{\overline{\boldsymbol{w}}_{9:}^{(m)}} h_{\mathcal{D}-3}\left(\boldsymbol{x}, \overline{\boldsymbol{w}}_{9:}^{(m)}, \overline{\boldsymbol{\pi}}_{9:}^{(m)}\right), \nabla_{\overline{\boldsymbol{w}}_{9:}^{(m)}} h_{\mathcal{D}-3}\left(\tilde{\boldsymbol{x}}, \overline{\boldsymbol{w}}_{9:}^{(m)}, \overline{\boldsymbol{\pi}}_{9:}^{(m)}\right) \right\rangle \right\|_{\psi_1} \tag{107}$$

$$+ \left\| \left\langle \nabla_{\overline{\boldsymbol{w}}_{10:}^{(m)}} h_{\mathcal{D}-3}\left(\boldsymbol{x}, \overline{\boldsymbol{w}}_{10:}^{(m)}, \overline{\boldsymbol{\pi}}_{10:}^{(m)}\right), \nabla_{\overline{\boldsymbol{w}}_{10:}^{(m)}} h_{\mathcal{D}-3}\left(\tilde{\boldsymbol{x}}, \overline{\boldsymbol{w}}_{10:}^{(m)}, \overline{\boldsymbol{\pi}}_{10:}^{(m)}\right) \right\rangle \right\|_{\psi_1} \tag{108}$$

$$+ \left\| \left\langle \nabla_{\overline{\boldsymbol{w}}_{11:}^{(m)}} h_{\mathcal{D}-3}\left(\boldsymbol{x}, \overline{\boldsymbol{w}}_{11:}^{(m)}, \overline{\boldsymbol{\pi}}_{11:}^{(m)}\right), \nabla_{\overline{\boldsymbol{w}}_{11:}^{(m)}} h_{\mathcal{D}-3}\left(\tilde{\boldsymbol{x}}, \overline{\boldsymbol{w}}_{11:}^{(m)}, \overline{\boldsymbol{\pi}}_{11:}^{(m)}\right) \right\rangle \right\|_{\psi_1}. \tag{109}$$

By recursively applying the above discussion until reaching the leaves of the tree, we find:

$$\left\| \left\langle \nabla_{\overline{\boldsymbol{w}}^{(m),(L)}} h_{\mathcal{D}-1}\left(\boldsymbol{x}, \overline{\boldsymbol{w}}^{(m),(L)}, \overline{\boldsymbol{\pi}}^{(m),(L)}\right), \nabla_{\overline{\boldsymbol{w}}^{(m),(L)}} h_{\mathcal{D}-1}\left(\tilde{\boldsymbol{x}}, \overline{\boldsymbol{w}}^{(m),(L)}, \overline{\boldsymbol{\pi}}^{(m),(L)}\right) \right\rangle \right\|_{\psi_1}$$

$$\leq 4\alpha^2 C\pi^{-1}\left(\frac{\mathcal{L}}{2}\right)^2 + 2 \times 4\alpha^2 C\pi^{-1}\left(\frac{\mathcal{L}}{4}\right)^2 + \cdots + 2^{\mathcal{D}-2} \times 4\alpha^2 C\pi^{-1}\left(\frac{\mathcal{L}}{2^{\mathcal{D}-1}}\right)^2. \tag{110}$$

Thus, we conclude:

$$\left\| \left\langle \nabla_{\overline{\boldsymbol{w}}^{(m),(L)}} h^{(m)}\left(\boldsymbol{x}; \boldsymbol{\theta}_0^{(m)}\right), \nabla_{\boldsymbol{w}^{(m),(L)}} h^{(m)}\left(\tilde{\boldsymbol{x}}; \boldsymbol{\theta}_0^{(m)}\right) \right\rangle \right\|_{\psi_1} \tag{111}$$

$$\leq \frac{4\alpha^2 C}{\pi} \sum_{i=1}^{\mathcal{D}-1} 2^{i-1} \frac{\mathcal{L}^2}{2^{2i}} \tag{112}$$

$$\leq \frac{2\alpha^2 C\mathcal{L}^2}{\pi} \sum_{i=1}^{\mathcal{D}-1} 2^{-i} \tag{113}$$

$$\leq \frac{2\alpha^2 C\mathcal{L}^2}{\pi} \tag{114}$$

$$\leq 4\alpha^2 C\mathcal{L}^2 \tag{115}$$

$$= K. \tag{116}$$

Finally, by applying centering lemma (Lemma E.3), Bernstein's inequality (Lemma E.2), and the union bound, we obtain the desired result. □

*Proof of Lemma B.3.* From the definition of $h^{(m)}$, we have:

$$\nabla_{\overline{\boldsymbol{\pi}}^{(m)}} h^{(m)}\left(\boldsymbol{x}; \boldsymbol{\theta}_0^{(m)}\right) = \left(p_1\left(\boldsymbol{x}; \overline{\boldsymbol{w}}^{(m)}\right), \ldots, p_{\mathcal{L}}\left(\boldsymbol{x}; \overline{\boldsymbol{w}}^{(m)}\right)\right)^\top \tag{117}$$

Noting that $|p_l\left(\boldsymbol{x}; \boldsymbol{w}\right)| \leq 1$, we have:

$$\left\| \left\langle \nabla_{\overline{\boldsymbol{\pi}}^{(m)}} h^{(m)}\left(\boldsymbol{x}; \boldsymbol{\theta}_0^{(m)}\right), \nabla_{\overline{\boldsymbol{\pi}}^{(m)}} h^{(m)}\left(\boldsymbol{x}; \boldsymbol{\theta}_0^{(m)}\right) \right\rangle \right\|_{\psi_2} \leq \sum_{l=1}^{\mathcal{L}} \left\| p_l\left(\boldsymbol{x}; \overline{\boldsymbol{w}}^{(m)}\right)^2 \right\|_{\psi_2} \tag{118}$$

$$\leq C\mathcal{L}. \tag{119}$$

Therefore, by applying the centering lemma (Lemma E.3) and the general Hoeffding's inequality (Lemma E.1) with union bounds, the desired result is obtained. □

Lemma 3.2 is derived by taking a union bound over the three preceding lemmas and rearranging the entire expression.

*Proof of Lemma 3.2.* Fix any $\epsilon > 0$ such that $\epsilon \leq K$. Then, $\min\left\{\frac{\epsilon^2}{K^2}, \frac{\epsilon}{K}\right\} = \frac{\epsilon^2}{K^2}$. Now,

$$M \geq \frac{K^2}{c\epsilon^2} \ln \frac{16}{\delta} \Rightarrow 1 - 4\exp\left(-c\frac{\epsilon^2}{K^2}M\right) \geq 1 - \frac{\delta}{4}, \tag{120}$$

$$M \geq \frac{\mathcal{L}^2}{\tilde{c}\epsilon^2} \ln \frac{16}{\delta} \Rightarrow 1 - 4\exp\left(-\frac{\tilde{c}\epsilon^2 M}{\mathcal{L}^2}\right) \geq 1 - \frac{\delta}{4}. \tag{121}$$

Therefore, from Lemma B.1, Lemma B.2, and Lemma B.3, by applying the union bound,

$$M \geq \max\left\{\frac{K^2}{c}, \frac{\mathcal{L}^2}{\tilde{c}}\right\} \epsilon^{-2} \ln \frac{16}{\delta} \tag{122}$$

$$\Rightarrow \mathbb{P}\left(|k_{\mathrm{TNTK}}(\boldsymbol{x}, \tilde{\boldsymbol{x}}) - \langle g(\boldsymbol{x}, \boldsymbol{\theta}_0), g(\tilde{\boldsymbol{x}}, \boldsymbol{\theta}_0)\rangle| \leq 4\varepsilon\right) \geq 1 - \delta. \tag{123}$$

Finally, let $\tilde{C} = \max\{1/c, 1/\tilde{c}\}$, then

$$M \geq \tilde{C} \max\left\{K^2, \mathcal{L}^2\right\} \epsilon^{-2} \ln \frac{16}{\delta} \tag{124}$$

$$\Rightarrow M \geq \max\left\{\frac{K^2}{c}, \frac{\mathcal{L}^2}{\tilde{c}}\right\} \epsilon^{-2} \ln \frac{16}{\delta}. \tag{125}$$

By defining $C_{\alpha,\mathcal{D}}^{(2)}$ as $C_{\alpha,\mathcal{D}}^{(2)} = K$, the desired result is obtained. $\square$

## B.2 Proof of Lemma 3.3

**Proof scketch** Since the parameters of the different soft trees are independent, we can confirm that the Hessian $\boldsymbol{H}(\boldsymbol{x}, \boldsymbol{\theta})$ is given as the block diagonal matrix. Since we know the fact that the spectral norm of the block diagonal matrix equals the maximum over the spectral norms of the block matrix, the remaining interest is the upper bound of the spectral norm of each block matrix. Then, we obtain Lemma 3.3 by carefully evaluating the upper bound of the spectral norm of each block matrix with its Frobenius norm.

*Proof of Lemma 3.3.* Define $\boldsymbol{H}^{(m)}\left(\boldsymbol{x}, \boldsymbol{\theta}^{(m)}\right) = \nabla_{\boldsymbol{\theta}^{(m)}}^2 h^{(m)}\left(\boldsymbol{x}; \boldsymbol{\theta}^{(m)}\right) \in \mathbb{R}^{\tilde{p} \times \tilde{p}}$, where $\tilde{p} = d\mathcal{N} + \mathcal{L}$. Then, $\boldsymbol{H}(\boldsymbol{x}, \boldsymbol{\theta})$ is represented by the following block diagonal matrix:

$$\boldsymbol{H}(\boldsymbol{x}, \boldsymbol{\theta}) = \frac{1}{\sqrt{M}} \begin{pmatrix} \boldsymbol{H}^{(1)}\left(\boldsymbol{x}, \boldsymbol{\theta}^{(1)}\right) & \boldsymbol{0}_{\tilde{p} \times \tilde{p}} & \cdots & \boldsymbol{0}_{\tilde{p} \times \tilde{p}} \\ \boldsymbol{0}_{\tilde{p} \times \tilde{p}} & \boldsymbol{H}^{(2)}\left(\boldsymbol{x}, \boldsymbol{\theta}^{(2)}\right) & \cdots & \boldsymbol{0}_{\tilde{p} \times \tilde{p}} \\ \vdots & \vdots & \ddots & \vdots \\ \boldsymbol{0}_{\tilde{p} \times \tilde{p}} & \boldsymbol{0}_{\tilde{p} \times \tilde{p}} & \cdots & \boldsymbol{H}^{(M)}\left(\boldsymbol{x}, \boldsymbol{\theta}^{(M)}\right) \end{pmatrix}, \tag{126}$$

where $\boldsymbol{0}_{\tilde{p} \times \tilde{p}}$ represents a $\tilde{p} \times \tilde{p}$ zero matrix. Therefore,

$$\|\boldsymbol{H}(\boldsymbol{x}, \boldsymbol{\theta})\| = \frac{1}{\sqrt{M}} \max_{m \in [M]} \left\|\boldsymbol{H}_m\left(\boldsymbol{x}, \boldsymbol{\theta}^{(m)}\right)\right\|. \tag{127}$$

Here, assume the following event holds:

$$\forall m \in [M], \ \forall l \in [\mathcal{L}], \ \forall n \in [\mathcal{N}],$$

$$\left|\bar{\pi}_l^{(m)}\right| \leq \sqrt{2\ln\frac{2M(\mathcal{L} + \mathcal{N})}{\delta}} \quad \text{and} \quad \left|\bar{\boldsymbol{w}}_n^{(m)\top}\boldsymbol{x}\right| \leq \sqrt{2\ln\frac{2M(\mathcal{L} + \mathcal{N})}{\delta}}. \tag{128}$$

Since $\boldsymbol{\theta}_0$ is initialized by a standard normal distribution, by the union bound, the above event occurs with probability at least $1 - \delta$. Therefore, it is sufficient to show that Eq. (11) holds under the event (128).

Now, the derivatives of $h^{(m)}(\boldsymbol{x}; \boldsymbol{\theta}^{(m)})$ up to the second order are given by:

$$\frac{\partial^2 h^{(m)}\left(\boldsymbol{x}; \boldsymbol{\theta}^{(m)}\right)}{\partial \boldsymbol{w}_n^{(m)} \partial \boldsymbol{w}_{\tilde{n}}^{(m)}} = \sum_{l=1}^{\mathcal{L}} \pi_l^{(m)} \frac{\partial^2 p_l(\boldsymbol{x}; \boldsymbol{w}^{(m)})}{\partial \boldsymbol{w}_n^{(m)} \partial \boldsymbol{w}_{\tilde{n}}^{(m)}}, \tag{129}$$

$$\frac{\partial^2 h^{(m)}\left(\boldsymbol{x}; \boldsymbol{\theta}^{(m)}\right)}{\partial \boldsymbol{w}_n^{(m)} \partial \pi_l^{(m)}} = \frac{\partial p_l(\boldsymbol{x}; \boldsymbol{w}^{(m)})}{\partial \boldsymbol{w}_n^{(m)}}, \tag{130}$$

$$\frac{\partial^2 h^{(m)}\left(\boldsymbol{x}; \boldsymbol{\theta}^{(m)}\right)}{\partial \pi_l^{(m)} \partial \pi_{\tilde{l}}^{(m)}} = 0. \tag{131}$$

From the definition of $p_l$, we have

$$\frac{\partial p_l(\boldsymbol{x}; \boldsymbol{w}^{(m)})}{\partial \boldsymbol{w}_n^{(m)}} = \left[ \mathbb{1}_{l \swarrow n} \boldsymbol{x} \dot{\sigma}\left(\boldsymbol{w}_n^{(m)\top}\boldsymbol{x}\right) - \mathbb{1}_{n \searrow l} \boldsymbol{x} \dot{\sigma}\left(\boldsymbol{w}_n^{(m)\top}\boldsymbol{x}\right) \right]$$
$$\times \prod_{\tilde{n} \neq n} \sigma\left(\boldsymbol{w}_{\tilde{n}}^{(m)\top}\boldsymbol{x}\right)^{\mathbb{1}_{l \swarrow \tilde{n}}} \left[1 - \sigma\left(\boldsymbol{w}_{\tilde{n}}^{(m)\top}\boldsymbol{x}\right)\right]^{\mathbb{1}_{\tilde{n} \searrow l}}, \tag{132}$$

$$\frac{\partial^2 p_l\left(\boldsymbol{x}; \boldsymbol{w}^{(m)}\right)}{\partial \boldsymbol{w}_n^{(m)} \partial \boldsymbol{w}_n^{(m)}} = \left[ \mathbb{1}_{l \swarrow n} \boldsymbol{x}\boldsymbol{x}^\top \ddot{\sigma}\left(\boldsymbol{w}_n^{(m)\top}\boldsymbol{x}\right) - \mathbb{1}_{n \searrow l} \boldsymbol{x}\boldsymbol{x}^\top \ddot{\sigma}\left(\boldsymbol{w}_n^{(m)\top}\boldsymbol{x}\right) \right]$$
$$\times \prod_{\tilde{n} \neq n} \sigma\left(\boldsymbol{w}_{\tilde{n}}^{(m)\top}\boldsymbol{x}\right)^{\mathbb{1}_{l \swarrow \tilde{n}}} \left[1 - \sigma\left(\boldsymbol{w}_{\tilde{n}}^{(m)\top}\boldsymbol{x}\right)\right]^{\mathbb{1}_{\tilde{n} \searrow l}}, \tag{133}$$

$$\frac{\partial^2 p_l\left(\boldsymbol{x}; \boldsymbol{w}^{(m)}\right)}{\partial \boldsymbol{w}_n^{(m)} \partial \boldsymbol{w}_{\hat{n}}^{(m)}} = \left[ \mathbb{1}_{l \swarrow n} \boldsymbol{x} \dot{\sigma}\left(\boldsymbol{w}_n^{(m)\top}\boldsymbol{x}\right) - \mathbb{1}_{n \searrow l} \boldsymbol{x} \dot{\sigma}\left(\boldsymbol{w}_n^{(m)\top}\boldsymbol{x}\right) \right]$$
$$\times \left[ \mathbb{1}_{l \swarrow \hat{n}} \boldsymbol{x} \dot{\sigma}\left(\boldsymbol{w}_{\hat{n}}^{(m)\top}\boldsymbol{x}\right) - \mathbb{1}_{\hat{n} \searrow l} \boldsymbol{x} \dot{\sigma}\left(\boldsymbol{w}_{\hat{n}}^{(m)\top}\boldsymbol{x}\right) \right]^\top \tag{134}$$
$$\times \prod_{\tilde{n} \neq n, \tilde{n} \neq \hat{n}} \sigma\left(\boldsymbol{w}_{\tilde{n}}^{(m)\top}\boldsymbol{x}\right)^{\mathbb{1}_{l \swarrow \tilde{n}}} \left[1 - \sigma\left(\boldsymbol{w}_{\tilde{n}}^{(m)\top}\boldsymbol{x}\right)\right]^{\mathbb{1}_{\tilde{n} \searrow l}}.$$

In the third equation, it was assumed that $n \neq \hat{n}$. Now, let us evaluate the upper bound of the above expressions. First, from the definition of $\sigma(\cdot)$, we know that $\|\sigma(\cdot)\|_\infty \leq 1$ and $\|\dot{\sigma}(\cdot)\|_\infty \leq \alpha/\sqrt{\pi}$. Additionally, for any $a \in \mathbb{R}$, $|\ddot{\sigma}(a)| \leq 2|a|\alpha^2/\sqrt{\pi}$. Then,

$$\left\| \frac{\partial^2 p_l\left(\boldsymbol{x}; \boldsymbol{w}^{(m)}\right)}{\partial \boldsymbol{w}_n^{(m)} \partial \boldsymbol{w}_n^{(m)}} \right\|_F \leq \left| \ddot{\sigma}\left(\boldsymbol{w}_n^{(m)\top}\boldsymbol{x}\right) \right| \left\| \mathbb{1}_{l \swarrow n} \boldsymbol{x}\boldsymbol{x}^\top - \mathbb{1}_{n \searrow l} \boldsymbol{x}\boldsymbol{x}^\top \right\|_F \tag{135}$$

$$\leq 2 \left| \boldsymbol{w}_n^{(m)\top}\boldsymbol{x} \right| \frac{\alpha^2}{\sqrt{\pi}} \left\| \boldsymbol{x}\boldsymbol{x}^\top \right\|_F \tag{136}$$

$$\leq 2 \left( \left| \boldsymbol{w}_n^{(m)\top}\boldsymbol{x} - \overline{\boldsymbol{w}}_n^{(m)\top}\boldsymbol{x} \right| + \left| \overline{\boldsymbol{w}}_n^{(m)\top}\boldsymbol{x} \right| \right) \frac{\alpha^2}{\sqrt{\pi}} \|\boldsymbol{x}\|_2^2 \tag{137}$$

$$\leq 2 \left( R + \sqrt{2 \ln \frac{2M(\mathcal{L} + \mathcal{N})}{\delta}} \right) \frac{\alpha^2}{\sqrt{\pi}}. \tag{138}$$

Furthermore, as for $n \neq \hat{n}$,

$$\left\| \frac{\partial^2 p_j(\boldsymbol{x}; \boldsymbol{w}^{(m)})}{\partial \boldsymbol{w}_n^{(m)} \partial \boldsymbol{w}_{\hat{n}}^{(m)}} \right\|_F \tag{139}$$

$$\leq \left\| \left[ \mathbb{1}_{l \swarrow n} \boldsymbol{x} \dot{\sigma}\left(\boldsymbol{w}_n^{(m)\top}\boldsymbol{x}\right) - \mathbb{1}_{n \searrow l} \boldsymbol{x} \dot{\sigma}\left(\boldsymbol{w}_n^{(m)\top}\boldsymbol{x}\right) \right] \right.$$
$$\left. \cdot \left[ \mathbb{1}_{l \swarrow \hat{n}} \boldsymbol{x} \dot{\sigma}\left(\boldsymbol{w}_{\hat{n}}^{(m)\top}\boldsymbol{x}\right) - \mathbb{1}_{\hat{n} \searrow l} \boldsymbol{x} \dot{\sigma}\left(\boldsymbol{w}_{\hat{n}}^{(m)\top}\boldsymbol{x}\right) \right]^\top \right\|_F \tag{140}$$

$$= \left\| \mathbb{1}_{l \swarrow n} \mathbb{1}_{l \swarrow \hat{n}} \dot{\sigma}\left(\boldsymbol{w}_n^{(m)\top}\boldsymbol{x}\right) \dot{\sigma}\left(\boldsymbol{w}_{\hat{n}}^{(m)\top}\boldsymbol{x}\right) \boldsymbol{x}\boldsymbol{x}^\top - \mathbb{1}_{n \searrow l} \mathbb{1}_{l \swarrow \hat{n}} \dot{\sigma}\left(\boldsymbol{w}_n^{(m)\top}\boldsymbol{x}\right) \dot{\sigma}\left(\boldsymbol{w}_{\hat{n}}^{(m)\top}\boldsymbol{x}\right) \boldsymbol{x}\boldsymbol{x}^\top \right.$$
$$\left. - \mathbb{1}_{l \swarrow n} \mathbb{1}_{\hat{n} \searrow l} \dot{\sigma}\left(\boldsymbol{w}_n^{(m)\top}\boldsymbol{x}\right) \dot{\sigma}\left(\boldsymbol{w}_{\hat{n}}^{(m)\top}\boldsymbol{x}\right) \boldsymbol{x}\boldsymbol{x}^\top + \mathbb{1}_{n \searrow l} \mathbb{1}_{\hat{n} \searrow l} \dot{\sigma}\left(\boldsymbol{w}_n^{(m)\top}\boldsymbol{x}\right) \dot{\sigma}\left(\boldsymbol{w}_{\hat{n}}^{(m)\top}\boldsymbol{x}\right) \boldsymbol{x}\boldsymbol{x}^\top \right\|_F$$
$$\tag{141}$$

$$\leq \|\dot{\sigma}(\cdot)\|_\infty^2 \|\boldsymbol{x}\boldsymbol{x}^\top\|_F \tag{142}$$

$$\leq \frac{\alpha^2}{\pi}. \tag{143}$$

Moreover, we have

$$\left\| \frac{\partial^2 h^{(m)}\left(\boldsymbol{x}; \boldsymbol{\theta}^{(m)}\right)}{\partial \boldsymbol{w}_n^{(m)} \partial \boldsymbol{w}_{\hat{n}}^{(m)}} \right\|_F \leq \sum_{l=1}^{\mathcal{L}} \left| \pi_l^{(m)} \right| \left\| \frac{\partial^2 p_l\left(\boldsymbol{x}; \boldsymbol{w}^{(m)}\right)}{\partial \boldsymbol{w}_n^{(m)} \partial \boldsymbol{w}_{\hat{n}}^{(m)}} \right\|_F \tag{144}$$

$$\leq \sum_{l=1}^{\mathcal{L}} \left( \left| \pi_l^{(m)} - \bar{\pi}_l^{(m)} \right| + \left| \bar{\pi}_l^{(m)} \right| \right) \left\| \frac{\partial^2 p_l\left(\boldsymbol{x}; \boldsymbol{w}^{(m)}\right)}{\partial \boldsymbol{w}_n^{(m)} \partial \boldsymbol{w}_{\hat{n}}^{(m)}} \right\|_F \tag{145}$$

$$\leq \sum_{l=1}^{\mathcal{L}} \left( R + \sqrt{2 \ln \frac{2M(\mathcal{L}+\mathcal{N})}{\delta}} \right) \left\| \frac{\partial^2 p_l\left(\boldsymbol{x}; \boldsymbol{w}^{(m)}\right)}{\partial \boldsymbol{w}_n^{(m)} \partial \boldsymbol{w}_{\hat{n}}^{(m)}} \right\|_F. \tag{146}$$

Therefore,

$$\left\| \boldsymbol{H}^{(m)}\left(\boldsymbol{x}, \boldsymbol{\theta}^{(m)}\right) \right\|^2 \tag{147}$$

$$\leq \left\| \boldsymbol{H}^{(m)}\left(\boldsymbol{x}, \boldsymbol{\theta}^{(m)}\right) \right\|_F^2 \tag{148}$$

$$= \sum_{n=1}^{\mathcal{N}} \sum_{\hat{n}=1}^{\mathcal{N}} \left\| \frac{\partial^2 h^{(m)}\left(\boldsymbol{x}; \boldsymbol{\theta}^{(m)}\right)}{\partial \boldsymbol{w}_n^{(m)} \partial \boldsymbol{w}_{\hat{n}}^{(m)}} \right\|_F^2 + 2 \sum_{n=1}^{\mathcal{N}} \sum_{l=1}^{\mathcal{L}} \left\| \frac{\partial^2 h^{(m)}\left(\boldsymbol{x}; \boldsymbol{\theta}^{(m)}\right)}{\partial \boldsymbol{w}_n^{(m)} \partial \pi_l^{(m)}} \right\|_F^2$$
$$+ \sum_{l=1}^{\mathcal{L}} \sum_{\tilde{l}=1}^{\mathcal{L}} \left( \frac{\partial^2 h^{(m)}\left(\boldsymbol{x}; \boldsymbol{\theta}^{(m)}\right)}{\partial \pi_l^{(m)} \partial \pi_{\tilde{l}}^{(m)}} \right)^2 \tag{149}$$

$$\leq \sum_{n=1}^{\mathcal{N}} \left\| \frac{\partial^2 h^{(m)}\left(\boldsymbol{x}; \boldsymbol{\theta}^{(m)}\right)}{\partial \boldsymbol{w}_n^{(m)} \partial \boldsymbol{w}_n^{(m)}} \right\|_F^2 + \sum_{n \neq \hat{n}} \left\| \frac{\partial^2 h^{(m)}\left(\boldsymbol{x}; \boldsymbol{\theta}^{(m)}\right)}{\partial \boldsymbol{w}_n^{(m)} \partial \boldsymbol{w}_{\hat{n}}^{(m)}} \right\|_F^2 + 2 \sum_{n=1}^{\mathcal{N}} \sum_{l=1}^{\mathcal{L}} \left\| \frac{\partial^2 h^{(m)}\left(\boldsymbol{x}; \boldsymbol{\theta}^{(m)}\right)}{\partial \boldsymbol{w}_n^{(m)} \partial \pi_l^{(m)}} \right\|_F^2 \tag{150}$$

$$\leq \sum_{n=1}^{\mathcal{N}} \left[ \sum_{l=1}^{\mathcal{L}} \left( R + \sqrt{2 \ln \frac{2M(\mathcal{L}+\mathcal{N})}{\delta}} \right) \left\| \frac{\partial^2 p_l\left(\boldsymbol{x}; \boldsymbol{w}^{(m)}\right)}{\partial \boldsymbol{w}_n^{(m)} \partial \boldsymbol{w}_n^{(m)}} \right\|_F \right]^2$$
$$+ \sum_{n \neq \hat{n}} \left[ \sum_{l=1}^{\mathcal{L}} \left( R + \sqrt{2 \ln \frac{2M(\mathcal{L}+\mathcal{N})}{\delta}} \right) \left\| \frac{\partial^2 p_l\left(\boldsymbol{x}; \boldsymbol{w}^{(m)}\right)}{\partial \boldsymbol{w}_n^{(m)} \partial \boldsymbol{w}_{\hat{n}}^{(m)}} \right\|_F \right]^2 \tag{151}$$
$$+ 2 \sum_{n=1}^{\mathcal{N}} \sum_{l=1}^{\mathcal{L}} \left\| \frac{\partial p_l\left(\boldsymbol{x}; \boldsymbol{w}^{(m)}\right)}{\partial \boldsymbol{w}_n^{(m)}} \right\|_F^2$$

$$\leq \sum_{n=1}^{\mathcal{N}} 4\mathcal{L}^2 \left( R + \sqrt{2 \ln \frac{2M(\mathcal{L}+\mathcal{N})}{\delta}} \right)^4 \frac{\alpha^4}{\pi} + \sum_{n \neq \hat{n}} \mathcal{L}^2 \left( R + \sqrt{2 \ln \frac{2M(\mathcal{L}+\mathcal{N})}{\delta}} \right)^2 \frac{\alpha^4}{\pi^2}$$
$$+ 2\mathcal{N}\mathcal{L}\frac{\alpha^2}{\pi} \tag{152}$$

$$\leq 4\mathcal{N}^2\mathcal{L}^2 \left( R + \sqrt{2 \ln \frac{2M(\mathcal{L}+\mathcal{N})}{\delta}} \right)^4 \frac{\alpha^4}{\pi} + 2\mathcal{N}\mathcal{L}\frac{\alpha^2}{\pi} \tag{153}$$

$$\leq 4\mathcal{N}^2\mathcal{L}^2 \left( R + \sqrt{2 \ln \frac{2M(\mathcal{L}+\mathcal{N})}{\delta}} \right)^4 \alpha^4 + 2\mathcal{N}^2\mathcal{L}^2\alpha^4 \tag{154}$$

$$\leq 6\mathcal{N}^2\mathcal{L}^2 \left( R + \sqrt{2 \ln \frac{2M(\mathcal{L}+\mathcal{N})}{\delta}} \right)^4 \alpha^4, \tag{155}$$

where:

- Eq. (150) follows from Eq. (131).

- Eq. (151) follows from Eqs. (146) and (130).

- The first and second term of Eq. (152) follows from Eq. (143) and Eq. (138), respectively. Furthermore, the third term of Eq. (152) follows from $\left\|\frac{\partial p_l\left(\boldsymbol{x};\boldsymbol{w}^{(j)}\right)}{\partial \boldsymbol{w}_n^{(j)}}\right\|_F \leq$
  $\left\|\mathbb{1}_{l\swarrow n}\boldsymbol{x}\dot{\sigma}\left(\boldsymbol{w}_n^{(j)\top}\boldsymbol{x}\right) - \mathbb{1}_{n\searrow l}\boldsymbol{x}\dot{\sigma}\left(\boldsymbol{w}_n^{(j)\top}\boldsymbol{x}\right)\right\|_F \leq \|\dot{\sigma}(\cdot)\|_\infty \leq \frac{\alpha}{\sqrt{\pi}}$.

- Eq. (154) follows from $1/\pi \leq 1$ and $\alpha \geq 1$.

- Eq. (155) follows from $\left(R + \sqrt{2\ln\frac{2M(\mathcal{L}+\mathcal{N})}{\delta}}\right)^4 \geq (\sqrt{2\ln 2})^4 \geq 1$.

By combining Eq. (155) with Eq. (127), we have

$$\|\boldsymbol{H}(\boldsymbol{x},\boldsymbol{\theta})\| \leq \frac{\sqrt{6}\alpha^2\mathcal{N}\mathcal{L}}{\sqrt{M}}\left(R + \sqrt{2\ln\frac{2M(\mathcal{L}+\mathcal{N})}{\delta}}\right)^2 \tag{156}$$

$$\leq \frac{\sqrt{6}\alpha^2 2^{2\mathcal{D}}}{\sqrt{M}}\left(R + \sqrt{2\ln\frac{2M(\mathcal{L}+\mathcal{N})}{\delta}}\right)^2. \tag{157}$$

Finally, from the definition of $C_{\alpha,\mathcal{D}}^{(3)}$,

$$\sqrt{6}\alpha^2 2^{2\mathcal{D}}(R + \sqrt{2})^2 = C_{\alpha,\mathcal{D}}^{(3)}(R + \sqrt{2})^2 \tag{158}$$

$$\Rightarrow \sqrt{6}\alpha^2 2^{2\mathcal{D}}\left(\frac{R}{\sqrt{\ln\frac{2M(\mathcal{L}+\mathcal{N})}{\delta}}} + \sqrt{2}\right)^2 \leq C_{\alpha,\mathcal{D}}^{(3)}(R + \sqrt{2})^2 \tag{159}$$

$$\Leftrightarrow \frac{\sqrt{6}\alpha^2 2^{2\mathcal{D}}}{\sqrt{M}}\left(R + \sqrt{2\ln\frac{2M(\mathcal{L}+\mathcal{N})}{\delta}}\right)^2 \leq \frac{C_{\alpha,\mathcal{D}}^{(3)}(R + \sqrt{2})^2}{\sqrt{M}}\ln\frac{2M(\mathcal{L}+\mathcal{N})}{\delta} \tag{160}$$

$$\Leftrightarrow \frac{\sqrt{6}\alpha^2 2^{2\mathcal{D}}}{\sqrt{M}}\left(R + \sqrt{2\ln\frac{2M(\mathcal{L}+\mathcal{N})}{\delta}}\right)^2 \leq \frac{C_{\alpha,\mathcal{D}}^{(3)}(R + \sqrt{2})^2}{\sqrt{M}}\ln\frac{2^{\mathcal{D}+2}M}{\delta}, \tag{161}$$

where Eq. (159) follows from $\ln(2M(\mathcal{N}+\mathcal{L})/\delta) \geq \ln 6 \geq 1$. Furthermore, Eq. (161) follows from $\mathcal{L}+\mathcal{N} \leq 2^{\mathcal{D}+1}$. By combining Eq. (161) with Eq. (157), we obtain the desired result. $\square$

## B.3 Proof of Theorem 3.2

Instead of showing Theorem 3.2 directly, we show the proof of the following detailed version of Theorem 3.2.

**Theorem B.1** (Detailed version of Theorem 3.2). *Suppose that Assumption 3.1 holds. Fix any $\delta \in (0,1)$, $\alpha \geq 1$, $\rho > 0$, and $\mathcal{D} \geq 2$. Furthermore, suppose that the number of ensemble $M$ is sufficiently large to satisfy the following four conditions:*

$$M \geq 64 C_{\alpha,\mathcal{D}}^{(6)}|\mathcal{X}|^2\lambda_0^{-2}\ln\frac{16|\mathcal{X}|^2}{\delta}, \tag{162}$$

$$M \geq C_{\alpha,\mathcal{D}}^{(6)}C_{\alpha,\mathcal{D}}^{(2)-2}\ln\frac{16|\mathcal{X}|^2}{\delta}, \tag{163}$$

$$\tilde{R}^4(\tilde{R} + 2)^4 \leq \frac{3\eta^2 M\rho^2}{56C_{\alpha,\mathcal{D}}^{(3)2}}\left(\overline{k}B + \sigma\sqrt{2\ln\frac{12T}{\delta}}\right)^2\left(\ln\frac{6\cdot 2^{\mathcal{D}+2}M}{\delta}\right)^{-2}, \tag{164}$$

$$\frac{C_{\alpha,\mathcal{D},T}^{(7)}}{\sqrt{M}}\left(\overline{k}B + \sigma\sqrt{2\ln\frac{12T}{\delta}}\right)\left(\ln\frac{6\cdot 2^{\mathcal{D}+2}M}{\delta}\right)\sqrt{\ln\frac{6M}{\delta}} \leq 1, \tag{165}$$

*where:*

$$\overline{R} = \tilde{R} + \frac{1}{2\rho}\left[(2\tilde{R} + 2^{\mathcal{D}})\sqrt{TC_{\alpha,\mathcal{D}}^{(4)}\ln\frac{6M}{\delta}}\left(\overline{k}B + \sigma\sqrt{2\ln\frac{12T}{\delta}}\right)\sqrt{3T} + \tilde{R}\right], \quad (166)$$

$$\tilde{R} = 2\left(\overline{k}B + \sigma\sqrt{2\ln\frac{12T}{\delta}}\right)\sqrt{\frac{T}{\rho}}. \quad (167)$$

*Then, if the learning rate $\eta$ satisfy $\eta \leq 4^{-1}\left(\rho + 2(2\tilde{R} + 2^{\mathcal{D}}\hat{C})^2 TC_{\alpha,\mathcal{D}}^{(4)}\ln\frac{6M}{\delta}\right)^{-1}$, with probability at least $1 - \delta$, the following inequality holds for any $t \in [T]$ and $\boldsymbol{x} \in \mathcal{X}$:*

$$|f(\boldsymbol{x}) - h(\boldsymbol{x};\boldsymbol{\theta}_{t-1})| \leq \frac{72T^2 C_{\alpha,\mathcal{D}}^{(3)}}{\sqrt{M}\rho^2}\left(\overline{k}B + \sigma\sqrt{2\ln\frac{12T}{\delta}}\right)^4 \ln\frac{6 \cdot 2^{\mathcal{D}+2}M}{\delta} + \beta\tilde{\sigma}_{t-1}(\boldsymbol{x}), \quad (168)$$

*where:*

$$\beta = \left(\sqrt{2}B + \frac{\sigma}{\sqrt{\rho}}\sqrt{2\left(\gamma_T + \frac{T\sqrt{TC_{\alpha,\mathcal{D}}^{(6)}\ln(96|\mathcal{X}|^2/\delta)}}{\rho\sqrt{M}} + \ln\frac{6}{\delta}\right)}\right)$$

$$+ \rho^{-1}\sqrt{\overline{k}^2 + 4C_{\alpha,\mathcal{D}}^{(2)}}\left[\frac{C_{\alpha,\mathcal{D},T}^{(7)}}{\sqrt{M}}\left(\overline{k}B + \sigma\sqrt{2\ln\frac{12T}{\delta}}\right)^2\left(\ln\frac{6 \cdot 2^{\mathcal{D}+2}M}{\delta}\right)\sqrt{\ln\frac{6M}{\delta}} \quad (169)$$

$$+ (1 - 2\eta\rho)^{J/2}\left(\overline{k}B + \sigma\sqrt{2\ln\frac{12T}{\delta}}\right)^2\sqrt{\frac{T}{\rho}}\right]\left(\rho + TC_{\alpha,\mathcal{D}}^{(4)}2^{2\mathcal{D}}\hat{C}^2\ln\frac{6M}{\delta}\right).$$

*Here, $C > 0$ and $\hat{C} > 0$ are absolute constants. Furthermore, $C_{\alpha,\mathcal{D}}^{(4)} > 0$, $C_{\alpha,\mathcal{D}}^{(5)} > 0$, and $C_{\alpha,\mathcal{D}}^{(6)} > 0$ are constants that depend on $\alpha$ and $\mathcal{D}$. Moreover, $\overline{k} := \max_{\boldsymbol{x} \in \mathcal{X}}\sqrt{k_{\mathrm{TNTK}}(\boldsymbol{x},\boldsymbol{x})}$ is the square root of the maximum value of TNTK, and $C_{\alpha,\mathcal{D},T}^{(7)} = \mathcal{O}(T^3)$ is the constant that depends on $\alpha$, $\mathcal{D}$, and $T$.*

### B.3.1 Proof overview

In this section, we briefly summarize the overview of our proof. We first define the following six events:

- $\mathcal{E}_1 = \left\{\forall\boldsymbol{\theta},\boldsymbol{x},R,\|\boldsymbol{\theta} - \boldsymbol{\theta}_0\|_2 \leq R \Rightarrow \|\boldsymbol{H}(\boldsymbol{x},\boldsymbol{\theta})\| \leq \frac{C_{\alpha,\mathcal{D}}^{(3)}(R+2)^2}{\sqrt{M}}\ln\frac{6 \cdot 2^{\mathcal{D}+2}M}{\delta}\right\}$.

- $\mathcal{E}_2 = \left\{\forall\boldsymbol{\theta},\boldsymbol{x},R,\|\boldsymbol{\theta} - \boldsymbol{\theta}_0\|_2 \leq R \Rightarrow \|\boldsymbol{g}(\boldsymbol{x};\boldsymbol{\theta})\|_2^2 \leq C_{\alpha,\mathcal{D}}^{(4)}(2R + 2^{\mathcal{D}}\hat{C})^2\ln\frac{6M}{\delta}\right\}$.

- $\mathcal{E}_3 = \left\{\forall\boldsymbol{\theta},\boldsymbol{x},R,\|\boldsymbol{\theta} - \boldsymbol{\theta}_0\|_2 \leq R \Rightarrow \|\boldsymbol{g}(\boldsymbol{x};\boldsymbol{\theta}) - \boldsymbol{g}(\boldsymbol{x};\boldsymbol{\theta}_0)\|_2^2 \leq \frac{C_{\alpha,\mathcal{D}}^{(5)}R^2}{M}\ln\frac{6M}{\delta}\right\}$.

- $\mathcal{E}_4 = \left\{\forall t \in [T], \|\boldsymbol{y}_t\|_2 \leq \left(\overline{k}B + \sigma\sqrt{2\ln\frac{12T}{\delta}}\right)\sqrt{t}\right\}$.

- $\mathcal{E}_5 = \left\{\forall\boldsymbol{x},\tilde{\boldsymbol{x}}, |k_{\mathrm{TNTK}}(\boldsymbol{x},\tilde{\boldsymbol{x}}) - \tilde{k}(\boldsymbol{x},\tilde{\boldsymbol{x}})| \leq \min\left\{\frac{\lambda_0}{2|\mathcal{X}|}, \sqrt{\frac{4C_{\alpha,\mathcal{D}}^{(6)}}{M}\ln\frac{96|\mathcal{X}|^2}{\delta}}, 4C_{\alpha,\mathcal{D}}^{(2)}\right\}\right\}$.

- $\mathcal{E}_6 = \left\{\forall t \in \mathbb{N}_+, \forall\boldsymbol{x} \in \mathcal{X}, |f(\boldsymbol{x}) - \tilde{\mu}_{t-1}(\boldsymbol{x})| \leq \left(\sqrt{2}B + \frac{\sigma}{\sqrt{\rho}}\sqrt{2(\tilde{\gamma}_t + \ln\frac{6}{\delta})}\right)\tilde{\sigma}_{t-1}(\boldsymbol{x})\right\}$.

The quantities $\overline{k}$, $C_{\alpha,\mathcal{D}}^{(4)}$, $C_{\alpha,\mathcal{D}}^{(5)}$, $C_{\alpha,\mathcal{D}}^{(6)}$, $\tilde{\mu}_{t-1}$, $\tilde{k}$, and $\tilde{\gamma}_t$ are defined in Lemma B.4–B.7. Only the above six events require probabilistic arguments in our proof. Actually, from Lemma 3.3 and Lemma B.4–B.7, which we will show later, we can confirm the events $\mathcal{E}_1, \ldots, \mathcal{E}_6$ simultaneously holds with probability at least $1 - \delta$ for sufficiently large $M$ by taking union bound; therefore, it is enough to show Eq. (168) under the event $\bigcap_{i \in [6]}\mathcal{E}_i$. Hereafter, we show Theorem B.1 in the following steps:

1. For sufficiently large $M$, we show that each of the events $\mathcal{E}_2 \cap \mathcal{E}_3$, $\mathcal{E}_4$, and $\mathcal{E}_5$ holds with probability at least $1 - \delta/6$ in Lemma B.4, Lemma B.5, and Lemma B.6, respectively. Furthermore, as shown in Lemma B.7, the event $\mathcal{E}_6$ holds with probability at least $1 - \delta/3$. Since we already know the event $\mathcal{E}_1$ holds with probability at least $1 - \delta/6$ from Lemma 3.3, we can show $\mathbb{P}\left(\bigcap_{i \in [6]} \mathcal{E}_i\right) \geq 1 - \delta$ in this step by applying the union bound.

2. As with the proof of Salgia [30], the error term $|f(\boldsymbol{x}) - h(\boldsymbol{x}; \boldsymbol{\theta}_t)|$ is decomposed as follows:

$$
\begin{aligned}
&|f(\boldsymbol{x}) - h(\boldsymbol{x}; \boldsymbol{\theta}_t)| \\
&\leq |f(\boldsymbol{x}) - \tilde{\mu}_t(\boldsymbol{x})| + |\tilde{\mu}_t(\boldsymbol{x}) - \langle \boldsymbol{g}(\boldsymbol{x}; \boldsymbol{\theta}_0), \boldsymbol{\theta}_t - \boldsymbol{\theta}_0 \rangle| + |\langle \boldsymbol{g}(\boldsymbol{x}; \boldsymbol{\theta}_0), \boldsymbol{\theta}_t - \boldsymbol{\theta}_0 \rangle - h(\boldsymbol{x}_t; \boldsymbol{\theta}_t)|.
\end{aligned}
\tag{170}
$$

Based on the above decomposition, we derive the upper bound of each term under the event $\bigcap_{i \in [6]} \mathcal{E}_i$ with sufficiently large $M$. The first term of the above inequality is bounded from above by combining the event $\mathcal{E}_6$ with Lemma B.12. The second term is bounded by resorting to the arguments from [41], which is based on the optimization error of the gradient descent of the linearized squared loss (Lemma B.9 and Lemma B.11). The upper bound of the third term is obtained by combining the event $\mathcal{E}_1$ with the fact that the error of the first-order Taylor approximation can be characterized by the spectral norm of the Hessian.

### B.3.2 Lemmas for the events $\mathcal{E}_2$–$\mathcal{E}_6$

**Lemma B.4** (Gradient norm bounds). *Let $\delta \in (0, 1)$, $M \geq 3$, and $\alpha \geq 1$. Furthermore, let $\boldsymbol{\theta}_0$ be an initial parameter of ST-UCB. Then, with probability at least $1 - \delta$, for any $\boldsymbol{x} \in \mathbb{S}^{d-1}$, $R \geq 0$, and $\boldsymbol{\theta} \in \mathbb{R}^p$ such that $\|\boldsymbol{\theta} - \boldsymbol{\theta}_0\|_2 \leq R$, we have*

$$
\|\boldsymbol{g}(\boldsymbol{x}; \boldsymbol{\theta})\|_2^2 \leq C_{\alpha, \mathcal{D}}^{(4)}(2R + 2^{\mathcal{D}}\hat{C})^2 \ln \frac{M}{\delta},
\tag{171}
$$

$$
\|\boldsymbol{g}(\boldsymbol{x}; \boldsymbol{\theta}) - \boldsymbol{g}(\boldsymbol{x}; \boldsymbol{\theta}_0)\|_2^2 \leq \frac{C_{\alpha, \mathcal{D}}^{(5)}R^2}{M} \ln \frac{M}{\delta},
\tag{172}
$$

*where $\hat{C} > 0$ is an absolute constant. Moreover, $C_{\alpha, \mathcal{D}}^{(4)} = 2^{\mathcal{D}+2}\alpha^2$ and $C_{\alpha, \mathcal{D}}^{(5)} = 7 \cdot 2^{3\mathcal{D}}\hat{C}\alpha^2$.*

*Proof.* Suppose there exists $u \geq 1$ such that the following event holds:

$$
\forall m \in [M], \sum_{l=1}^{\mathcal{L}} |\bar{\pi}_l^{(m)}| \leq u.
\tag{173}
$$

Following the proof of Lemma 8 in [21], we can derive that:

$$
\|\boldsymbol{g}(\boldsymbol{x}; \boldsymbol{\theta})\|_2^2 \leq \mathcal{N}(R + u)^2\alpha^2 + \mathcal{L},
\tag{174}
$$

$$
\|\boldsymbol{g}(\boldsymbol{x}; \boldsymbol{\theta}) - \boldsymbol{g}(\boldsymbol{x}; \boldsymbol{\theta}_0)\|_2^2
\tag{175}
$$

$$
\leq \frac{1}{M} \sum_{m=1}^{M} \left[ \sum_{n=1}^{\mathcal{N}} \left( \alpha \sum_{l=1}^{\mathcal{L}} |\pi_l^{(m)} - \bar{\pi}_l^{(m)}| + 2\alpha u \sum_{\tilde{n}=1}^{\mathcal{N}} \|\boldsymbol{w}_{\tilde{n}}^{(m)} - \overline{\boldsymbol{w}}_{\tilde{n}}^{(m)}\|_2 \right)^2 \right.
\tag{176}
$$

$$
\left. + \sum_{l=1}^{\mathcal{L}} \left( \sum_{n=1}^{\mathcal{N}} \|\boldsymbol{w}_n^{(m)} - \overline{\boldsymbol{w}}_n^{(m)}\|_2 \right)^2 \right].
\tag{177}
$$

Furthermore, in the second inequality, we obtain the following upper bound from the Schwarz's inequality:

$$\frac{1}{M}\sum_{m=1}^{M}\left[\sum_{n=1}^{\mathcal{N}}\left(\alpha\sum_{l=1}^{\mathcal{L}}|\pi_l^{(m)}-\overline{\pi}_l^{(m)}|+2\alpha u\sum_{\tilde{n}=1}^{\mathcal{N}}\|\boldsymbol{w}_{\tilde{n}}^{(m)}-\overline{\boldsymbol{w}}_{\tilde{n}}^{(m)}\|_2\right)^2\right.$$
(178)

$$\left.+\sum_{l=1}^{\mathcal{L}}\left(\sum_{n=1}^{\mathcal{N}}\|\boldsymbol{w}_n^{(m)}-\overline{\boldsymbol{w}}_n^{(m)}\|_2\right)^2\right]$$
(179)

$$\leq\frac{1}{M}\sum_{m=1}^{M}\left[\sum_{n=1}^{\mathcal{N}}\left(2\alpha^2\mathcal{L}\|\boldsymbol{\pi}^{(m)}-\overline{\boldsymbol{\pi}}^{(m)}\|_2^2+4\alpha^2u^2\mathcal{N}^2\|\boldsymbol{w}_n^{(m)}-\overline{\boldsymbol{w}}_n^{(m)}\|_2^2\right)\right.$$
(180)

$$\left.+\sum_{l=1}^{\mathcal{L}}\mathcal{N}\|\boldsymbol{w}_n^{(m)}-\overline{\boldsymbol{w}}_n^{(m)}\|_2^2\right]$$
(181)

$$\leq\frac{1}{M}\sum_{m=1}^{M}\mathcal{N}\left(2\alpha^2\mathcal{L}+4\alpha^2u^2\mathcal{N}+\mathcal{L}\right)\|\boldsymbol{\theta}^{(m)}-\overline{\boldsymbol{\theta}}^{(m)}\|_2^2$$
(182)

$$=\frac{\mathcal{N}\left(2\alpha^2\mathcal{L}+4\alpha^2u^2\mathcal{N}+\mathcal{L}\right)}{M}\|\boldsymbol{\theta}-\overline{\boldsymbol{\theta}}\|_2^2$$
(183)

$$\leq\frac{7\alpha^2\mathcal{N}\mathcal{L}u^2}{M}R^2$$
(184)

Here, using the general Hoeffding's inequality (Lemma E.1) and the union bound, the event (173) holds with probability at least $1-\delta$ when $u=\sqrt{\hat{C}\mathcal{L}\ln(M/\delta)}$, where $\hat{C}\geq 1$ is an absolute constant. Therefore, with probability at least $1-\delta$:

$$\|\boldsymbol{g}(\boldsymbol{x};\boldsymbol{\theta})\|_2^2\leq\mathcal{N}\left(R+\sqrt{\hat{C}\mathcal{L}\ln\frac{M}{\delta}}\right)^2\alpha^2+\mathcal{L},$$

$$\|\boldsymbol{g}(\boldsymbol{x};\boldsymbol{\theta})-\boldsymbol{g}(\boldsymbol{x};\boldsymbol{\theta}_0)\|_2^2\leq\frac{C_{\alpha,\mathcal{D}}^{(5)}R^2}{M}\ln\frac{M}{\delta}.$$
(185)

Finally, from the definition of $C_{\alpha,\mathcal{D}}^{(4)}$,

$$2^{\mathcal{D}+2}(2R+\mathcal{L}\hat{C})^2\alpha^2=C_{\alpha,\mathcal{D}}^{(4)}(2R+\mathcal{L}\hat{C})^2$$
(186)

$$\Leftrightarrow 2^{\mathcal{D}+1}(2R+\mathcal{L}\hat{C})^2\alpha^2+2(2R+\mathcal{L}\hat{C})^2\alpha^2=C_{\alpha,\mathcal{D}}^{(4)}(2R+\mathcal{L}\hat{C})^2$$
(187)

$$\Rightarrow 2^{\mathcal{D}}(2R+\mathcal{L}\hat{C})^2\alpha^2+2\mathcal{L}\leq C_{\alpha,\mathcal{D}}^{(4)}(2R+\mathcal{L}\hat{C})^2$$
(188)

$$\Rightarrow \mathcal{N}\left(2R+\sqrt{\mathcal{L}\hat{C}}\right)^2\alpha^2+2\mathcal{L}\leq C_{\mathcal{D},\alpha}^{(4)}(2R+\mathcal{L}\hat{C})^2$$
(189)

$$\Rightarrow \mathcal{N}\left(\frac{R}{\sqrt{\ln 2}}+\sqrt{\mathcal{L}\hat{C}}\right)^2\alpha^2+\frac{\mathcal{L}}{\ln 2}\leq C_{\mathcal{D},\alpha}^{(4)}(2R+\mathcal{L}\hat{C})^2$$
(190)

$$\Rightarrow \mathcal{N}\left(\frac{R}{\sqrt{\ln(M/\delta)}}+\sqrt{\mathcal{L}\hat{C}}\right)^2\alpha^2+\frac{\mathcal{L}}{\ln(M/\delta)}\leq C_{\mathcal{D},\alpha}^{(4)}(2R+\mathcal{L}\hat{C})^2$$
(191)

$$\Leftrightarrow \mathcal{N}\left(R+\sqrt{\mathcal{L}\hat{C}\ln\frac{M}{\delta}}\right)^2\alpha^2+\mathcal{L}\leq C_{\mathcal{D},\alpha}^{(4)}(2R+\mathcal{L}\hat{C})^2\ln\frac{M}{\delta},$$
(192)

where:

- Eq. (188) follows from $(2R+\mathcal{L}\hat{C})^2\alpha^2\geq\mathcal{L}$ since $\alpha\geq 1$, $\hat{C}\geq 1$, and $R\geq 0$.

- Eq. (189) follows from $\mathcal{N}\leq 2^{\mathcal{D}}$ and $\mathcal{L}\hat{C}\geq 1$.

- Eq. (190) follows from $\sqrt{\ln 2} \geq \ln 2 \geq 0.5$.

- Eq. (191) follows from the fact that $\ln(M/\delta) \geq \ln 2$ holds under $M \geq 2$.

$\square$

**Lemma B.5.** *Fix any $\delta \in (0,1)$ and $f \in \mathcal{H}_{\mathrm{TNTK}}$ with $\|f\|_{\mathrm{TNTK}} \leq B$. Furthermore, suppose that $\epsilon_t$ is a $\sigma$-sub-Gauss random variable for any $t \in [T]$. Then, with probability at least $1 - \delta$, the following inequality holds for any $t \in [T]$:*

$$\|\boldsymbol{y}_t\|_2 \leq \left( \overline{k}B + \sigma\sqrt{2\ln\frac{2T}{\delta}} \right)\sqrt{t}, \tag{193}$$

*where $\overline{k} = \max_{\boldsymbol{x} \in \mathcal{X}} \sqrt{k_{\mathrm{TNTK}}(\boldsymbol{x}, \boldsymbol{x})}$.*

*Proof.* From the reproducing property of RKHS and Schwarz's inequality, for any $\boldsymbol{x} \in \mathcal{X}$, we have

$$f(\boldsymbol{x}) = \langle f, k_{\mathrm{TNTK}}(\boldsymbol{x}, \cdot)\rangle_{\mathcal{H}_{\mathrm{TNTK}}} \tag{194}$$
$$= \|f\|_{\mathrm{TNTK}}\|k_{\mathrm{TNTK}}(\boldsymbol{x}, \cdot)\|_{\mathrm{TNTK}} \tag{195}$$
$$= \|f\|_{\mathrm{TNTK}}\sqrt{k_{\mathrm{TNTK}}(\boldsymbol{x}, \boldsymbol{x})} \tag{196}$$
$$\leq B\overline{k}. \tag{197}$$

Thus,

$$\|\boldsymbol{y}_t\|_2^2 = \sum_{i=1}^{t}[f(\boldsymbol{x}_i) + \epsilon_i]^2 \tag{198}$$

$$\leq \sum_{i=1}^{t}\left(B\overline{k} + |\epsilon_i|\right)^2. \tag{199}$$

By using the concentration property of $\sigma$-sub-Gauss random variable, for any $t \in [T]$ and $\tilde{\delta} \in (0,1)$,

$$\mathbb{P}\left(|\epsilon_i| \leq \sigma\sqrt{2\ln\frac{2}{\tilde{\delta}}}\right) \geq 1 - \tilde{\delta}. \tag{200}$$

By setting $\tilde{\delta}$ as $\tilde{\delta} = \delta/T$ and taking the union bound, we complete the proof. $\square$

**Lemma B.6.** *Let $\delta \in (0,1)$, $\mathcal{D} \geq 2$, and $\mathcal{X} \subset \mathbb{S}^{d-1}$. Furthermore, let $\boldsymbol{K}_{\mathrm{TNTK}}(\mathcal{X}) := [k_{\mathrm{TNTK}}(\boldsymbol{x}, \tilde{\boldsymbol{x}})]_{\boldsymbol{x}, \tilde{\boldsymbol{x}} \in \mathcal{X}} \in \mathbb{R}^{|\mathcal{X}| \times |\mathcal{X}|}$ and $\lambda_0 = \lambda_{\min}(\boldsymbol{K}_{\mathrm{TNTK}}(\mathcal{X})) > 0$ be kernel matrix over $\mathcal{X} \times \mathcal{X}$ and the minimum eigenvalue of $\boldsymbol{K}_{\mathrm{TNTK}}(\mathcal{X})$, respectively. Moreover, assume that*

$$M \geq 64C_{\alpha,\mathcal{D}}^{(6)}|\mathcal{X}|^2\lambda_0^{-2}\ln\frac{16|\mathcal{X}|^2}{\delta} \quad \text{and} \quad M \geq C_{\alpha,\mathcal{D}}^{(6)}C_{\alpha,\mathcal{D}}^{(2)-2}\ln\frac{16|\mathcal{X}|^2}{\delta} \tag{201}$$

*hold, where $C_{\alpha,\mathcal{D}}^{(6)} = \tilde{C}\max\{C_{\alpha,\mathcal{D}}^{(2)2}, 2^{2\mathcal{D}}\}$. Here, $\tilde{C}$ and $C_{\alpha,\mathcal{D}}^{(2)}$ are defined in Lemma 3.2. Then, with probability at least $1 - \delta$, the following inequality holds for any $\boldsymbol{x}, \tilde{\boldsymbol{x}} \in \mathcal{X}$:*

$$|k_{\mathrm{TNTK}}(\boldsymbol{x}, \tilde{\boldsymbol{x}}) - \tilde{k}(\boldsymbol{x}, \tilde{\boldsymbol{x}})| \leq \min\left\{ \frac{\lambda_0}{2|\mathcal{X}|}, \sqrt{\frac{4C_{\alpha,\mathcal{D}}^{(6)}}{M}\ln\frac{16|\mathcal{X}|^2}{\delta}}, 4C_{\alpha,\mathcal{D}}^{(2)} \right\}, \tag{202}$$

*where $\tilde{k}(\boldsymbol{x}, \tilde{\boldsymbol{x}}) = \langle \boldsymbol{g}(\boldsymbol{x}; \boldsymbol{\theta}_0), \boldsymbol{g}(\tilde{\boldsymbol{x}}; \boldsymbol{\theta}_0)\rangle$.*

*Proof.* From Lemma 3.2 and the union bound, for any $\varepsilon \in (0, C_{\alpha,\mathcal{D}}^{(2)})$, we have

$$M \geq C_{\alpha,\mathcal{D}}^{(6)}\varepsilon^{-2}\ln\frac{16|\mathcal{X}|^2}{\delta}$$
$$\Rightarrow \mathbb{P}(\forall \boldsymbol{x}, \tilde{\boldsymbol{x}} \in \mathcal{X}, |k_{\mathrm{TNTK}}(\boldsymbol{x}, \tilde{\boldsymbol{x}}) - \langle \boldsymbol{g}(\boldsymbol{x}, \boldsymbol{\theta}_0), \boldsymbol{g}(\tilde{\boldsymbol{x}}, \boldsymbol{\theta}_0)\rangle| \leq 4\varepsilon) \geq 1 - \delta. \tag{203}$$

Here, we set $\varepsilon$ as $\varepsilon = \min\{\lambda_0/(8|\mathcal{X}|), \sqrt{C_{\alpha,\mathcal{D}}^{(6)} \ln(16|\mathcal{X}|^2/\delta)/M}, C_{\alpha,\mathcal{D}}^{(2)}\}$; then, $\varepsilon \in (0, C_{\alpha,\mathcal{D}}^{(2)})$. Therefore, by using Eq. (203), we have

$$M \geq C_{\alpha,\mathcal{D}}^{(6)} \min\left\{\frac{\lambda_0}{8|\mathcal{X}|}, \sqrt{\frac{C_{\alpha,\mathcal{D}}^{(6)}}{M} \ln\frac{16|\mathcal{X}|^2}{\delta}}, C_{\alpha,\mathcal{D}}^{(2)}\right\}^{-2} \ln\frac{16|\mathcal{X}|^2}{\delta} \tag{204}$$

$$\Rightarrow \mathbb{P}\left(\forall \boldsymbol{x}, \tilde{\boldsymbol{x}} \in \mathcal{X}, |k_{\text{TNTK}}(\boldsymbol{x}, \tilde{\boldsymbol{x}}) - \tilde{k}(\boldsymbol{x}, \tilde{\boldsymbol{x}})| \leq \min\left\{\frac{\lambda_0}{2|\mathcal{X}|}, \sqrt{\frac{4C_{\alpha,\mathcal{D}}^{(6)}}{M} \ln\frac{16|\mathcal{X}|^2}{\delta}}, 4C_{\alpha,\mathcal{D}}^{(2)}\right\}\right)$$

$$\geq 1 - \delta. \tag{205}$$

Furthermore,

$$M \geq 64 C_{\alpha,\mathcal{D}}^{(6)} |\mathcal{X}|^2 \lambda_0^{-2} \ln\frac{16|\mathcal{X}|^2}{\delta} \text{ and } M \geq C_{\alpha,\mathcal{D}}^{(6)} C_{\alpha,\mathcal{D}}^{(2)-2} \ln\frac{16|\mathcal{X}|^2}{\delta}$$

$$\Rightarrow M \geq C_{\alpha,\mathcal{D}}^{(6)} \min\left\{\frac{\lambda_0}{8|\mathcal{X}|}, \sqrt{\frac{C_{\alpha,\mathcal{D}}^{(6)}}{M} \ln\frac{16|\mathcal{X}|^2}{\delta}}, C_{\alpha,\mathcal{D}}^{(2)}\right\}^{-2} \ln\frac{16|\mathcal{X}|^2}{\delta}. \tag{206}$$

By combining the above implication with Eq. (203), we complete the proof. $\qquad\square$

**Lemma B.7.** *Fix any $\delta \in (0,1)$ and $f \in \mathcal{H}_{\text{TNTK}}$ with $\|f\|_{\text{TNTK}} \leq B$. Let us define $\tilde{k}$ as $\tilde{k}(\boldsymbol{x}, \tilde{\boldsymbol{x}}) = \langle \boldsymbol{g}(\boldsymbol{x}; \boldsymbol{\theta}_0), \boldsymbol{g}(\tilde{\boldsymbol{x}}; \boldsymbol{\theta}_0)\rangle$. Furthermore, suppose that $(\epsilon_t)_{t \in \mathbb{N}_+}$ are conditionally $\sigma$-sub-Gaussian random variables. Then, under the event $\mathcal{E}_5$, with probability at least $1 - \delta$,*

$$\forall t \in \mathbb{N}_+, \forall \boldsymbol{x} \in \mathcal{X}, |f(\boldsymbol{x}) - \tilde{\mu}_{t-1}(\boldsymbol{x})| \leq \left(\sqrt{2}B + \frac{\sigma}{\sqrt{\rho}}\sqrt{2\left(\tilde{\gamma}_t + \ln\frac{1}{\delta}\right)}\right)\tilde{\sigma}_{t-1}(\boldsymbol{x}). \tag{207}$$

*Here, we respectively define $\tilde{\mu}_{t-1}(\boldsymbol{x})$ and $\tilde{\gamma}_t$ as*

$$\tilde{\mu}_t(\boldsymbol{x}) = \tilde{\boldsymbol{k}}_t^\top(\boldsymbol{x})\left(\tilde{\boldsymbol{K}}_t + \rho\boldsymbol{I}_t\right)^{-1}\boldsymbol{y}_t, \tag{208}$$

$$\tilde{\gamma}_t = \frac{1}{2}\max_{\boldsymbol{x}_1,\ldots,\boldsymbol{x}_t} \ln\det\left(\boldsymbol{I}_t + \rho^{-1}\tilde{\boldsymbol{K}}_t\right), \tag{209}$$

*where $\tilde{\boldsymbol{k}}_t(\boldsymbol{x}) = [\tilde{k}(\boldsymbol{x}, \boldsymbol{x}_i)]_{i \in [t]} \in \mathbb{R}^t$ and $\tilde{\boldsymbol{K}}_t = [\tilde{k}(\boldsymbol{x}_i, \boldsymbol{x}_j)]_{i,j \in [t]} \in \mathbb{R}^{t \times t}$ with $\tilde{k}(\boldsymbol{x}, \tilde{\boldsymbol{x}}) = \langle \boldsymbol{g}(\boldsymbol{x}; \boldsymbol{\theta}_0), \boldsymbol{g}(\tilde{\boldsymbol{x}}; \boldsymbol{\theta}_0)\rangle$.*

*Proof.* From the definition of $\mathcal{E}_5$, we have $|k_{\text{TNTK}}(\boldsymbol{x}, \tilde{\boldsymbol{x}}) - \langle \boldsymbol{g}(\boldsymbol{x}, \boldsymbol{\theta}_0), \boldsymbol{g}(\tilde{\boldsymbol{x}}, \boldsymbol{\theta}_0)\rangle| \leq \lambda_0/(2|\mathcal{X}|)$ for any $\boldsymbol{x}, \tilde{\boldsymbol{x}} \in \mathcal{X}$. Therefore, $\sqrt{\sum_{\boldsymbol{x}, \tilde{\boldsymbol{x}} \in \mathcal{X}} |k_{\text{TNTK}}(\boldsymbol{x}, \tilde{\boldsymbol{x}}) - \langle \boldsymbol{g}(\boldsymbol{x}, \boldsymbol{\theta}_0), \boldsymbol{g}(\tilde{\boldsymbol{x}}, \boldsymbol{\theta}_0)\rangle|^2} \leq \lambda_0/2$. Here, by combining this inequality with the arguments of the proof of Lemma C.5 in [24], under the event $\mathcal{E}_5$, we have $f \in \mathcal{H}_{\tilde{k}}$ with $\|f\|_{\tilde{k}} \leq \sqrt{2}B$. Therefore, since $\tilde{\mu}_t$ and $\tilde{\sigma}_t$ are defined as the posterior mean and the posterior variance of Gaussian process characterized by the kernel function $\tilde{k}$, we obtain the desired result by applying Lemma 3.11 in [2]. $\qquad\square$

**Lemma B.8.** *Fix any $\delta \in (0,1)$; then, $\mathbb{P}(\cap_{i \in [6]}\mathcal{E}_i) \geq 1 - \delta$ holds.*

*Proof.* From Lemma 3.3, B.5, and B.6, we have $\mathbb{P}(\mathcal{E}_i^c) \leq \delta/6$ for any $i \in [5]/\{2,3\}$. In addition, from Lemma B.4, we have $\mathbb{P}(\mathcal{E}_2^c \cup \mathcal{E}_3^c) \leq \delta/6$. Here, from Lemma B.6 and Lemma B.7, we have

$$\mathbb{P}(\mathcal{E}_6^c) = \mathbb{P}(\mathcal{E}_6^c \mid \mathcal{E}_5)\mathbb{P}(\mathcal{E}_5) + \mathbb{P}(\mathcal{E}_6^c \mid \mathcal{E}_5^c)\mathbb{P}(\mathcal{E}_5^c) \tag{210}$$

$$\leq \frac{\delta}{6} + \frac{\delta}{6} \tag{211}$$

$$= \frac{\delta}{3}. \tag{212}$$

Therefore, by taking the union bound, we have

$$\mathbb{P}(\cap_{i \in [6]} \mathcal{E}_i) = 1 - \mathbb{P}(\cup_{i \in [6]} \mathcal{E}_i^c) \tag{213}$$

$$\geq 1 - [\mathbb{P}(\mathcal{E}_1^c) + \mathbb{P}(\mathcal{E}_2^c \cup \mathcal{E}_3^c) + \mathbb{P}(\mathcal{E}_4^c) + \mathbb{P}(\mathcal{E}_5^c) + \mathbb{P}(\mathcal{E}_6^c)] \tag{214}$$

$$\geq 1 - \delta. \tag{215}$$

$\square$

## B.4 Lemmas for the upper bounds of Eq. (170)

**Definition B.1.** *Define $\tilde{L}_t(\boldsymbol{\theta})$ for any $t \in \mathbb{N}_+$:*

$$\tilde{L}_t(\boldsymbol{\theta}) = \left\| \boldsymbol{G}_t^\top (\boldsymbol{\theta} - \boldsymbol{\theta}_0) - \boldsymbol{y}_t \right\|_2^2 + \rho \left\| \boldsymbol{\theta} - \boldsymbol{\theta}_0 \right\|_2^2. \tag{216}$$

*Furthermore, let us define $\tilde{\boldsymbol{\theta}}_{t;1}, \ldots, \tilde{\boldsymbol{\theta}}_{t;J}$ as*

$$\tilde{\boldsymbol{\theta}}_{t;j} = \tilde{\boldsymbol{\theta}}_{t;j-1} - \eta \left\{ 2\boldsymbol{G}_t \left[ \boldsymbol{G}_t^\top \left( \tilde{\boldsymbol{\theta}}_{t;j-1} - \boldsymbol{\theta}_0 \right) - \boldsymbol{y}_t \right] + 2\rho \left( \tilde{\boldsymbol{\theta}}_{t;j-1} - \boldsymbol{\theta}_0 \right) \right\}, \tag{217}$$

*where $\tilde{\boldsymbol{\theta}}_{t;0} = \boldsymbol{\theta}_0$.*

**Lemma B.9** (Adapted from Lemma C.4 in [41])**.** *Suppose that the events $\mathcal{E}_2$ and $\mathcal{E}_4$ simultaneously hold. Furthermore, assume that $\eta \leq 2^{-1} \left( T\hat{C}^2 2^{2\mathcal{D}} C_{\alpha,\mathcal{D}}^{(4)} \ln(6M/\delta) + \rho \right)^{-1}$ holds. Then, the following inequalities hold for any $t \in [T]$ and $j \in [J]$:*

$$\left\| \tilde{\boldsymbol{\theta}}_{t;j} - \boldsymbol{\theta}_0 \right\|_2 \leq \left( \overline{k}B + \sigma \sqrt{2 \ln \frac{12T}{\delta}} \right) \sqrt{\frac{t}{\rho}}, \tag{218}$$

$$\left\| \tilde{\boldsymbol{\theta}}_{t;j} - \boldsymbol{\theta}_0 - \left( \rho \boldsymbol{I}_p + \boldsymbol{G}_t \boldsymbol{G}_t^\top \right)^{-1} \boldsymbol{G}_t \boldsymbol{y}_t \right\|_2 \leq (1 - 2\eta\rho)^{j/2} \left( \overline{k}B + \sigma \sqrt{2 \ln \frac{12T}{\delta}} \right) \sqrt{\frac{t}{\rho}}, \tag{219}$$

*where $\overline{k}$ is defined in Lemma B.5. Furthermore, the constants $\hat{C}$ and $C_{\alpha,\mathcal{D}}^{(4)}$ are defined in Lemma B.4.*

*Proof.* From the definition of $\tilde{L}_t(\boldsymbol{\theta})$, we have

$$\nabla_{\boldsymbol{\theta}}^2 \tilde{L}_t(\boldsymbol{\theta}) = 2\boldsymbol{G}_t \boldsymbol{G}_t^\top + 2\rho \boldsymbol{I}_p \tag{220}$$

$$\preceq 2 \left( \|\boldsymbol{G}_t\|_F^2 + \rho \right) \boldsymbol{I}_p \tag{221}$$

$$\preceq 2 \left( t\hat{C}^2 2^{2\mathcal{D}} C_{\alpha,\mathcal{D}}^{(4)} \ln \frac{6M}{\delta} + \rho \right) \boldsymbol{I}_p, \tag{222}$$

where Eq. (222) follows from Lemma B.13. Therefore, $\tilde{L}_t(\boldsymbol{\theta})$ is $2 \left( t\hat{C}^2 2^{2\mathcal{D}} C_{\alpha,\mathcal{D}}^{(4)} \ln \frac{6M}{\delta} + \rho \right)$-smooth function. Furthermore, $\tilde{L}_t(\boldsymbol{\theta})$ is $2\rho$-strong convex because $\nabla_{\boldsymbol{\theta}}^2 \tilde{L}_t(\boldsymbol{\theta}) \succeq 2\rho \boldsymbol{I}_p$ holds. By combining the definition of $\eta$ with the standard result of gradient descent for the strongly convex and smooth objective function (e.g., Theorem 3.6 in [16]), $\tilde{L}_t(\tilde{\boldsymbol{\theta}}_{t;j}) \geq \tilde{L}_t(\tilde{\boldsymbol{\theta}}_{t;j-1})$ holds for any $j \in [J]$. Therefore,

$$\rho \left\| \tilde{\boldsymbol{\theta}}_{t;J} - \boldsymbol{\theta}_0 \right\|_2^2 \leq \left\| \boldsymbol{G}_t^\top \left( \tilde{\boldsymbol{\theta}}_{t;J} - \boldsymbol{\theta}_0 \right) - \boldsymbol{y}_t \right\|_2^2 + \rho \left\| \tilde{\boldsymbol{\theta}}_{t;J} - \boldsymbol{\theta}_0 \right\|_2^2 \tag{223}$$

$$\leq \left\| \boldsymbol{G}_t^\top \left( \tilde{\boldsymbol{\theta}}_{t;0} - \boldsymbol{\theta}_0 \right) - \boldsymbol{y}_t \right\|_2^2 + \rho \left\| \tilde{\boldsymbol{\theta}}_{t;0} - \boldsymbol{\theta}_0 \right\|_2^2 \tag{224}$$

$$\leq \|\boldsymbol{y}_t\|_2^2 \tag{225}$$

$$\leq \left( \overline{k}B + \sigma \sqrt{2 \ln \frac{12T}{\delta}} \right)^2 t, \tag{226}$$

where Eq. (226) follows from the event $\mathcal{E}_4$. Furthermore, since the unique minimum of $\tilde{L}_t(\boldsymbol{\theta})$ is given as $\boldsymbol{\theta}^* := \boldsymbol{\theta}_0 + \left( \rho \boldsymbol{I}_p + \boldsymbol{G}_t \boldsymbol{G}_t^\top \right)^{-1} \boldsymbol{G}_t \boldsymbol{y}_t$, we have the following inequalities from Theorem 3.6

in [16]:

$$\left\| \tilde{\boldsymbol{\theta}}_{t;j} - \boldsymbol{\theta}_0 - \left(\rho \boldsymbol{I}_p + \boldsymbol{G}_t \boldsymbol{G}_t^\top\right)^{-1} \boldsymbol{G}_t \boldsymbol{y}_t \right\|_2^2 \tag{227}$$

$$\leq (1 - 2\eta\rho)^j \left\| \boldsymbol{\theta}_0 - \boldsymbol{\theta}^* \right\|_2^2 \tag{228}$$

$$\leq (1 - 2\eta\rho)^j \frac{1}{\rho} \left[ \tilde{L}_t(\boldsymbol{\theta}_0) - \tilde{L}_t(\boldsymbol{\theta}^*) \right] \tag{229}$$

$$\leq (1 - 2\eta\rho)^j \frac{\tilde{L}_t(\boldsymbol{\theta}_0)}{\rho} \tag{230}$$

$$\leq (1 - 2\eta\rho)^j \frac{\|\boldsymbol{y}_t\|_2^2}{\rho} \tag{231}$$

$$\leq (1 - 2\eta\rho)^j \frac{\left(\overline{k}B + \sigma\sqrt{2\ln\frac{12T}{\delta}}\right)^2 t}{\rho}, \tag{232}$$

where:

- Eq. (229) follows from $\nabla \tilde{L}_t(\boldsymbol{\theta}^*) = \boldsymbol{0}$ and the fact that $\tilde{L}_t$ is the $2\rho$-strong convex function.
- Eq. (230) follows from $\tilde{L}_t(\boldsymbol{\theta}^*) \geq 0$.
- Eq. (232) follows from the event $\mathcal{E}_4$.

$\square$

**Lemma B.10** (Adapted from Lemma C.3 in [41]). *Fix any $R \geq 0$, $j \in [J]$ and $t \in [T]$. Suppose that the learning rate $\eta$ satisfies $\eta \leq 4^{-1}\left(\rho + 2(2R + 2^{\mathcal{D}}\hat{C})^2 T C_{\alpha,\mathcal{D}}^{(4)} \ln\frac{6M}{\delta}\right)^{-1}$, and $\boldsymbol{\theta}_{t;\tilde{j}}$ satisfies $\|\boldsymbol{\theta}_{t;\tilde{j}} - \boldsymbol{\theta}_0\|_2 \leq R$ for all $\tilde{j} \in [j]$. Furthermore, assume that $R$ satisfies the following inequality:*

$$\overline{R}^4(\overline{R} + 2)^4 \leq \frac{3\eta^2 M \rho^2}{56 C_{\alpha,\mathcal{D}}^{(3)2}} \left(\overline{k}B + \sigma\sqrt{2\ln\frac{12T}{\delta}}\right)^2 \left(\ln\frac{6 \cdot 2^{\mathcal{D}+2}M}{\delta}\right)^{-2}, \tag{233}$$

*where:*

$$\overline{R} = R + \frac{1}{2\rho}\left[(2R + 2^{\mathcal{D}})\sqrt{tC_{\alpha,\mathcal{D}}^{(4)}\ln\frac{6M}{\delta}}\left(\overline{k}B + \sigma\sqrt{2\ln\frac{12T}{\delta}}\right)\sqrt{3t} + R\right]. \tag{234}$$

*Then, under the events $\mathcal{E}_2$ and $\mathcal{E}_4$,*

$$\|\boldsymbol{h}_{t;j+1} - \boldsymbol{y}_t\|_2 \leq \left(\overline{k}B + \sigma\sqrt{2\ln\frac{12T}{\delta}}\right)\sqrt{3t}, \tag{235}$$

*where $\boldsymbol{h}_{t;j} = [h(\boldsymbol{x}_1; \boldsymbol{\theta}_{t;j}), \ldots, h(\boldsymbol{x}_t; \boldsymbol{\theta}_{t;j})]^\top \in \mathbb{R}^t$.*

*Proof.* We first assume that $\|\boldsymbol{h}_{t;j} - \boldsymbol{y}_t\| \leq \left(\overline{k}B + \sigma\sqrt{2\ln\frac{12T}{\delta}}\right)\sqrt{3t}$ holds. Here, by resorting the same arguments as the proof of Lemma C.3 in [41], for any $\boldsymbol{\theta}, \boldsymbol{\theta}' \in \mathbb{R}^d$ such that $\|\boldsymbol{\theta} - \boldsymbol{\theta}_0\| \leq R$, we have

$$- \frac{\|\nabla_{\boldsymbol{\theta}} L_t(\boldsymbol{\theta})\|_2^2}{\rho} - 2\|\boldsymbol{h}_t(\boldsymbol{\theta}) - \boldsymbol{y}_t\|_2 \|\boldsymbol{e}(\boldsymbol{\theta}', \boldsymbol{\theta})\|_2 \tag{236}$$

$$\leq L_t(\boldsymbol{\theta}') - L_t(\boldsymbol{\theta}) \tag{237}$$

$$\leq 2\langle \nabla_{\boldsymbol{\theta}} L_t(\boldsymbol{\theta}), \boldsymbol{\theta}' - \boldsymbol{\theta} \rangle + 2\|\boldsymbol{h}_t(\boldsymbol{\theta}) - \boldsymbol{y}_t\|_2 \|\boldsymbol{e}(\boldsymbol{\theta}', \boldsymbol{\theta})\|_2$$

$$+ 2\left[(2R + 2^{\mathcal{D}}\hat{C})\sqrt{tC_{\alpha,\mathcal{D}}^{(4)}\ln\frac{6M}{\delta}}\right]^2 \|\boldsymbol{\theta}' - \boldsymbol{\theta}\|_2^2 + 2\|\boldsymbol{e}(\boldsymbol{\theta}', \boldsymbol{\theta})\|_2^2 + \rho\|\boldsymbol{\theta}' - \boldsymbol{\theta}\|_2^2, \tag{238}$$

where $e(\boldsymbol{\theta}', \boldsymbol{\theta}) = \boldsymbol{h}_t(\boldsymbol{\theta}') - \boldsymbol{h}_t(\boldsymbol{\theta}) - \boldsymbol{G}_t(\boldsymbol{\theta})^\top (\boldsymbol{\theta}' - \boldsymbol{\theta})$ with $\boldsymbol{h}_t(\boldsymbol{\theta}) = (h(\boldsymbol{x}_1; \boldsymbol{\theta}), \dots, h(\boldsymbol{x}_t; \boldsymbol{\theta}))^\top \in \mathbb{R}^t$ and $\boldsymbol{G}_t(\boldsymbol{\theta}) = (\boldsymbol{g}(\boldsymbol{x}_1; \boldsymbol{\theta}), \dots, \boldsymbol{g}(\boldsymbol{x}_t; \boldsymbol{\theta}))^\top \in \mathbb{R}^{p \times t}$. From the upper bound of $L_t(\boldsymbol{\theta}') - L_t(\boldsymbol{\theta})$, by setting $\boldsymbol{\theta}' \in \mathbb{R}^p$ as $\boldsymbol{\theta}' = \boldsymbol{\theta} - \eta \nabla_{\boldsymbol{\theta}} L_t(\boldsymbol{\theta})$, we have

$$L_t(\boldsymbol{\theta} - \eta \nabla_{\boldsymbol{\theta}} L_t(\boldsymbol{\theta})) - L_t(\boldsymbol{\theta}) \tag{239}$$

$$\leq -2c\eta \|\nabla_{\boldsymbol{\theta}} L_t(\boldsymbol{\theta})\|_2^2$$
$$+ 2\|\boldsymbol{h}_t(\boldsymbol{\theta}) - \boldsymbol{y}_t\|_2 \|e(\boldsymbol{\theta} - \eta \nabla_{\boldsymbol{\theta}} L_t(\boldsymbol{\theta}), \boldsymbol{\theta})\|_2 + 2\|e(\boldsymbol{\theta} - \eta \nabla_{\boldsymbol{\theta}} L_t(\boldsymbol{\theta}), \boldsymbol{\theta})\|_2^2, \tag{240}$$

where $c = \left\{ 1 - \eta \left[ \rho + 2(2R + 2^{\mathcal{D}} \hat{C})^2 t C_{\alpha, \mathcal{D}}^{(4)} \ln \frac{6M}{\delta} \right] \right\} \in (0, 1)$. Furthermore, for any $\boldsymbol{\theta}' \in \mathbb{R}^p$, we obtain the following inequality by combining the lower bound of $L_t(\boldsymbol{\theta}') - L_t(\boldsymbol{\theta})$ with the above inequality,

$$L_t(\boldsymbol{\theta} - \eta \nabla_{\boldsymbol{\theta}} L_t(\boldsymbol{\theta})) - L_t(\boldsymbol{\theta}) \tag{241}$$

$$\leq 2c\eta\rho \left[ L_t(\boldsymbol{\theta}') - L_t(\boldsymbol{\theta}) + 2\|\boldsymbol{h}_t(\boldsymbol{\theta}) - \boldsymbol{y}_t\|_2 \|e(\boldsymbol{\theta}', \boldsymbol{\theta})\|_2 \right]$$
$$+ 2\|\boldsymbol{h}_t(\boldsymbol{\theta}) - \boldsymbol{y}_t\|_2 \|e(\boldsymbol{\theta} - \eta \nabla_{\boldsymbol{\theta}} L_t(\boldsymbol{\theta}), \boldsymbol{\theta})\|_2 + 2\|e(\boldsymbol{\theta} - \eta \nabla_{\boldsymbol{\theta}} L_t(\boldsymbol{\theta}), \boldsymbol{\theta})\|_2^2 \tag{242}$$

$$\leq 2c\eta\rho \left[ L_t(\boldsymbol{\theta}') - L_t(\boldsymbol{\theta}) + \frac{1}{4}\|\boldsymbol{h}_t(\boldsymbol{\theta}) - \boldsymbol{y}_t\|_2^2 + 4\|e(\boldsymbol{\theta}', \boldsymbol{\theta})\|_2^2 \right]$$
$$+ 2c\eta\rho \frac{1}{4}\|\boldsymbol{h}_t(\boldsymbol{\theta}) - \boldsymbol{y}_t\|_2^2 + \frac{4}{2c\eta\rho}\|e(\boldsymbol{\theta} - \eta \nabla_{\boldsymbol{\theta}} L_t(\boldsymbol{\theta}), \boldsymbol{\theta})\|_2^2 + 2\|e(\boldsymbol{\theta} - \eta \nabla_{\boldsymbol{\theta}} L_t(\boldsymbol{\theta}), \boldsymbol{\theta})\|_2^2 \tag{243}$$

$$\leq 2c\eta\rho \left[ L_t(\boldsymbol{\theta}') - \frac{1}{2} L_t(\boldsymbol{\theta}) \right] + 8c\eta\rho \|e(\boldsymbol{\theta}', \boldsymbol{\theta})\|_2^2$$
$$+ \frac{2}{c\eta\rho}\|e(\boldsymbol{\theta} - \eta \nabla_{\boldsymbol{\theta}} L_t(\boldsymbol{\theta}), \boldsymbol{\theta})\|_2^2 + 2\|e(\boldsymbol{\theta} - \eta \nabla_{\boldsymbol{\theta}} L_t(\boldsymbol{\theta}), \boldsymbol{\theta})\|_2^2, \tag{244}$$

where the second inequality follows from the Peter-Paul inequality, and the last inequality follows from $\|\boldsymbol{h}_t(\boldsymbol{\theta}) - \boldsymbol{y}_t\|_2^2 \leq L_t(\boldsymbol{\theta})$. Rearranging the above inequality with $\boldsymbol{\theta} = \boldsymbol{\theta}_{t;j}$ and $\boldsymbol{\theta}' = \boldsymbol{\theta}_0$, we have

$$L_t(\boldsymbol{\theta}_{t;j+1}) - L_t(\boldsymbol{\theta}_0) \tag{245}$$

$$\leq (1 - c\eta\rho) \left[ L_t(\boldsymbol{\theta}_{t;j}) - L_t(\boldsymbol{\theta}_0) \right] + c\eta\rho L_t(\boldsymbol{\theta}_0)$$
$$+ 8c\eta\rho \|e(\boldsymbol{\theta}_0, \boldsymbol{\theta}_{t;j})\|_2^2 + \frac{2}{c\eta\rho}\|e(\boldsymbol{\theta}_{t;j+1}, \boldsymbol{\theta}_{t;j})\|_2^2 + 2\|e(\boldsymbol{\theta}_{t;j+1}, \boldsymbol{\theta}_{t;j})\|_2^2 \tag{246}$$

$$\leq (1 - c\eta\rho) \left[ L_t(\boldsymbol{\theta}_{t;j}) - L_t(\boldsymbol{\theta}_0) \right] + c\eta\rho \left( \overline{k}B + \sigma\sqrt{2\ln\frac{12T}{\delta}} \right)^2 t$$
$$+ 8c\eta\rho \|e(\boldsymbol{\theta}_0, \boldsymbol{\theta}_{t;j})\|_2^2 + \frac{2}{c\eta\rho}\|e(\boldsymbol{\theta}_{t;j+1}, \boldsymbol{\theta}_{t;j})\|_2^2 + 2\|e(\boldsymbol{\theta}_{t;j+1}, \boldsymbol{\theta}_{t;j})\|_2^2. \tag{247}$$

Then, from Lemma B.15,

$$\|e(\boldsymbol{\theta}_0, \boldsymbol{\theta}_{t;j})\|_2^2 = \left\| \boldsymbol{h}_t(\boldsymbol{\theta}_0) - \boldsymbol{h}_t(\boldsymbol{\theta}_{t;j}) - \boldsymbol{G}_t(\boldsymbol{\theta}_{t;j})^\top (\boldsymbol{\theta}_0 - \boldsymbol{\theta}_{t;j}) \right\|_2^2 \tag{248}$$

$$= \sum_{i=1}^{t} \left( h(\boldsymbol{x}_i; \boldsymbol{\theta}_0) - h(\boldsymbol{x}_i; \boldsymbol{\theta}_{t;j}) - \langle \boldsymbol{g}(\boldsymbol{x}_i; \boldsymbol{\theta}_{t;j}), \boldsymbol{\theta}_0 - \boldsymbol{\theta}_{t;j} \rangle \right)^2 \tag{249}$$

$$\leq \frac{4tR^4(R+2)^4 C_{\alpha,\mathcal{D}}^{(3)2}}{M} \left( \ln \frac{6 \cdot 2^{\mathcal{D}+2} M}{\delta} \right)^2 \tag{250}$$

$$\leq \frac{4t\overline{R}^4(\overline{R}+2)^4 C_{\alpha,\mathcal{D}}^{(3)2}}{M} \left( \ln \frac{6 \cdot 2^{\mathcal{D}+2} M}{\delta} \right)^2. \tag{251}$$

Furthermore, from $\|\boldsymbol{h}_{t;j} - \boldsymbol{y}_t\|_2 \le \left(\overline{k}B + \sigma\sqrt{2\ln\frac{12T}{\delta}}\right)\sqrt{3t}$, we have

$$\|\boldsymbol{\theta}_{t;j+1} - \boldsymbol{\theta}_{t;j}\| = \eta\|\nabla_{\boldsymbol{\theta}_{t;j}}L_t(\boldsymbol{\theta}_{t;j})\|_2 \tag{252}$$

$$\le \frac{1}{4\rho}\|2\boldsymbol{G}_{t;j}(\boldsymbol{h}_{t;j} - \boldsymbol{y}_t) + 2(\boldsymbol{\theta}_{t;j} - \boldsymbol{\theta}_0)\|_2 \tag{253}$$

$$\le \frac{1}{2\rho}\left(\|\boldsymbol{G}_{t;j}\|\|\boldsymbol{h}_{t;j} - \boldsymbol{y}_t\|_2 + \|\boldsymbol{\theta}_{t;j} - \boldsymbol{\theta}_0\|_2\right) \tag{254}$$

$$\le \frac{1}{2\rho}\left[(2R + 2^{\mathcal{D}})\sqrt{tC_{\alpha,\mathcal{D}}^{(4)}\ln\frac{6M}{\delta}}\left(\overline{k}B + \sigma\sqrt{2\ln\frac{12T}{\delta}}\right)\sqrt{3t} + R\right] \tag{255}$$

$$\Rightarrow \|\boldsymbol{\theta}_{t;j+1} - \boldsymbol{\theta}_0\| \le \overline{R}. \tag{256}$$

Combining the above inequality with Lemma B.15, we have

$$\|\boldsymbol{e}(\boldsymbol{\theta}_{t;j+1}, \boldsymbol{\theta}_{t;j})\|_2^2 \le \frac{4t\overline{R}^4(\overline{R}+2)^4 C_{\alpha,\mathcal{D}}^{(3)2}}{M}\left(\ln\frac{6\cdot 2^{\mathcal{D}+2}M}{\delta}\right)^2. \tag{257}$$

Therefore,

$$8c\eta\rho\|\boldsymbol{e}(\boldsymbol{\theta}_0, \boldsymbol{\theta}_{t;j})\|_2^2 + \frac{2}{c\eta\rho}\|\boldsymbol{e}(\boldsymbol{\theta}_{t;j+1}, \boldsymbol{\theta}_{t;j})\|_2^2 + 2\|\boldsymbol{e}(\boldsymbol{\theta}_{t;j+1}, \boldsymbol{\theta}_{t;j})\|_2^2 \tag{258}$$

$$\le \left(8c\eta\rho + \frac{2}{c\eta\rho} + 2\right)\frac{4t\overline{R}^4(\overline{R}+2)^4 C_{\alpha,\mathcal{D}}^{(3)2}}{M}\left(\ln\frac{6\cdot 2^{\mathcal{D}+2}M}{\delta}\right)^2 \tag{259}$$

$$\le \frac{3}{c\eta\rho}\frac{4t\overline{R}^4(\overline{R}+2)^4 C_{\alpha,\mathcal{D}}^{(3)2}}{M}\left(\ln\frac{6\cdot 2^{\mathcal{D}+2}M}{\delta}\right)^2 \tag{260}$$

$$\le c\eta\rho\left(\overline{k}B + \sigma\sqrt{2\ln\frac{12T}{\delta}}\right)^2 t, \tag{261}$$

where the second line follows from the fact that $8c\eta\rho + 2 \le 1/(c\eta\rho)$ holds due to $\eta\rho \le 1/4$ and $c \in (0,1)$. Furthermore, the last line follows from the condition (233) with $c \ge 3/4$. By combining Eq. (245) with Eq. (261), we have

$$\|\boldsymbol{h}_{t;j+1} - \boldsymbol{y}_t\|_2^2 - \|\boldsymbol{y}_t\|_2^2 \tag{262}$$
$$= L_t(\boldsymbol{\theta}_{t;j+1}) - L_t(\boldsymbol{\theta}_0) \tag{263}$$

$$\le (1 - c\eta\rho)[L_t(\boldsymbol{\theta}_{t;j}) - L_t(\boldsymbol{\theta}_0)] + 2c\eta\rho\left(\overline{k}B + \sigma\sqrt{2\ln\frac{12T}{\delta}}\right)^2 t \tag{264}$$

$$\le \frac{2c\eta\rho\left(\overline{k}B + \sigma\sqrt{2\ln\frac{12T}{\delta}}\right)^2 t}{1 - (1 - c\eta\rho)} \tag{265}$$

$$= 2\left(\overline{k}B + \sigma\sqrt{2\ln\frac{12T}{\delta}}\right)^2 t. \tag{266}$$

By combining the event $\mathcal{E}_4$ with the above inequality, we obtain the desired inequality.

Finally, we check the assumption $\|\boldsymbol{h}_{t;j} - \boldsymbol{y}_t\|_2 \le \left(\overline{k}B + \sigma\sqrt{2\ln\frac{12T}{\delta}}\right)\sqrt{3t}$. If $\tilde{j} = 0$, $\|\boldsymbol{h}_{t;\tilde{j}} - \boldsymbol{y}_t\|_2 \le \left(\overline{k}B + \sigma\sqrt{2\ln\frac{12T}{\delta}}\right)\sqrt{3t}$ clearly holds from the event $\mathcal{E}_2$ and $\boldsymbol{h}_{t;0} = \boldsymbol{0}$. Here, by applying the aforementioned arguments, we can also verify $\|\boldsymbol{h}_{t;\tilde{j}} - \boldsymbol{y}_t\|_2 \le \left(\overline{k}B + \sigma\sqrt{2\ln\frac{12T}{\delta}}\right)\sqrt{3t}$ for $\tilde{j} = 1$. Repeating the same arguments for $\tilde{j} = 2, 3, \ldots, j$, we obtain the inequality $\|\boldsymbol{h}_{t;j} - \boldsymbol{y}_t\|_2 \le \left(\overline{k}B + \sigma\sqrt{2\ln\frac{12T}{\delta}}\right)\sqrt{3t}$.

$\square$

**Lemma B.11** (Adapted from Lemma B.2 in [41])**.** *Suppose the following inequalities hold:*

$$\overline{R}^4(\overline{R}+2)^4 \le \frac{3\eta^2 M\rho^2}{56 C_{\alpha,\mathcal{D}}^{(3)2}}\left(\overline{k}B + \sigma\sqrt{2\ln\frac{12T}{\delta}}\right)^2\left(\ln\frac{6\cdot 2^{\mathcal{D}+2}M}{\delta}\right)^{-2}, \tag{267}$$

$$\frac{C_{\alpha,\mathcal{D},T}^{(7)}}{\sqrt{M}}\left(\overline{k}B + \sigma\sqrt{2\ln\frac{12T}{\delta}}\right)\left(\ln\frac{6\cdot 2^{\mathcal{D}+2}M}{\delta}\right)\sqrt{\ln\frac{6M}{\delta}} \le 1, \tag{268}$$

$$\eta \le 4^{-1}\left(\rho + 2(2\tilde{R}+2^{\mathcal{D}}\hat{C})^2 T C_{\alpha,\mathcal{D}}^{(4)}\ln\frac{6M}{\delta}\right)^{-1}, \tag{269}$$

*where:*

$$\overline{R} = \tilde{R} + \frac{1}{2\rho}\left[(2\tilde{R}+2^{\mathcal{D}})\sqrt{T C_{\alpha,\mathcal{D}}^{(4)}\ln\frac{6M}{\delta}}\left(\overline{k}B + \sigma\sqrt{2\ln\frac{12T}{\delta}}\right)\sqrt{3T} + \tilde{R}\right], \tag{270}$$

$$\tilde{R} = 2\left(\overline{k}B + \sigma\sqrt{2\ln\frac{12T}{\delta}}\right)\sqrt{\frac{T}{\rho}}. \tag{271}$$

*Furthermore, we set $C_{\alpha,\mathcal{D},T}^{(7)}$ as*

$$C_{\alpha,\mathcal{D},T}^{(7)} = 2\sqrt{3}T^{3/2}\rho^{-3/2}\sqrt{C_{\alpha,\mathcal{D}}^{(5)}} + 16T^2\rho^{-2}(\tilde{R}+2)^2(2+2^{\mathcal{D}}\hat{C})C_{\alpha,\mathcal{D}}^{(3)}\sqrt{C_{\alpha,\mathcal{D}}^{(4)}}. \tag{272}$$

*Then, under the events $\mathcal{E}_2$, $\mathcal{E}_3$, and $\mathcal{E}_4$, the following inequalities hold for any $t \in [T]$ and $j \in [J]$:*

$$\|\boldsymbol{\theta}_{t;j} - \boldsymbol{\theta}_0\|_2 \le 2\left(\overline{k}B + \sigma\sqrt{2\ln\frac{12T}{\delta}}\right)\sqrt{\frac{T}{\rho}}, \tag{273}$$

$$\left\|\boldsymbol{\theta}_{t;j} - \tilde{\boldsymbol{\theta}}_{t;j}\right\|_2 \le \frac{C_{\alpha,\mathcal{D},T}^{(7)}}{\sqrt{M}}\left(\overline{k}B + \sigma\sqrt{2\ln\frac{12T}{\delta}}\right)^2\left(\ln\frac{6\cdot 2^{\mathcal{D}+2}M}{\delta}\right)\sqrt{\ln\frac{6M}{\delta}}. \tag{274}$$

*Proof.* We show by induction. Let us define $\boldsymbol{G}_{t;j}$ and $\boldsymbol{h}_{t;j}$ as $\boldsymbol{G}_{t;j} = [\boldsymbol{g}(\boldsymbol{x}_1; \boldsymbol{\theta}_{t;j}), \ldots, \boldsymbol{g}(\boldsymbol{x}_t; \boldsymbol{\theta}_{t;j})] \in \mathbb{R}^{p\times t}$ and $\boldsymbol{h}_{t;j} = [h(\boldsymbol{x}_1; \boldsymbol{\theta}_{t;j}), \ldots, h(\boldsymbol{x}_t; \boldsymbol{\theta}_{t;j})] \in \mathbb{R}^t$, respectively. First, Eqs. (273) and (274) clearly hold if $j = 0$. Next, fix any $j \in [J]$, and suppose that Eqs. (273) and (274) hold for any $\tilde{j} < j$. Then, as with Lemma B.2 in [41], we have

$$\left\|\boldsymbol{\theta}_{t;j} - \tilde{\boldsymbol{\theta}}_{t;j}\right\|_2 \le \left\|\left[\boldsymbol{I}_p - 2\eta\left(\rho\boldsymbol{I}_p + \boldsymbol{G}_t\boldsymbol{G}_t^\top\right)\right]\left(\boldsymbol{\theta}_{t;j-1} - \tilde{\boldsymbol{\theta}}_{t;j-1}\right)\right\|_2$$
$$+ 2\eta\left\|(\boldsymbol{G}_{t;j-1} - \boldsymbol{G}_t)(\boldsymbol{h}_{t;j-1} - \boldsymbol{y}_t)\right\|_2 \tag{275}$$
$$+ 2\eta\left\|\boldsymbol{G}_t\left[\boldsymbol{h}_{t;j-1} - \boldsymbol{G}_t^\top(\boldsymbol{\theta}_{t;j-1} - \boldsymbol{\theta}_0)\right]\right\|_2.$$

By resorting the similar argument of the proof of Lemma B.2 in [41], the first term is bounded from above as follows:

$$\left\|\left[\boldsymbol{I}_p - 2\eta\left(\rho\boldsymbol{I}_p + \boldsymbol{G}_t\boldsymbol{G}_t^\top\right)\right]\left(\boldsymbol{\theta}_{t;j-1} - \tilde{\boldsymbol{\theta}}_{t;j-1}\right)\right\|_2 \le (1 - 2\eta\rho)\left\|\boldsymbol{\theta}_{t;j-1} - \tilde{\boldsymbol{\theta}}_{t;j-1}\right\|_2. \tag{276}$$

As for the second term of Eq. (275), from Lemma B.10 and Lemma B.15, we have

$$\left\|(\boldsymbol{G}_{t;j-1} - \boldsymbol{G}_t)(\boldsymbol{h}_{t;j-1} - \boldsymbol{y}_t)\right\|_2 \tag{277}$$

$$\le \|\boldsymbol{G}_{t;j-1} - \boldsymbol{G}_t\|\|\boldsymbol{h}_{t;j-1} - \boldsymbol{y}_t\|_2 \tag{278}$$

$$\le \|\boldsymbol{G}_{t;j-1} - \boldsymbol{G}_t\|_F\|\boldsymbol{h}_{t;j-1} - \boldsymbol{y}_t\|_2 \tag{279}$$

$$\le \sqrt{\sum_{i=1}^t \|\boldsymbol{g}(\boldsymbol{x}_i; \boldsymbol{\theta}_{t;j-1}) - \boldsymbol{g}(\boldsymbol{x}_i; \boldsymbol{\theta}_0)\|_2^2}\sqrt{3t}\left(\overline{k}B + \sigma\sqrt{2\ln\frac{12T}{\delta}}\right) \tag{280}$$

$$\le \sqrt{3}T\tilde{R}M^{-1/2}\left(\overline{k}B + \sigma\sqrt{2\ln\frac{12T}{\delta}}\right)\sqrt{C_{\alpha,\mathcal{D}}^{(5)}\ln\frac{6M}{\delta}}, \tag{281}$$

where:

- Eq. (280) follows from Lemma B.10, the condition (267), and the induction hypothesis.

- Eq. (281) follows from the event $\mathcal{E}_3$ and the induction hypothesis.

Furthermore, from Lemma B.13 and Lemma B.15, we have

$$\left\| \boldsymbol{G}_t \left[ \boldsymbol{h}_{t;j-1} - \boldsymbol{G}_t^\top \left( \boldsymbol{\theta}_{t;j-1} - \boldsymbol{\theta}_0 \right) \right] \right\|_2 \tag{282}$$

$$\leq \|\boldsymbol{G}_t\| \left\| \boldsymbol{h}_{t;j-1} - \boldsymbol{G}_t^\top \left( \boldsymbol{\theta}_{t;j-1} - \boldsymbol{\theta}_0 \right) \right\|_2 \tag{283}$$

$$\leq \frac{2T\tilde{R}^2(\tilde{R}+2)^2(2+2^{\mathcal{D}}\hat{C})C_{\alpha,\mathcal{D}}^{(3)}\left(\ln \frac{6 \cdot 2^{\mathcal{D}+2}M}{\delta}\right)\sqrt{C_{\alpha,\mathcal{D}}^{(4)}\ln\frac{6M}{\delta}}}{\sqrt{M}}. \tag{284}$$

By combining Eqs. (276), (281), and (284) with Eq. (275), we have

$$\left\| \boldsymbol{\theta}_{t;j} - \tilde{\boldsymbol{\theta}}_{t;j} \right\|_2 \tag{285}$$

$$\leq (1-2\eta\rho)\|\boldsymbol{\theta}_{t;j-1} - \tilde{\boldsymbol{\theta}}_{t;j-1}\|_2$$

$$+ 2\sqrt{3}\eta T\tilde{R}M^{-1/2}\left(\overline{k}B + \sigma\sqrt{2\ln\frac{12T}{\delta}}\right)\sqrt{C_{\alpha,\mathcal{D}}^{(5)}\ln\frac{6M}{\delta}} \tag{286}$$

$$+ \frac{4\eta T\tilde{R}^2(\tilde{R}+2)^2(2+2^{\mathcal{D}}\hat{C})C_{\alpha,\mathcal{D}}^{(3)}\left(\ln \frac{6 \cdot 2^{\mathcal{D}+2}M}{\delta}\right)\sqrt{C_{\alpha,\mathcal{D}}^{(4)}\ln\frac{6M}{\delta}}}{\sqrt{M}}$$

$$\leq \sqrt{3}TM^{-1/2}\rho^{-1}\tilde{R}\left(\overline{k}B + \sigma\sqrt{2\ln\frac{12T}{\delta}}\right)\sqrt{C_{\alpha,\mathcal{D}}^{(5)}\ln\frac{6M}{\delta}} \tag{287}$$

$$+ 4T\rho^{-1}M^{-1/2}\tilde{R}^2(\tilde{R}+2)^2(2+2^{\mathcal{D}}\hat{C})C_{\alpha,\mathcal{D}}^{(3)}\left(\ln \frac{6 \cdot 2^{\mathcal{D}+2}M}{\delta}\right)\sqrt{C_{\alpha,\mathcal{D}}^{(4)}\ln\frac{6M}{\delta}}$$

$$\leq M^{-1/2}\left(\overline{k}B + \sigma\sqrt{2\ln\frac{12T}{\delta}}\right)^2\left[2\sqrt{3}T^{3/2}\rho^{-3/2}\sqrt{C_{\alpha,\mathcal{D}}^{(5)}\ln\frac{6M}{\delta}}\right. \tag{288}$$

$$\left. + 16T^2\rho^{-2}(\tilde{R}+2)^2(2+2^{\mathcal{D}}\hat{C})C_{\alpha,\mathcal{D}}^{(3)}\left(\ln \frac{6 \cdot 2^{\mathcal{D}+2}M}{\delta}\right)\sqrt{C_{\alpha,\mathcal{D}}^{(4)}\ln\frac{6M}{\delta}}\right]$$

$$\leq \left(\overline{k}B + \sigma\sqrt{2\ln\frac{12T}{\delta}}\right)\sqrt{\frac{T}{\rho}}, \tag{289}$$

where Eq. (289) follows from the condition (268). From the triangle inequality and Lemma B.9,

$$\|\boldsymbol{\theta}_{t;j} - \boldsymbol{\theta}_0\|_2 \leq \left\|\tilde{\boldsymbol{\theta}}_{t;j} - \boldsymbol{\theta}_0\right\|_2 + \left\|\tilde{\boldsymbol{\theta}}_{t;j} - \boldsymbol{\theta}_{t;j}\right\|_2 \tag{290}$$

$$\leq 2\left(\overline{k}B + \sigma\sqrt{2\ln\frac{12T}{\delta}}\right)\sqrt{\frac{T}{\rho}}. \tag{291}$$

$\square$

**Lemma B.12** (Adapted from Lemma C.1 in [24]). *Under the event $\mathcal{E}_5$, the following inequality holds for any $t \in \mathbb{N}_+$:*

$$\tilde{\gamma}_t \leq \gamma_t + \frac{t\sqrt{tC_{\alpha,\mathcal{D}}^{(6)}\ln(96|\mathcal{X}|^2/\delta)}}{\rho\sqrt{M}}, \tag{292}$$

*where the constant $C_{\alpha,\mathcal{D}}^{(6)}$ is defined in Lemma B.6.*

*Proof.* Fix any $\boldsymbol{x}_1, \ldots, \boldsymbol{x}_t \in \mathcal{X}$. Then,

$$\frac{1}{2} \ln \det \left( \boldsymbol{I}_t + \rho^{-1} \boldsymbol{G}_t^\top \boldsymbol{G}_t \right) \tag{293}$$

$$= \frac{1}{2} \ln \det \left[ \boldsymbol{I}_t + \rho^{-1} \boldsymbol{K}_t + \rho^{-1} \left( \boldsymbol{G}_t^\top \boldsymbol{G}_t - \boldsymbol{K}_t \right) \right] \tag{294}$$

$$\leq \frac{1}{2} \ln \det \left( \boldsymbol{I}_t + \rho^{-1} \boldsymbol{K}_t \right) + \frac{1}{2\rho} \left\langle \left( \boldsymbol{I}_t + \rho^{-1} \boldsymbol{K}_t \right)^{-1}, \boldsymbol{G}_t^\top \boldsymbol{G}_t - \boldsymbol{K}_t \right\rangle \tag{295}$$

$$\leq \gamma_t + \frac{1}{2\rho} \left\| \left( \boldsymbol{I}_t + \rho^{-1} \boldsymbol{K}_t \right)^{-1} \right\|_F \left\| \boldsymbol{G}_t^\top \boldsymbol{G}_t - \boldsymbol{K}_t \right\|_F \tag{296}$$

$$\leq \gamma_t + \frac{\sqrt{t}}{2\rho} \sqrt{\sum_{\boldsymbol{x}, \tilde{\boldsymbol{x}} \in \{\boldsymbol{x}_1, \ldots, \boldsymbol{x}_t\}} |k_{\text{TNTK}}(\boldsymbol{x}, \tilde{\boldsymbol{x}}) - \langle g(\boldsymbol{x}, \boldsymbol{\theta}_0), g(\tilde{\boldsymbol{x}}, \boldsymbol{\theta}_0) \rangle|^2} \tag{297}$$

$$\leq \gamma_t + \frac{t \sqrt{t C_{\alpha, \mathcal{D}}^{(6)} \ln(96 |\mathcal{X}|^2 / \delta)}}{\rho \sqrt{M}}, \tag{298}$$

where:

- Eq. (295) follows from the concavity of $\ln \det(\cdot)$ and the fact that $\nabla_{\boldsymbol{X}} \ln \det \boldsymbol{X} = \boldsymbol{X}^{-1}$ holds for any symmetric matrix $\boldsymbol{X}$. In Eq. (295), $\langle \cdot, \cdot \rangle$ represents the matrix inner product.

- Eq. (296) follows from the definition of $\gamma_t$.

- Eq. (297) follows from $\left\| \left( \boldsymbol{I}_t + \rho^{-1} \boldsymbol{K}_t \right)^{-1} \right\|_F \leq \left\| \boldsymbol{I}_t^{-1} \right\|_F \leq \sqrt{t}$.

- Eq. (298) follows from the event $\mathcal{E}_5$.

$\square$

**Lemma B.13.** *Let us define $\boldsymbol{G}_t(\boldsymbol{\theta})$ as $\boldsymbol{G}_t(\boldsymbol{\theta}) = (g(\boldsymbol{x}_1; \boldsymbol{\theta}), \ldots, g(\boldsymbol{x}_t; \boldsymbol{\theta}))^\top \in \mathbb{R}^{p \times t}$. Then, under the event $\mathcal{E}_2$, the following inequality holds for any $R \geq 0$, $t \in \mathbb{N}_+$, and $\boldsymbol{\theta} \in \mathbb{R}^p$ such that $\|\boldsymbol{\theta} - \boldsymbol{\theta}_0\|_2 \leq R$:*

$$\|\boldsymbol{G}_t(\boldsymbol{\theta})\|_F \leq (2R + 2^{\mathcal{D}} \hat{C}) \sqrt{t C_{\alpha, \mathcal{D}}^{(4)} \ln \frac{6M}{\delta}}, \tag{299}$$

*where the constants $\hat{C}$ and $C_{\alpha, \mathcal{D}}^{(4)}$ are defined in Lemma B.4.*

*Proof.* From the definition of $\| \cdot \|_F$, we have

$$\|\boldsymbol{G}_t(\boldsymbol{\theta})\|_F^2 = \sum_{i=1}^t \|g(\boldsymbol{x}_i; \boldsymbol{\theta})\|_2^2 \tag{300}$$

$$\leq t C_{\alpha, \mathcal{D}}^{(4)} (2R + 2^{\mathcal{D}} \hat{C})^2 \ln \frac{6M}{\delta}. \tag{301}$$

Here, the last inequality follows from the condition $\mathcal{E}_2$. $\square$

**Lemma B.14.** *Under the event $\mathcal{E}_2$, the following inequality holds for any $t \in \mathbb{N}_+$:*

$$\left\| \rho \boldsymbol{I}_p + \boldsymbol{G}_t \boldsymbol{G}_t^\top \right\| \leq \rho + t C_{\alpha, \mathcal{D}}^{(4)} \hat{C}^2 2^{\mathcal{D}} \ln \frac{6M}{\delta}, \tag{302}$$

*where the constants $\hat{C}$ and $C_{\alpha, \mathcal{D}}^{(4)}$ are defined in Lemma B.4.*

*Proof.* Under the event $\mathcal{E}_2$, we have

$$\left\|\rho \boldsymbol{I}_p + \boldsymbol{G}_t \boldsymbol{G}_t^\top\right\| = \rho + \left\|\boldsymbol{G}_t \boldsymbol{G}_t^\top\right\| \tag{303}$$

$$\leq \rho + \sum_{i=1}^t \left\|\boldsymbol{g}(\boldsymbol{x}_i; \boldsymbol{\theta}_0)\boldsymbol{g}(\boldsymbol{x}_i; \boldsymbol{\theta}_0)^\top\right\| \tag{304}$$

$$\leq \rho + \sum_{i=1}^t \left\|\boldsymbol{g}(\boldsymbol{x}_i; \boldsymbol{\theta}_0)\boldsymbol{g}(\boldsymbol{x}_i; \boldsymbol{\theta}_0)^\top\right\|_F \tag{305}$$

$$= \rho + \sum_{i=1}^t \left\|\boldsymbol{g}(\boldsymbol{x}_i; \boldsymbol{\theta}_0)\right\|_2^2 \tag{306}$$

$$\leq \rho + t C_{\alpha,\mathcal{D}}^{(4)} 2^{\mathcal{D}} \hat{C}^2 \ln \frac{6M}{\delta}, \tag{307}$$

where:

- Eq. (304) follows from $\boldsymbol{G}_t \boldsymbol{G}_t^\top = \sum_{i=1}^t \boldsymbol{g}(\boldsymbol{x}_i; \boldsymbol{\theta}_0)\boldsymbol{g}(\boldsymbol{x}_i; \boldsymbol{\theta}_0)^\top$ and the triangle inequality.
- Eq. (306) follows from the fact that $\|\boldsymbol{x}\boldsymbol{x}^\top\|_F = \|\boldsymbol{x}\|_2^2$ holds for any $\boldsymbol{x}$.
- Eq. (307) follows from the event $\mathcal{E}_2$.

$\square$

**Lemma B.15.** *Under the event $\mathcal{E}_1$, the following inequality holds for any $\boldsymbol{x} \in \mathbb{S}^{d-1}$, $R \geq 0$, and $\tilde{\boldsymbol{\theta}}, \hat{\boldsymbol{\theta}} \in \mathbb{R}^p$ such that $\|\tilde{\boldsymbol{\theta}} - \boldsymbol{\theta}_0\|_2 \leq R$ and $\|\hat{\boldsymbol{\theta}} - \boldsymbol{\theta}_0\|_2 \leq R$:*

$$|h(\boldsymbol{x}; \tilde{\boldsymbol{\theta}}) - h(\boldsymbol{x}; \hat{\boldsymbol{\theta}}) - \langle \boldsymbol{g}(\boldsymbol{x}; \hat{\boldsymbol{\theta}}), \tilde{\boldsymbol{\theta}} - \hat{\boldsymbol{\theta}}\rangle| \leq \frac{2R^2 C_{\alpha,\mathcal{D}}^{(3)}(R+2)^2}{\sqrt{M}} \ln \frac{6 \cdot 2^{\mathcal{D}+2}M}{\delta}. \tag{308}$$

*where the constant $C_{\alpha,\mathcal{D}}^{(3)}$ is defined in Lemma 3.3.*

*Proof.* From Taylor's theorem, there exists $a \in [0, 1]$ such that

$$\left|h(\boldsymbol{x}; \tilde{\boldsymbol{\theta}}) - h(\boldsymbol{x}; \hat{\boldsymbol{\theta}}) - \langle \boldsymbol{g}(\boldsymbol{x}; \hat{\boldsymbol{\theta}}), \tilde{\boldsymbol{\theta}} - \hat{\boldsymbol{\theta}}\rangle\right| \tag{309}$$

$$= \frac{1}{2}\left|(\tilde{\boldsymbol{\theta}} - \hat{\boldsymbol{\theta}})^\top \boldsymbol{H}\left(\boldsymbol{x}, a\tilde{\boldsymbol{\theta}} + (1-a)\hat{\boldsymbol{\theta}}\right)(\tilde{\boldsymbol{\theta}} - \hat{\boldsymbol{\theta}})\right| \tag{310}$$

$$\leq \frac{1}{2}\left\|\tilde{\boldsymbol{\theta}} - \hat{\boldsymbol{\theta}}\right\|_2 \left\|\boldsymbol{H}\left(\boldsymbol{x}, a\tilde{\boldsymbol{\theta}} + (1-a)\hat{\boldsymbol{\theta}}\right)(\tilde{\boldsymbol{\theta}} - \hat{\boldsymbol{\theta}})\right\|_2 \tag{311}$$

$$\leq \frac{1}{2}\left\|\tilde{\boldsymbol{\theta}} - \hat{\boldsymbol{\theta}}\right\|_2^2 \left\|\boldsymbol{H}\left(\boldsymbol{x}, a\tilde{\boldsymbol{\theta}} + (1-a)\hat{\boldsymbol{\theta}}\right)\right\|. \tag{312}$$

Here, from the conditions of $\tilde{\boldsymbol{\theta}}$ and $\hat{\boldsymbol{\theta}}$, we have

$$\left\|\tilde{\boldsymbol{\theta}} - \hat{\boldsymbol{\theta}}\right\|_2 \leq \left\|\tilde{\boldsymbol{\theta}} - \boldsymbol{\theta}_0\right\|_2 + \left\|\boldsymbol{\theta}_0 - \hat{\boldsymbol{\theta}}\right\|_2 \leq 2R \tag{313}$$

and

$$\left\|a\tilde{\boldsymbol{\theta}} + (1-a)\hat{\boldsymbol{\theta}} - \boldsymbol{\theta}_0\right\|_2 \leq a\left\|\tilde{\boldsymbol{\theta}} - \boldsymbol{\theta}_0\right\|_2 + (1-a)\left\|\boldsymbol{\theta}_0 - \hat{\boldsymbol{\theta}}\right\|_2 \leq R. \tag{314}$$

Therefore, from the event $\mathcal{E}_1$,

$$\left|h(\boldsymbol{x}; \tilde{\boldsymbol{\theta}}) - h(\boldsymbol{x}; \hat{\boldsymbol{\theta}}) - \langle \boldsymbol{g}(\boldsymbol{x}; \hat{\boldsymbol{\theta}}), \tilde{\boldsymbol{\theta}} - \hat{\boldsymbol{\theta}}\rangle\right| \leq \frac{1}{2}\left\|\tilde{\boldsymbol{\theta}} - \hat{\boldsymbol{\theta}}\right\|_2^2 \left\|\boldsymbol{H}\left(\boldsymbol{x}, a\tilde{\boldsymbol{\theta}} + (1-a)\hat{\boldsymbol{\theta}}\right)\right\| \tag{315}$$

$$\leq \frac{2R^2 C_{\alpha,\mathcal{D}}^{(3)}(R+2)^2}{\sqrt{M}} \ln \frac{6 \cdot 2^{\mathcal{D}+2}M}{\delta}. \tag{316}$$

$\square$

*Proof of Theorem B.1.* Suppose that the events $\mathcal{E}_1$–$\mathcal{E}_6$ hold. As proposed in [30], we decompose the error term $|f(\boldsymbol{x}) - h(\boldsymbol{x}; \boldsymbol{\theta}_t)|$:

$$
\begin{aligned}
&|f(\boldsymbol{x}) - h(\boldsymbol{x}; \boldsymbol{\theta}_t)| \\
&\leq |f(\boldsymbol{x}) - \tilde{\mu}_t(\boldsymbol{x})| + |\tilde{\mu}_t(\boldsymbol{x}) - \langle \boldsymbol{g}(\boldsymbol{x}; \boldsymbol{\theta}_0), \boldsymbol{\theta}_t - \boldsymbol{\theta}_0 \rangle| + |\langle \boldsymbol{g}(\boldsymbol{x}; \boldsymbol{\theta}_0), \boldsymbol{\theta}_t - \boldsymbol{\theta}_0 \rangle - h(\boldsymbol{x}_t; \boldsymbol{\theta}_t)|.
\end{aligned}
\tag{317}
$$

By combining the event $\mathcal{E}_6$ with Lemma B.12, the first term of Eq. (317) is bounded from above as follows:

$$
|f(\boldsymbol{x}) - \tilde{\mu}_t(\boldsymbol{x})| \tag{318}
$$

$$
\leq \left( \sqrt{2}B + \frac{\sigma}{\sqrt{\rho}} \sqrt{2 \left( \tilde{\gamma}_t + \ln \frac{6}{\delta} \right)} \right) \tilde{\sigma}_{t-1}(\boldsymbol{x}) \tag{319}
$$

$$
\leq \left( \sqrt{2}B + \frac{\sigma}{\sqrt{\rho}} \sqrt{2 \left( \gamma_t + \frac{t\sqrt{tC_{\alpha,\mathcal{D}}^{(6)} \ln(96|\mathcal{X}|^2/\delta)}}{\rho\sqrt{M}} + \ln \frac{6}{\delta} \right)} \right) \tilde{\sigma}_{t-1}(\boldsymbol{x}). \tag{320}
$$

Furthermore, we obtain the following inequalities for the second term:

$$
|\tilde{\mu}_t(\boldsymbol{x}) - \langle \boldsymbol{g}(\boldsymbol{x}; \boldsymbol{\theta}_0), \boldsymbol{\theta}_t - \boldsymbol{\theta}_0 \rangle| \tag{321}
$$

$$
= \left| g(\boldsymbol{x}; \boldsymbol{\theta}_0)^\top \left( \rho \boldsymbol{I}_p + \boldsymbol{G}_t \boldsymbol{G}_t^\top \right)^{-1} \boldsymbol{G}_t \boldsymbol{y}_t - \langle \boldsymbol{g}(\boldsymbol{x}; \boldsymbol{\theta}_0), \boldsymbol{\theta}_t - \boldsymbol{\theta}_0 \rangle \right| \tag{322}
$$

$$
\leq \rho^{-1} \left\| \boldsymbol{\theta}_t - \boldsymbol{\theta}_0 - \left( \rho \boldsymbol{I}_p + \boldsymbol{G}_t \boldsymbol{G}_t^\top \right)^{-1} \boldsymbol{G}_t \boldsymbol{y}_t \right\|_2 \left\| \rho \boldsymbol{I}_p + \boldsymbol{G}_t \boldsymbol{G}_t^\top \right\| \tilde{\sigma}_t^2(\boldsymbol{x}) \tag{323}
$$

$$
\leq \rho^{-1} \sqrt{\bar{k}^2 + 4C_{\alpha,\mathcal{D}}^{(2)}} \left\| \boldsymbol{\theta}_t - \boldsymbol{\theta}_0 - \left( \rho \boldsymbol{I}_p + \boldsymbol{G}_t \boldsymbol{G}_t^\top \right)^{-1} \boldsymbol{G}_t \boldsymbol{y}_t \right\|_2 \left\| \rho \boldsymbol{I}_p + \boldsymbol{G}_t \boldsymbol{G}_t^\top \right\| \tilde{\sigma}_t(\boldsymbol{x}) \tag{324}
$$

$$
\begin{aligned}
\leq{}& \rho^{-1} \sqrt{\bar{k}^2 + 4C_{\alpha,\mathcal{D}}^{(2)}} \left[ \frac{C_{\alpha,\mathcal{D},T}^{(7)}}{\sqrt{M}} \left( \bar{k}B + \sigma\sqrt{2\ln \frac{12T}{\delta}} \right)^2 \left( \ln \frac{6 \cdot 2^{\mathcal{D}+2} M}{\delta} \right) \sqrt{\ln \frac{6M}{\delta}} \right. \\
&\left. + (1 - 2\eta\rho)^{J/2} \left( \bar{k}B + \sigma\sqrt{2\ln \frac{12T}{\delta}} \right)^2 \sqrt{\frac{T}{\rho}} \right] \\
&\times \left( \rho + TC_{\alpha,\mathcal{D}}^{(4)} 2^{2\mathcal{D}} \hat{C}^2 \ln \frac{6M}{\delta} \right) \tilde{\sigma}_t(\boldsymbol{x}),
\end{aligned}
\tag{325}
$$

where:

- Eq. (322) follows from the feature space representation of $\tilde{\mu}_t$. Actually, we have $\tilde{\mu}_t(\boldsymbol{x}) = g(\boldsymbol{x}; \boldsymbol{\theta}_0)^\top \boldsymbol{G}_t (\rho \boldsymbol{I}_t + \boldsymbol{G}_t^\top \boldsymbol{G}_t)^{-1} \boldsymbol{y}_t = g(\boldsymbol{x}; \boldsymbol{\theta}_0)^\top \left( \rho \boldsymbol{I}_p + \boldsymbol{G}_t \boldsymbol{G}_t^\top \right)^{-1} \boldsymbol{G}_t \boldsymbol{y}_t$, where the last equality follows from the matrix identity $\boldsymbol{G}_t (\rho \boldsymbol{I}_t + \boldsymbol{G}_t^\top \boldsymbol{G}_t)^{-1} = \left( \rho \boldsymbol{I}_p + \boldsymbol{G}_t \boldsymbol{G}_t^\top \right)^{-1} \boldsymbol{G}_t$ (e.g., Lemma 3 in [29]).

- Eq. (323) follows from the fact that $\langle \boldsymbol{z}_1, \boldsymbol{z}_2 \rangle \leq (\boldsymbol{z}_1^\top A^{-1} \boldsymbol{z}_1) \cdot (\boldsymbol{z}_2^\top A \boldsymbol{z}_2) \leq (\boldsymbol{z}_1^\top A^{-1} \boldsymbol{z}_1) \|A\|_2 \|\boldsymbol{z}_2\|_2$ holds for any positive definite matrix $A \in \mathbb{R}^{p \times p}$ and $\boldsymbol{z}_1, \boldsymbol{z}_2 \in \mathbb{R}^p$.

- Eq. (324) follows from $\tilde{\sigma}_t(\boldsymbol{x}) \leq \sqrt{\tilde{k}(\boldsymbol{x}, \boldsymbol{x})} \leq \sqrt{\bar{k}^2 + 4C_{\alpha,\mathcal{D}}^{(2)}}$, where the last inequality follows from the event $\mathcal{E}_5$.

- Eq. (325) follows from Lemma B.9 and Lemma B.11.

By using Taylor's theorem for the third term, there exist $a \in [0, 1]$ such that:

$$|h(\boldsymbol{x}; \boldsymbol{\theta}_t) - \langle \boldsymbol{g}(\boldsymbol{x}; \boldsymbol{\theta}_0), \boldsymbol{\theta}_t - \boldsymbol{\theta}_0 \rangle| \tag{326}$$

$$= \frac{1}{2} \left| (\boldsymbol{\theta}_t - \boldsymbol{\theta}_0)^\top \boldsymbol{H}(a\boldsymbol{\theta}_t + (1-a)\boldsymbol{\theta}_0)(\boldsymbol{\theta}_t - \boldsymbol{\theta}_0) \right| \tag{327}$$

$$\leq \frac{1}{2} \|\boldsymbol{\theta}_t - \boldsymbol{\theta}_0\|_2 \|\boldsymbol{H}(a\boldsymbol{\theta}_t + (1-a)\boldsymbol{\theta}_0)(\boldsymbol{\theta}_t - \boldsymbol{\theta}_0)\|_2 \tag{328}$$

$$\leq \frac{1}{2} \|\boldsymbol{\theta}_t - \boldsymbol{\theta}_0\|_2^2 \|\boldsymbol{H}(a\boldsymbol{\theta}_t + (1-a)\boldsymbol{\theta}_0)\| \tag{329}$$

$$\leq \frac{72T^2 C_{\alpha,\mathcal{D}}^{(3)}}{\sqrt{M}\rho^2} \left( \bar{k}B + \sigma\sqrt{2\ln\frac{12T}{\delta}} \right)^4 \ln\frac{6 \cdot 2^{\mathcal{D}+2}M}{\delta}, \tag{330}$$

where Eq. (327) follows from $h(\boldsymbol{x}; \boldsymbol{\theta}_0) = 0$ holds for any $\boldsymbol{x} \in \mathcal{X}$, and Eq. (330) follows from Lemma B.11 and the event $\mathcal{E}_1$. Finally, since the events $\mathcal{E}_1$–$\mathcal{E}_6$ holds with probability at least $1 - \delta$ from Lemma B.8, we obtain the desired result by aggregating Eqs. (320), (325), and (330). $\qquad\square$

## C  Proof of Theorem 3.3

**Theorem C.1** (Detailed version of Theorem 3.3). *Suppose that Assumption 3.1 holds. Fix any* $\delta \in (0, 1)$, $\alpha \geq 1$, $\rho > 0$, *and* $\mathcal{D} \geq 2$. *Furthermore, suppose that the number of ensemble* $M$ *is sufficiently large to satisfy Eqs.* (162)–(165). *Then, if the learning rate* $\eta$ *satisfy* $\eta \leq 4^{-1}\left( \rho + 2(2\tilde{R} + 2^{\mathcal{D}}\hat{C})^2 T C_{\alpha,\mathcal{D}}^{(4)} \ln\frac{6M}{\delta} \right)^{-1}$, *with probability at least* $1 - \delta$, *the following inequality holds:*

$$R_T \leq \frac{144T^3 C_{\alpha,\mathcal{D}}^{(3)}}{\sqrt{M}\rho} \left( \bar{k}B + \sigma\sqrt{2\ln\frac{12T}{\delta}} \right)^4 \ln\frac{6 \cdot 2^{\mathcal{D}+2}M}{\delta} \tag{331}$$

$$+ \beta\sqrt{\frac{8T}{\ln(1 + \rho^{-2})} \left( \gamma_T + \frac{T\sqrt{TC_{\alpha,\mathcal{D}}^{(6)} \ln(96|\mathcal{X}|^2/\delta)}}{\rho\sqrt{M}} \right)}, \tag{332}$$

*where* $\beta$ *is defined in Theorem B.1. Furthermore, if* $M$ *and* $J$ *is sufficiently large to satisfy the following additional three conditions:*

$$\frac{144T^3 C_{\alpha,\mathcal{D}}^{(3)}}{\sqrt{M}\rho} \left( \bar{k}B + \sigma\sqrt{2\ln\frac{12T}{\delta}} \right)^4 \ln\frac{6 \cdot 2^{\mathcal{D}+2}M}{\delta} \leq 1, \tag{333}$$

$$\frac{T\sqrt{TC_{\alpha,\mathcal{D}}^{(6)} \ln(96|\mathcal{X}|^2/\delta)}}{\rho\sqrt{M}} \leq 1, \tag{334}$$

$$\rho^{-1}\sqrt{\bar{k}^2 + 4C_{\alpha,\mathcal{D}}^{(2)}} \left[ \frac{C_{\alpha,\mathcal{D},T}^{(7)}}{\sqrt{M}} \left( \bar{k}B + \sigma\sqrt{2\ln\frac{12T}{\delta}} \right)^2 \left( \ln\frac{6 \cdot 2^{\mathcal{D}+2}M}{\delta} \right) \sqrt{\ln\frac{6M}{\delta}} \right.$$
$$\left. + (1 - 2\eta\rho)^{J/2} \left( \bar{k}B + \sigma\sqrt{2\ln\frac{12T}{\delta}} \right)^2 \sqrt{\frac{T}{\rho}} \right] \left( \rho + TC_{\alpha,\mathcal{D}}^{(4)} 2^{2\mathcal{D}}\hat{C}^2 \ln\frac{6M}{\delta} \right) \leq 1, \tag{335}$$

*then,*

$$R_T \leq 1 + \left( \sqrt{2}B + 1 + \frac{\sigma}{\sqrt{\rho}}\sqrt{2\left( \gamma_T + 1 + \ln\frac{6}{\delta} \right)} \right) \sqrt{\frac{8T(\gamma_T + 1)}{\ln(1 + \rho^{-2})}}. \tag{336}$$

The proof of Theorem C.1 leverages the following lemma, which describe the relation between the sum of $\tilde{\sigma}_{t-1}(\boldsymbol{x}_t)$ and MIG.

**Lemma C.1** (Lemma 5.3 and Lemma 5.4 in [32]). *Fix any $T \in \mathbb{N}_+$. Then, for any sequence $\boldsymbol{x}_1, \ldots, \boldsymbol{x}_t$, the following inequality holds:*

$$\sum_{t=1}^{T} \tilde{\sigma}_{t-1}(\boldsymbol{x}_t) \leq \sqrt{\frac{8T\tilde{\gamma}_T}{\ln(1 + \rho^{-2})}}. \tag{337}$$

*Proof of Theorem C.1.* From Theorem B.1, with probability at least $1 - \delta$,

$$R_T = \sum_{t=1}^{T} [f(\boldsymbol{x}_t^*) - f(\boldsymbol{x}_t)] \tag{338}$$

$$\leq \frac{144T^3 C_{\alpha,\mathcal{D}}^{(3)}}{\sqrt{M}\rho} \left( \bar{k}B + \sigma\sqrt{2\ln\frac{12T}{\delta}} \right)^4 \ln \frac{6 \cdot 2^{\mathcal{D}+2} M}{\delta} + 2\beta \sum_{t=1}^{T} \tilde{\sigma}_{t-1}(\boldsymbol{x}_t) \tag{339}$$

$$\leq \frac{144T^3 C_{\alpha,\mathcal{D}}^{(3)}}{\sqrt{M}\rho} \left( \bar{k}B + \sigma\sqrt{2\ln\frac{12T}{\delta}} \right)^4 \ln \frac{6 \cdot 2^{\mathcal{D}+2} M}{\delta} + \beta\sqrt{\frac{8T\tilde{\gamma}_T}{\ln(1 + \rho^{-2})}} \tag{340}$$

$$\leq \frac{144T^3 C_{\alpha,\mathcal{D}}^{(3)}}{\sqrt{M}\rho} \left( \bar{k}B + \sigma\sqrt{2\ln\frac{12T}{\delta}} \right)^4 \ln \frac{6 \cdot 2^{\mathcal{D}+2} M}{\delta}$$

$$+ \beta\sqrt{\frac{8T}{\ln(1 + \rho^{-2})} \left( \gamma_T + \frac{T\sqrt{TC_{\alpha,\mathcal{D}}^{(6)} \ln(96|\mathcal{X}|^2/\delta)}}{\rho\sqrt{M}} \right)}, \tag{341}$$

where Eq. (339) follows from the definition of $\boldsymbol{x}_t$, and Eq. (340) follows from Lemma C.1. Furthermore, Eq. (341) follows from Lemma B.12. $\qquad\square$

# D  Proof of Lemma 4.1

*Proof.* Fix any function $f \in \mathcal{H}_{\mathrm{TNTK}}$. According to Mercer's representation theorem (see, e.g., Theorem 4.5.1 in [33]), there exists a sequence $(w_{n,j})$ such that $\sum_{n=0}^{\infty} \sum_{j=1}^{N_{d,n}} w_{n,j}^2 < \infty$ and

$$f(\cdot) = \sum_{n=0}^{\infty} \sum_{j=1}^{N_{d,n}} w_{n,j} \lambda_n^{1/2} Y_{n,j}(\cdot). \tag{342}$$

Furthermore, the RKHS norm $\|f\|_{\mathrm{TNTK}}$ is obtained as $\|f\|_{\mathrm{TNTK}}^2 = \sum_{n=0}^{\infty} \sum_{j=1}^{N_{d,n}} w_{n,j}^2$.

Note that, similar to the TNTK, the ReLU-based NTK can be expanded using spherical harmonics $(Y_{n,j})$ as follows (see, e.g., [35]):

$$k_{\mathrm{NTK}}(\boldsymbol{x}, \tilde{\boldsymbol{x}}) = \sum_{n=0}^{\infty} \sum_{j=1}^{N_{d,n}} \tilde{\lambda}_n Y_{n,j}(\boldsymbol{x}) Y_{n,j}(\tilde{\boldsymbol{x}}), \tag{343}$$

where $(\tilde{\lambda}_n)_{n \in \mathbb{N}}$ are the eigenvalues of the NTK. Here, the function $f$ can be written as

$$f(\cdot) = \sum_{n=0}^{\infty} \sum_{j=1}^{N_{d,n}} w_{n,j} \left( \sqrt{\frac{\lambda_n}{\tilde{\lambda}_n}} \right) \lambda_n^{1/2} Y_{n,j}(\cdot). \tag{344}$$

By noting both TNTK and NTK can be expanded by $(Y_{n,j})$, the following equation holds from Mercer's representation theorem:

$$\|f\|_{\mathrm{NTK}}^2 = \sum_{n=0}^{\infty} \sum_{j=1}^{N_{d,n}} w_{n,j}^2 \frac{\lambda_n}{\tilde{\lambda}_n}. \tag{345}$$

According to Bietti and Bach [6], $\tilde{\lambda}_n = \Theta(n^{-d})$ and there exists a constant $C_{d,L} > 0$ such that $C_{d,L}n^{-d} \leq \tilde{\lambda}_n$. Combining this with Lemma 3.1, we have:

$$\|f\|_{\mathrm{NTK}}^2 \leq \sum_{n=0}^{\infty} \sum_{j=1}^{N_{d,n}} w_{n,j}^2 C_{\alpha,\mathcal{D}}^{(1)} C_{d,L}^{-1} n^d \exp\left(-\ln\left(1 + \frac{1}{4\alpha^2}\right)\mathcal{D}n\right). \tag{346}$$

Since $n^d \exp\left(-\ln\left(1 + \frac{1}{4\alpha^2}\right)\mathcal{D}n\right) \to 0$ (as $n \to \infty$), there exists a constant $C_{\alpha,d} > 0$ such that $n^d \exp\left(-\ln\left(1 + \frac{1}{4\alpha^2}\right)\mathcal{D}n\right) \leq C_{\alpha,d}$ holds for any $n \in \mathbb{N}$. Thus,

$$\|f\|_{\mathrm{NTK}}^2 \leq C_{\alpha,\mathcal{D}}^{(1)} C_{d,L}^{-1} C_{\alpha,d} \sum_{n=0}^{\infty} \sum_{j=1}^{N_{d,n}} w_{n,j}^2 = C_{\alpha,\mathcal{D}}^{(1)} C_{d,L}^{-1} C_{\alpha,d} \|f\|_{\mathrm{TNTK}}^2 < \infty. \tag{347}$$

From the above, it follows that $\mathcal{H}_{\mathrm{TNTK}} \subset \mathcal{H}_{\mathrm{NTK}}$. $\qquad\square$

# E  Helper Lemmas

**Definition E.1** (Sub-Gaussian norm, Definition 2.5.6 in [39])**.** *Let $X$ be a real-valued random variable. Then, the following quantity $\|X\|_{\psi_2}$ is called the sub-Gaussian norm of $X$:*

$$\|X\|_{\psi_2} = \inf\left\{t \geq 0 \,\middle|\, \mathbb{E}\left[\exp\left(\frac{X^2}{t^2}\right)\right] \leq 2\right\}. \tag{348}$$

*Moreover, if $\|X\|_{\psi_2} < \infty$ holds, we call the random variable $X$ a sub-Gaussian random variable.*

**Definition E.2** (Sub-exponential norm, Definition 2.7.5 in [39])**.** *Let $X$ be a real-valued random variable. Then, the following quantity $\|X\|_{\psi_1}$ is called the sub-exponential norm of $X$:*

$$\|X\|_{\psi_1} = \inf\left\{t \geq 0 \,\middle|\, \mathbb{E}\left[\exp\left(\frac{|X|}{t}\right)\right] \leq 2\right\}. \tag{349}$$

*Moreover, if $\|X\|_{\psi_1} < \infty$ holds, we call the random variable $X$ a sub-exponential random variable.*

**Lemma E.1** (General Hoeffding's inequality, Theorem 2.6.2 in [39])**.** *Let $X_1, \ldots, X_N$ be independent, mean-zero, sub-Gaussian random variables. Then, for every $t \geq 0$, the following holds:*

$$\mathbb{P}\left(\left|\sum_{i=1}^{N} X_i\right| \geq t\right) \leq 2\exp\left(-\frac{ct^2}{\sum_{i=1}^{N} \|X\|_{\psi_2}^2}\right), \tag{350}$$

*where $c > 0$ is an absolute constant.*

**Lemma E.2** (Bernstain's inequality, Theorem 2.8.1 in [39])**.** *Let $X_1, \ldots, X_N$ be independent, mean-zero, sub-exponential random variables. Then, for every $t \geq 0$, the following holds:*

$$\mathbb{P}\left(\left|\sum_{i=1}^{N} X_i\right| \geq t\right) \leq 2\exp\left(-c\min\left\{\frac{t^2}{\sum_{i=1}^{N} \|X_i\|_{\psi_1}^2}, \frac{t}{\max_{i \in [N]} \|X_i\|_{\psi_1}}\right\}\right), \tag{351}$$

*where $c > 0$ is an absolute constant.*

**Lemma E.3** (Centering, Lemma 2.6.8 and Exercise 2.7.10 in [39])**.** *For any sub-Gaussian random variable $X$, $\|X - \mathbb{E}[X]\|_{\psi_2} \leq C\|X\|_{\psi_2}$ holds. Furthermore, for any sub-exponential random variable $Y$, $\|Y - \mathbb{E}[Y]\|_{\psi_1} \leq C\|Y\|_{\psi_1}$ holds, where $C > 0$ is an absolute constant.*

**Lemma E.4** (Product of sub-Gaussians is sub-exponential, Lemma 2.7.7 in [39])**.** *Let $X$ and $Y$ be sub-Gaussian random variables. Then, $XY$ is a sub-exponential random variable whose sub-exponential norm satisfies $\|XY\|_{\psi_1} \leq \|X\|_{\psi_2}\|Y\|_{\psi_2}$.*

# F  Details of numerical experiments

## F.1  Our implementation of algorithms

Here, we provide additional information on the implementation of the ST-UCB and NN-UCB algorithms. Our implementation includes the following three simplifications:

(a) In the calculation of the gradient in line 5 of Algorithm 1, we use $\boldsymbol{g}(\boldsymbol{x}; \boldsymbol{\theta}_{t-1})$ from the previous round, rather than the initial gradient $\boldsymbol{g}(\boldsymbol{x}; \boldsymbol{\theta}_0)$.

(b) In the regularization of parameters in line 3 of Algorithm 3, we do not consider the residual from the initial parameters $\boldsymbol{\theta}_0$. In other words, we apply L2 regularization directly to the parameters themselves.

(c) Instead of initializing $\boldsymbol{\theta}_0$ as described in Sec. 3.1, we initialized $\boldsymbol{\theta}_0$ by the Glorot's uniform initializer [17].

It should be noted that the simplification (a) is the same implementation as the original NN-UCB, while the other simplifications are for the sake of simplicity in implementation. We train two models (ST, NN) using stochastic gradient descent (SGD) with a momentum term. The learning rate and the momentum are set to 0.01 and 0.9, respectively. When the momentum is greater than zero, past gradients are considered as a weighted average. SGD is performed in all rounds, with a mini-batch size of 64 and 5 epochs, and we do not use early stopping.

## F.2 Parameter sensitivity

In the results shown in Fig. 2 of the experimental section, we presented the outcomes with optimal hyperparameters of $\epsilon, \beta$. Here, the experimental results for each parameter $\epsilon \in \{0.05, 0.1, 0.2\}, \beta \in \{0.01, 0.1, 1\}$ are summarized in Fig. 3 (real-world dataset) and Fig. 4 (synthetic dataset). In most cases with the real-world dataset, as the rounds progressed, ST-UCB demonstrated better performance than NN-UCB, and UCB-based policies outperformed $\epsilon$-greedy based policies when $\beta = 0.01$. In the $f^{(1)}$ setting for the synthetic data, the regret of ST-UCB converged the fastest. On the other hand, in the $f^{(2)}$ and $f^{(3)}$ settings, NN-UCB sometimes performed well, however the trend of cumulative regret over the rounds was comparable between ST-UCB and NN-UCB.

## G  Summary of the existing works

### G.1  Derivation of TNTK

Our analysis relies on the TNTK derived by Kanoh and Sugiyama [21]. From the definition of the soft tree ensemble model, we have

$$\langle \nabla_{\boldsymbol{\theta}} h(\boldsymbol{x}; \boldsymbol{\theta}), \nabla_{\boldsymbol{\theta}} h(\boldsymbol{x}; \boldsymbol{\theta}) \rangle = \frac{1}{M} \sum_{m=1}^{M} \langle \nabla_{\boldsymbol{\theta}^{(m)}} \tilde{h}(\boldsymbol{x}; \boldsymbol{\theta}^{(m)}), \nabla_{\boldsymbol{\theta}^{(m)}} \tilde{h}(\boldsymbol{x}; \boldsymbol{\theta}^{(m)}) \rangle. \tag{352}$$

If $\boldsymbol{\theta} \sim \mathcal{N}(\mathbf{0}, \boldsymbol{I}_p)$, $(\langle \nabla_{\boldsymbol{\theta}^{(m)}} \tilde{h}(\boldsymbol{x}; \boldsymbol{\theta}^{(m)}), \nabla_{\boldsymbol{\theta}^{(m)}} \tilde{h}(\boldsymbol{x}; \boldsymbol{\theta}^{(m)}) \rangle)_{m \in [M]}$ is mutually independent; therefore, from the law of large number, the inner product $\langle \nabla_{\boldsymbol{\theta}} h(\boldsymbol{x}; \boldsymbol{\theta}), \nabla_{\boldsymbol{\theta}} h(\boldsymbol{x}; \boldsymbol{\theta}) \rangle$ converges to $\mathbb{E}[\langle \nabla_{\boldsymbol{\theta}^{(m)}} \tilde{h}(\boldsymbol{x}; \boldsymbol{\theta}^{(m)}), \nabla_{\boldsymbol{\theta}^{(m)}} \tilde{h}(\boldsymbol{x}; \boldsymbol{\theta}^{(m)}) \rangle]$ in probability as $M \to \infty$. Kanoh and Sugiyama [21] shows that $\mathbb{E}[\langle \nabla_{\boldsymbol{\theta}^{(m)}} \tilde{h}(\boldsymbol{x}; \boldsymbol{\theta}^{(m)}), \nabla_{\boldsymbol{\theta}^{(m)}} \tilde{h}(\boldsymbol{x}; \boldsymbol{\theta}^{(m)}) \rangle]$ equals the expression in Eq. (1) by relying on the recursive expressions of the soft tree (such as Eq. (81)).

### G.2  MIG and effective dimension

As described in Section 3.2, MIG is commonly used as the problem complexity parameter of the kernel-based decision-making problem. On the other hand, instead of MIG, some existing works quantify the problem complexity based on the following *effective dimension* $\tilde{d}$ [10, 37, 40, 41]:

$$\tilde{d} = \mathrm{Tr}(\boldsymbol{K}_T(\boldsymbol{K}_T + \rho \boldsymbol{I}_T)^{-1}). \tag{353}$$

Due to the following inequality [9, 10], the MIG is bounded from above by the worst-case effective dimension up to logarithmic scale:

$$\ln \det(\rho^{-1} \boldsymbol{K}_T + \boldsymbol{I}_T) \leq \mathrm{Tr}(\boldsymbol{K}_T(\boldsymbol{K}_T + \rho \boldsymbol{I}_T)^{-1})(1 + \ln(\rho^{-1} \|\boldsymbol{K}_T\| + 1)). \tag{354}$$

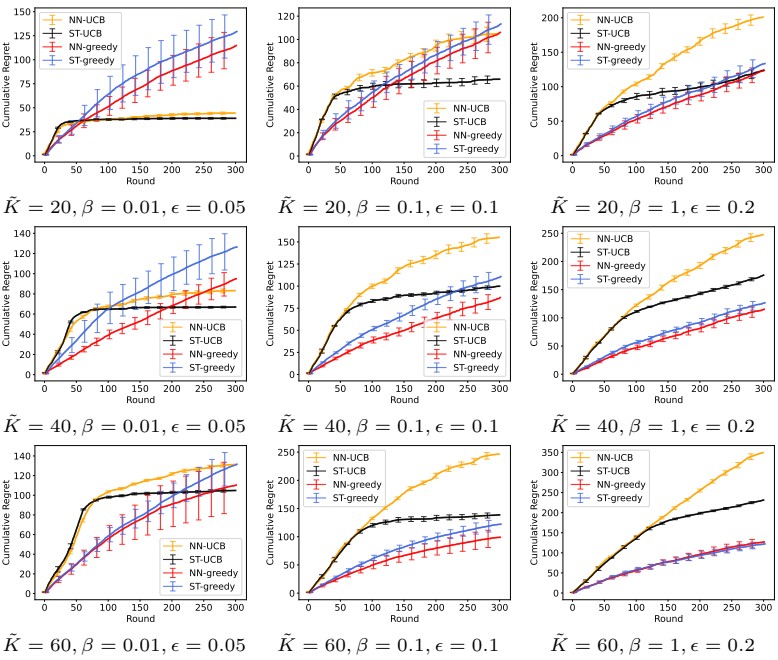

Figure 3: The average cumulative regret with one standard error in the real-world dataset.

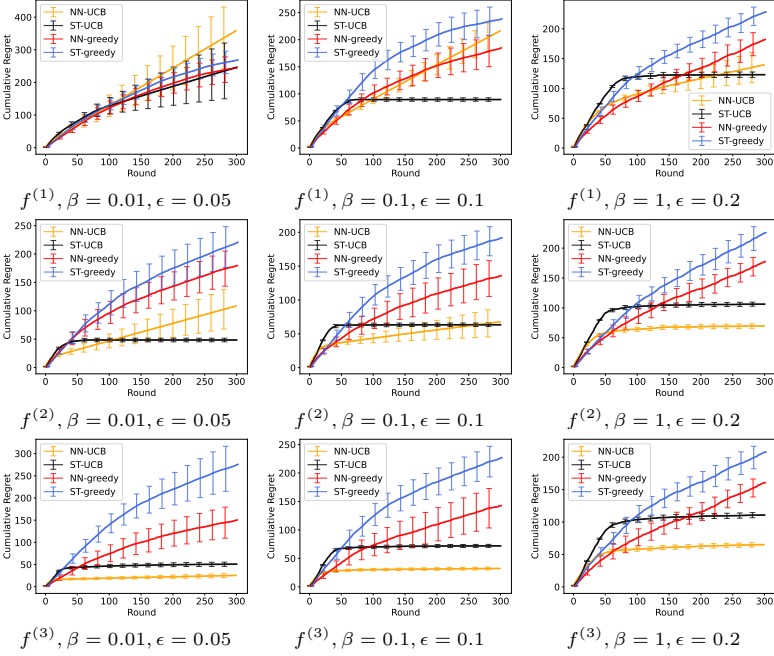

Figure 4: The average cumulative regret with one standard error in the synthetic dataset.

