# OpenReview forum: "No-Regret Bandit Exploration based on Soft Tree Ensemble Model"
_NeurIPS.cc/2024/Conference — NeurIPS 2024 poster_

### Official Review · Reviewer_X7L7 · 2024-06-21

**Soundness:** 3
**Presentation:** 2
**Contribution:** 3
**Rating:** 5
**Confidence:** 3

**Summary:**

This thesis presents a stochastic bandit algorithm ST-UCB based on the soft-tree ensemble model for reward estimation and regret minimization. This algorithm exploits the properties of the soft-tree model and extends the neural robber theory to a tree-based structure, proving that under appropriate assumptions, ST-UCB achieves lower cumulative regrets, demonstrating superiority over the traditional robber algorithm based on ReLU neural networks. In addition, this paper explores the theoretical connection between soft and hard trees, and proposes the prospect of tree-based modeling in complex data analysis.

**Strengths:**

1. The research applies neural bandit (NB) theory to non-neural network models, which extends the applicability of stochastic bandit algorithms and also provides new regret minimization algorithms for tree-based models.
2. The paper not only has a proof of theory and an in-depth analysis of the properties of the soft tree ensemble model, but also conducts some experiments to verify the performance of the algorithm.

**Weaknesses:**

Although the paper proves that the ST-UCB algorithm achieves regret-free performance under specific regularity conditions, these conditions may be difficult to satisfy or validate in practical applications.

**Questions:**

It is recommended to explore the applicability of the algorithm under a wider range of conditions and to focus on the efficiency of the soft-tree model, taking into account the complex structure and high computational demands that may be involved.

**Limitations:**

Authors have discussed the limitations.

---

> ### Author Rebuttal · Authors · 2024-08-06
>
> We thank the reviewer for their comments and suggestions.
>
> > Although the paper proves that the ST-UCB algorithm achieves regret-free performance under specific regularity conditions, these conditions may be difficult to satisfy or validate in practical applications.
>
> > It is recommended to explore the applicability of the algorithm under a wider range of conditions and to focus on the efficiency of the soft-tree model, taking into account the complex structure and high computational demands that may be involved.
>
> We will make an effort to consider a wider range of applicability in future work.
> However, since the theoretical assumptions (Assumption 3.1) of our analysis rely only on the standard set of assumptions in the existing neural bandits literature (as described in Remark 1),
> the applicability of our theory is at the same level as that of existing neural bandit algorithms (e.g., NN-UCB [1]).
> Therefore, we believe that the applicability of our algorithm does not diminish the significance of our contributions.
>
> - [1] Zhou, Dongruo, Lihong Li, and Quanquan Gu. "Neural contextual bandits with ucb-based exploration." International Conference on Machine Learning. PMLR, 2020.

---

### Official Review · Reviewer_SWN3 · 2024-07-08

**Soundness:** 4
**Presentation:** 4
**Contribution:** 3
**Rating:** 7
**Confidence:** 3

**Summary:**

The paper proposes an algorithm for multi-armed bandits, where the reward function is estimated with a soft tree ensemble method. The paper present their model as an extension to NN-UCB but with sublinear regret guarantees at the cost of a reduced hypothesis space. The generalized theory of neural tangent kernels, i.e. the tree neural tangent kernel, was utilized in a similar fashion as neural tangent kernel were used in previous works for the NN-UCB algorithm. Numerical results are provided at the end of the paper.

**Strengths:**

The paper proposes a new approach, to the best of my knowledge.

It is shown that their model is an extension to the existing neural bandit case using.

It improves it regret bound to be sublinear  a clear improvement over neural bandits.

A new confidence bound on for their model is provided.

**Weaknesses:**

In line 218 wihtin Theorem 3.2: a concrete dependency on T for the lower bound on M would have been nice, in order to insert it into Theorem 3.2 to see how the confidence bound depens on time. Without that a reader could think that the confidence bound is super linear in T.

Minor errors: in Lines 112 and 113 I think the gradient operators are missing within the inner products. Line 124 should have bold g in the inner product.

**Questions:**

The theory suggests that the regret bound improves with increasing M. Is there a point where M is to large and the tree ensemble becomes too complex?

**Limitations:**

The paper clearly indicates that the imrpovement they make over exisiting work comes at the cost of a reduced hypothesis space.

---

> ### Author Rebuttal · Authors · 2024-08-06
>
> We thank the reviewer for the overall positive assessment of our paper.
> We describe the answers to the question of the reviewer
>
> > Is there a point where M is to large and the tree ensemble becomes too complex?
>
> In theoretical perceptive, the huge $M$ improves the regret even if the complexity of the soft tree models becomes overly large.
> The theoretical superiority observed in such overly complex models is similar to the recent theoretical findings on NTK in regression problems (e.g., [1], [2]).
> On the other hands, in practical perceptive, our regret analysis (Theorem 3.3) need the learner to choose sufficiently small (large) learning rate (step size) of the gradient descent;
> Such theoretical demand makes it difficult to apply ST-UCB algorithm for overly large soft tree models in practice, and we leave additional effort to more computationally efficient algorithm as future work.
> Finally, we would like to note that such high computational demands also exist in current neural bandit algorithms (e.g., NN-UCB [3]). Therefore, we believe that the high computational requirements described above do not diminish the significance of our contributions.
>
> - [1] Jacot, Arthur, Franck Gabriel, and Clément Hongler. "Neural tangent kernel: Convergence and generalization in neural networks." Advances in neural information processing systems 31 (2018).
> - [2] Arora, Sanjeev, et al. "On exact computation with an infinitely wide neural net." Advances in neural information processing systems 32 (2019).
> - [3] Zhou, Dongruo, Lihong Li, and Quanquan Gu. "Neural contextual bandits with ucb-based exploration." International Conference on Machine Learning. PMLR, 2020.

---

> > ### Comment · Reviewer_SWN3 · 2024-08-08
> >
> > I thank the authors for the detailed answer to my question. I decided to leave my score as it is and wish the authors good luck.

---

### Official Review · Reviewer_PrZ2 · 2024-07-10

**Soundness:** 3
**Presentation:** 1
**Contribution:** 3
**Rating:** 5
**Confidence:** 4

**Summary:**

This paper presents the Tree Neutral Tangent Kernel (TNTK) for soft tree models and introduces the ST-UCB algorithm based on this kernel. The authors also provide theoretical guarantees for the kernel and algorithm, including a regret bound. The contribution of this study is substantial; however, many sections require further refinement to enhance the clarity and overall presentation.

**Strengths:**

This paper provides solid theories for Tree Neutral Tangent Kernel (TNTK) and the suggested algorithm. The contribution of these theories seems significant. Also, the experiments demonstrate that the algorithm the ST-UCB algorithm performs well.

**Weaknesses:**

A piece of the main contribution is the introduction of TNTK and its theoretical properties. But, these are not presented well. A more detailed and clear description for the introduction of TNTK seems necessary. Even the definition is missing. The authors should show the derivation steps of TNTK in a more structured way. There are some other problems with presentation:

- The first paragraph in the first section is unclear. How various combinations of users and items can be connected to unobserved actions?
- Lines 42-53 are not clear. The authors should explain the effective dimension more or describe the bounds in terms of the ambient dimension d.
- Citation formats should be corrected.
- For Section 2, lines 107-120 should be clearer. Especially, lines 111-113 seem to have some errors.
- The proofs in Appendices need much work for improved presentations.
- The statement of Lemma 4.1 seems wrong.

**Questions:**

- For the subGaussian assumption in Assumption 3.1, do we need the conditional expectation given H_t-1? Due to the independency of noise, it does not seem to be necessary. If needed, could you please explain?
- The exponential eigenvalue decay is surprising. Could you please provide some intuition on how the improvement can be achieved as compared to NTK?
- What is the key difference between NN-UCB and ST-UCT in their mechanism?

**Limitations:**

The derivation of TNTK does not seem markedly different from that of the Neural Tangent Kernel (NTK), which may slightly weaken the perceived novelty of the study. Nonetheless, the application of mathematical techniques used for NTK to develop TNTK is not straightforward and still represents a significant contribution.

---

> ### Author Rebuttal · Authors · 2024-08-06
>
> We thank the reviewer for their useful feedback and suggestions.
> To address the readability issues that the reviewer kindly pointed out, we will carefully revise our paper as follows:
>
> - The description of TNTK:
>     - In Section 2, we will add the detailed introduction of TNTK, including its definition.
>     - Furthermore, in the Appendix, we will add an independent section that describes the detailed derivation of TNTK.
> - The proofs in Appendices:
>     - We will delete redundant expressions as much as possible to make the proofs more concise and coherent.
>     - Furthermore, we will add brief proof sketches in Appendices A, B.1, and B.2.
> - The description about effective dimension and MIG:
>     - In Section 3, we will add a description of the effective dimension by relating it to MIG.
>     - Furthermore, in Appendix, we will add an independent section that summarizes the effective dimension and MIG.
>
> With these modifications, we believe that the reviewer's concerns can be resolved.
>
> Below, we provide answers to the reviewer's questions.
>
> > For the subGaussian assumption in Assumption 3.1, do we need the conditional expectation given H_t-1? Due to the independency of noise, it does not seem to be necessary. If needed, could you please explain?
>
> Our analysis **does not** assume the independence of noise sequence $(\epsilon_t)$; therefore, our conditional sub-Gaussian assumption is milder than the sub-Gaussian noise assumption with independence.
> Please note that the conditional sub-Gaussian assumption is already utilized in existing works on linear or kernelized bandits (e.g., see [1, 2, 3]).
>
> - [1]Abbasi-Yadkori, Yasin, Dávid Pál, and Csaba Szepesvári. "Improved algorithms for linear stochastic bandits." Advances in neural information processing systems 24 (2011).
> - [2]Abbasi-Yadkori, Yasin. "Online learning for linearly parametrized control problems." (2013).
> - [3]Chowdhury, Sayak Ray, and Aditya Gopalan. "On kernelized multi-armed bandits." International Conference on Machine Learning. PMLR, 2017.
>
> In addition, these existing works also do not assume the independence of the noise sequence.
>
>
>
> > The exponential eigenvalue decay is surprising. Could you please provide some intuition on how the improvement can be achieved as compared to NTK?
>
> The difference in the eigenvalue decay behavior between NTK and TNTK mainly arises from the difference in smoothness between NTK and TNTK.
> Specifically, TNTK is infinitely differentiable over $\mathbb{S}^{d-1} \times \mathbb{S}^{d-1}$, whereas NTK is not (see, e.g., [4]).
> Our analysis shows that TNTK achieves exponential eigendecay by carefully leveraging existing result on the dot product kernel (Definition A.1 and Lemma A.1),
> which fundamentally relies on the infinite differentiability of the underlying kernel.
>
> - [4] Bietti, Alberto, and Francis Bach. "Deep Equals Shallow for ReLU Networks in Kernel Regimes." ICLR 2021-International Conference on Learning Representations. 2021.
>
>
> > What is the key difference between NN-UCB and ST-UCB in their mechanism?
>
> The main difference between NN-UCB and ST-UCB lies in the rate at which the width of the confidence bound converges to zero.
> Specifically, in ST-UCB, the width of the confidence bound given by Theorem 3.2 converges to zero at a rate of $\tilde{O}(1/\sqrt{T})$.
> In contrast, the width of the confidence bound used in NN-UCB generally does not converge to zero unless additional assumptions on $\mathcal{X}_t$ are introduced (Section 4, Lines 253-255).
> This difference in the behavior of the confidence bounds results in the different rates of cumulative regret for ST-UCB and NN-UCB, as discussed in Section 4.
>
>
> Furthermore, we would like to address the comments provided by the reviewer as follows:
>
> > The first paragraph in the first section is unclear. How various combinations of users and items can be connected to unobserved actions?
>
> We will revise this paragraph in revision to improve clarity. The content we intended to convey in this paragraph is as follows:
>
> In situations where the size of the action (arm) candidate set $\mathcal{X}$ can be extremely large compared $T$ to the total number of steps (i.e., $|\mathcal{X}| \gg T$),
> it is difficult to apply conventional finite-armed bandit algorithms that assume the independence of arm rewards.
> Specifically, when assuming the independence of arm rewards,
> it is necessary to observe each arm at least once to estimate its reward; however, it is infeasible to run when $\mathcal{X}$ is enormous.
> Therefore, in such scenarios, it is essential to estimate the rewards of all actions (arms) in,
> including unobserved ones, from limited observational data and to balance the trade-off between exploration and exploitation.
> As an example of such situation, this paper considers a recommendation system where the action candidate set $\mathcal{X}$ is given by the combination of users and items.
> As mentioned in [5], in large-scale recommendation systems with a large number of user or item candidates, the action candidate set $\mathcal{X}$ can become extremely large.
>
>
> Lastly, it should be noted that examples of large-scale arm candidate sets are not limited to recommendation systems.
> Many such examples of applications have been reported in existing works (e.g., [6], [7]), and large $\mathcal{X}$ setting is a common motivation for introducing model-based bandit algorithm like kernel bandits (see, e.g., Introduction in [6]).
>
> - [5]Vanchinathan, Hastagiri P., et al. "Explore-exploit in top-n recommender systems via gaussian processes." Proceedings of the 8th ACM Conference on Recommender systems. 2014.
> - [6]Chowdhury, Sayak Ray, and Aditya Gopalan. "On kernelized multi-armed bandits." International Conference on Machine Learning. PMLR, 2017.
> - [7]Kassraie, Parnian, Andreas Krause, and Ilija Bogunovic. "Graph neural network bandits." Advances in Neural Information Processing Systems 35 (2022): 34519-34531.

---

> > ### Comment · Reviewer_PrZ2 · 2024-08-14
> >
> > I thank the authors for their detailed response.

---

### Official Review · Reviewer_ghBb · 2024-07-15

**Soundness:** 3
**Presentation:** 3
**Contribution:** 2
**Rating:** 5
**Confidence:** 4

**Summary:**

The paper investigates soft tree ensemble model for reward modeling in bandit algorithms.

**Strengths:**

The work provides comprehensive formal evidence for the proposed methods.
The empirical study is also comprehensive.

**Weaknesses:**

Not sure if there is some new insights beyond other similar works that utilized the NTK theory such as Neural Thompson Sampling.

**Questions:**

(1) Could you share more insights on why soft tree ensemble model is of particular interest? In which case we should prefer tree-based reward model than ReLu-based reward model?

**Limitations:**

The paper did discuss the theoretical limitation.

---

> ### Author Rebuttal · Authors · 2024-08-06
>
> We thank the reviewer for their comments.
>
> > Could you share more insights on why soft tree ensemble model is of particular interest? In which case we should prefer tree-based reward model than ReLu-based reward model?
>
> From a practical perspective, the performance of a bandit algorithm depends on how accurately the reward model can represent the underlying data structure.
> Therefore, our soft tree-based algorithm is expected to perform better than the ReLU-based algorithm on tabular data, where existing soft-tree regression works have reported high empirical performance (e.g., [1,2]).
>
> - [1]Popov, Sergei, Stanislav Morozov, and Artem Babenko. "Neural oblivious decision ensembles for deep learning on tabular data." arXiv preprint arXiv:1909.06312 (2019).
>
> - [2]Hazimeh, Hussein, et al. "The tree ensemble layer: Differentiability meets conditional computation." International Conference on Machine Learning. PMLR, 2020.
>
> From a theoretical perspective, our analysis shows that the soft tree-based ST-UCB algorithm has superior performance compared to the ReLU-based NN-UCB algorithm,
> albeit at the cost of a smaller hypothesis space for the soft tree model (see Section 4).
> We believe that this result will motivate practitioners to use our soft tree-based algorithm instead of the ReLU-based algorithm,
> as the superior performance of ST-UCB is guaranteed if the underlying reward function lies within the hypothesis space of the soft tree model.

---

### Author Response · Authors · 2024-08-13

Dear reviewers,

We would like to thank you for taking the time to review the manuscript.
We truly appreciated your valuable comments and suggestions, which helped us improve the clarity of the paper.
We hope that our response resolves the reviewer's initial concerns and clarifies any ambiguous points about our paper.
If you have any further questions or concerns, please let us know. We will make an effort to provide additional responses throughout the duration of the discussion period.

---

### Decision · Program_Chairs · 2024-09-25

**Decision:**

Accept (poster)

**Comment:**

This paper proposes a new bandit algorithm where the reward function is estimated with a soft tree ensemble method. The authors present their model as an extension to NN-UCB but with sublinear regret guarantees at the cost of a reduced hypothesis space. The generalized theory of neural tangent kernels, i.e. the tree neural tangent kernel, was utilized in a similar fashion as neural tangent kernel were used in previous works for the NN-UCB algorithm. Numerical results are provided at the end of the paper.

The reviewers agree that there is more merit in this paper than its limitations. In particular, the proposed soft tree based USB has better theoretical guarantees than NN-UCB, the current state of the art (this, however, comes with a cost).  On the other hand, the key weakness of the paper is its presentation. In its current format there a numerous unclear parts. If this paper gets accepted, I strongly recommend the authors to polish the paper to improve the readability of the text.